# A Two-Phase Deep Learning Framework for Adaptive Time-Stepping in High-Speed Flow Modeling

**Jacob Helwig**[1]    **Sai Sreeharsha Adavi**[2]    **Xuan Zhang**[1]    **Yuchao Lin**[1]
**Felix S. Chim**[2]    **Luke Takeshi Vizzini**[2]    **Haiyang Yu**[1]    **Muhammad Hasnain**[3]
**Saykat Kumar Biswas**[3]    **John J. Holloway**[1]    **Narendra Singh**[2]    **N. K. Anand**[3]
**Swagnik Guhathakurta**[2*]    **Shuiwang Ji**[1, 3*]
[1]Department of Computer Science and Engineering, Texas A&M University
[2]Department of Aerospace Engineering, Texas A&M University
[3]J. Mike Walker '66 Department of Mechanical Engineering, Texas A&M University
{jacob.a.helwig,swagnik,sji}@tamu.edu

## Abstract

We consider the problem of modeling high-speed flows using machine learning methods. While most prior studies focus on low-speed fluid flows in which uniform time-stepping is practical, flows approaching and exceeding the speed of sound exhibit sudden changes such as shock waves. In such cases, it is essential to use adaptive time-stepping methods to allow a temporal resolution sufficient to resolve these phenomena while simultaneously balancing computational costs. Here, we propose a two-phase machine learning method, known as ShockCast, to model high-speed flows with adaptive time-stepping. In the first phase, we propose to employ a machine learning model to predict the timestep size. In the second phase, the predicted timestep is used as an input along with the current fluid fields to advance the system state by the predicted timestep. We explore several physically-motivated components for timestep prediction and introduce timestep conditioning strategies inspired by neural ODE and Mixture of Experts. We evaluate our methods by generating three supersonic flow datasets, available at https://huggingface.co/divelab. Our code is publicly available as part of the AIRS library (https://github.com/divelab/AIRS).

## 1 Introduction

Learning fluid dynamics aims to accelerate fluid modeling using machine learning models (Li et al., 2021; Zhang et al., 2023). Because this is an emerging area of research, most current works focus on low-speed scenarios in which flows are assumed to be incompressible. In such cases, the time scale of dynamics is relatively stable, enabling the use of time-stepping schemes with uniform step sizes without substantially affecting solution quality or the required computational effort. In contrast, time scales vary greatly for high-speed flows such that uniform time-stepping is no longer a tractable strategy. For example, supersonic flow occurs when a fluid moves faster than the local speed of sound (Anderson, 2023; 2020). The speed of such flows is typically characterized by the Mach number $M$, defined as the ratio of the flow velocity $v$ to the local speed of sound $a$ as $M = v/a$. Flows in the supersonic regime (commonly $1 < M < 5$) exhibit distinct phenomena with small time scales, including shock waves, expansion fans, and significant compressibility effects (Anderson, 2020). Hypersonic flow refers to extremely high-speed fluid flows, conventionally defined by Mach numbers greater than 5. In the hypersonic regime, encountered in the design of spacecraft, missiles, and atmospheric reentry vehicles, flows exhibit unique and complex behaviors such as heating, strong shock wave interactions, and chemical reactions.

---

*Equal senior authorship

For both supersonic and hypersonic flows, the time scale required to accurately resolve these phenomena is much smaller than other parts of the dynamics. Therefore, uniform time-stepping is no longer practical, as it would require the use of the smallest time scale for all steps, inflating the required computation prohibitively. Instead, high-speed flow solvers employ adaptive time-stepping schemes which dynamically adjust the timestep size such that smaller steps are taken in the presence of sharp gradients. Adaptive time-stepping can also benefit neural solvers through more balanced objectives. When using a uniform step size, the amount of evolution the model is required to learn can vary greatly between states with sharp gradients and smoother states, which is an especially pertinent consideration for high-speed flows. By instead inversely scaling the step size according to the rate of change, the difficulties of different training pairs are more evenly distributed.

However, because neural solvers achieve speedup through use of coarsened space-time meshes, classical approaches for determining the timestep size are not applicable. We therefore develop ShockCast, a machine learning framework for temporally-adaptive modeling of high-speed flows. ShockCast consists of two phases: a neural CFL phase, where the timestep is predicted, and a neural solver phase, where the flow field is evolved forward in time by the predicted timestep size. We investigate the effect of physically-motivated components in our neural CFL model, and develop several novel timestep conditioning strategies for neural solvers inspired by neural ODE (Chen et al., 2018) and Mixture of Experts (Shazeer et al., 2017). To evaluate our framework, we generate three new high-speed flow datasets modeling a circular blast (maximum Mach numbers vary from $0.49$ to $2.97$ across cases), a multiphase coal dust explosion (Mach numbers of the initial shock vary from $1.2$ to $2.1$), and a shock sweeping over an airfoil (Mach numbers of the initial shock again vary from $1.2$ to $2.1$). Our work takes steps towards developing machine learning models for high-speed flows, where there is great potential for neural acceleration due to the immense computational requirements of classical methods.

## 2 BACKGROUND

In Section 2.1, we briefly introduce how PDEs are solved numerically before describing the role that the CFL condition plays in this task in Section 2.2. We then overview several prominent machine learning approaches for solving PDEs in Section 2.3.

### 2.1 SOLVING TIME-DEPENDENT PARTIAL DIFFERENTIAL EQUATIONS

Time-dependent PDEs are common in engineering, with some of the most prominent applications arising in fluid dynamics. In two spatial dimensions, they typically equate a first derivative in time to some operator $\mathcal{H}$ of spatial derivatives for an unknown solution function $\boldsymbol{u}(x, y, t) \in \mathbb{R}^D$ as

$$\partial_t \boldsymbol{u} = \mathcal{H}(\boldsymbol{u}, \partial_x \boldsymbol{u}, \partial_{xx} \boldsymbol{u}, \partial_y \boldsymbol{u}, \partial_{yy} \boldsymbol{u}, \partial_{xy} \boldsymbol{u}, \ldots), \tag{1}$$

with boundary conditions and initial conditions imposing additional constraints on $\boldsymbol{u}$. To solve a PDE, we need to identify a function $\boldsymbol{u}$ satisfying these constraints. In most real-world applications of PDE modeling, including fluid dynamics, an analytical form of this solution is intractable to obtain, and so we instead rely on producing $\boldsymbol{u}$ in numerical form, that is, obtaining point-wise evaluations on a discrete set of collocation points in space-time. This is done by evolving the PDE forward in time by first approximating the spatial derivatives on the right-hand side of Equation (1) using finite difference methods, finite volume methods, finite element methods (Reddy et al., 2022), or spectral methods (Gottlieb & Orszag, 1977; Gottlieb & Hesthaven, 2001; Canuto et al., 2007; Kopriva, 2009). Plugging these quantities into $\mathcal{H}$ gives $\partial_t \boldsymbol{u}(t)$, which is then time-integrated to advance the solution in time by a step size of $\Delta t$.

### 2.2 THE COURANT–FRIEDRICHS–LEWY CONDITION

Numerical time integrators are very sensitive to the timestep size $\Delta t$. When $\boldsymbol{u}(t)$ is changing rapidly, or more formally, when $\|\partial_t \boldsymbol{u}(t)\|$ grows large, too large $\Delta t$ can lead to divergence of the numerical solution (Anderson, 2023) or nonphysical solutions containing negative densities or pressures (Patkar et al., 2016). In low-speed flows, the time scale does not vary drastically, which is to say that the magnitudes of temporal derivatives do not fluctuate substantially. In such cases, $\Delta t$ can be chosen as a fixed value to match the smallest time scale, simplifying the numerical solution

process. On the other hand, time scales vary greatly in high-speed flows. For example, shock wave interactions in supersonic and hypersonic flows produce extremely sharp spatial gradients that can only be resolved with small timestep sizes. Following the dissipation of these phenomena, the solution can become smoother such that the time scale is substantially larger. Uniform time-stepping schemes, which must maintain a timestep small enough to resolve sharp gradients even in smooth regions, therefore impose greater computational burden compared to *adaptive time-stepping methods*, where timestep sizes are dynamically adjusted according to the rate of change of the solution.

Adaptive time-stepping methods employ the Courant–Friedrichs–Lewy (CFL) Condition (Courant et al., 1967) to determine the timestep size. The CFL condition is a necessary condition on the timestep size to attain convergence of the numerical solution (Bartels, 2016). For a single-phase flow in two spatial dimensions and a target Courant number $C \in (0, 1)$, the condition requires that

$$\Delta t \leq \frac{C}{\lambda_{\max}} \min_{x,y} (\Delta x, \Delta y), \tag{2}$$

where $\min_{x,y}(\Delta x, \Delta y)$ is the minimum cell height and width in the spatial discretization and the maximum wave speed $\lambda_{\max}$ is defined as

$$\lambda_{\max} := \max_{x,y} \lambda(x, y) \qquad \lambda(x, y) := \max\left(|u(x,y)| + a(x,y), |v(x,y)| + a(x,y)\right), \tag{3}$$

where $u$ and $v$ denote the $x$ and $y$ components of the velocity, and $a(x, y) := \sqrt{\gamma R T(x, y)}$ is the local sound speed defined by the ratio of specific heats $\gamma$, the specific gas constant $R$ and the temperature $T$. Intuitively, the CFL condition restricts information flow for stability such that information propagates no more than one cell in any direction per timestep, which can be seen from the scaling by the minimum cell size and inverse scaling by the wave speed in Equation (2).

## 2.3 NEURAL SOLVERS

As previously discussed, numerically solving PDEs is a computationally intensive process. Deep surrogate models which can accelerate the solution of PDEs are therefore of great interest. Various approaches have emerged over the last decade, including Physics-Informed Neural Networks (Raissi et al., 2019), hybrid solvers (Kochkov et al., 2021; Kaneda et al., 2023; Sun et al., 2023), and operator learning (Kovachki et al., 2023). Speedups over classical methods are primarily achieved by the ability of neural solvers to learn solution mappings on coarsened grids in space-time (Stachenfeld et al., 2021; Kochkov et al., 2021). To maintain stability, classical methods require computational grids to be sufficiently fine in time, as specified by the CFL condition, as well as in space. On the other hand, neural methods can learn to map between solutions spaced hundreds of classical solver steps apart on much lower-dimensional spatial discretizations, thereby realizing substantial speedups. Additionally, classical methods require that all $D$ flow variables comprising the PDE be evolved, whereas neural solvers can learn to explicitly model only a subset of variables of interest while implicitly learning to incorporate the effect of the omitted variables.

## 3 METHODS

### 3.1 LEARNING HIGH-SPEED FLOWS

High-speed flows are one of the most resource-intensive applications of PDE modeling and therefore stand to benefit greatly from the speedup offered by neural solvers. In such settings, time-adaptive meshes present a more balanced one-step objective for neural solvers. When using a uniform step size $\Delta t$, the difference between inputs and targets $\|\boldsymbol{u}(t + \Delta t) - \boldsymbol{u}(t)\|$ tends to grow with the rate of change of inputs with respect to time $\|\partial_t \boldsymbol{u}(t)\|$, which follows from the definition of a derivative $\partial_t u(t) := \lim_{h \to 0}(u(t + h) - u(t))/h$. This implies that inputs with sharp gradients are less correlated with targets such that the mapping for sharp-gradient inputs is more difficult to learn compared to smoother ones. By instead following an adaptive scheme to inversely scale $\Delta t$ according to the rate of change of $\boldsymbol{u}(t)$, the difference $\|\boldsymbol{u}(t) - \boldsymbol{u}(t + \Delta t)\|$ is more uniformly distributed across inputs with varying degrees of sharpness in gradients, thereby reducing variance in the training objective. This is a particularly relevant consideration for high-speed flows, where the sharpness of gradients varies greatly throughout time due to phenomena such as shock waves. Furthermore, the ability

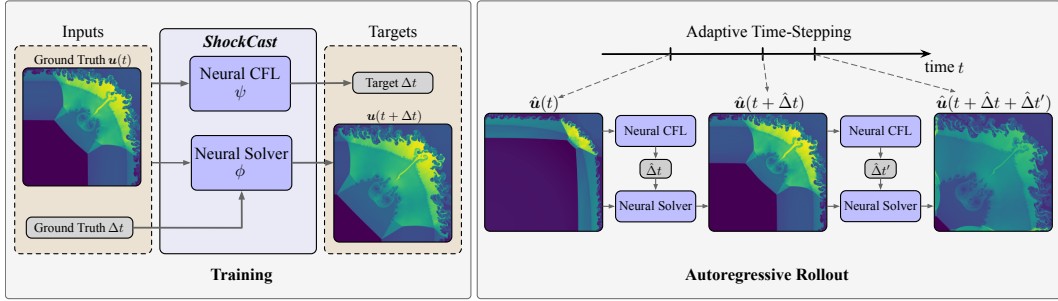

Figure 1: Overview of the ShockCast framework for time-adaptive modeling of high-speed flows. *Left*: Training pipeline. The neural CFL model and time-conditioned neural solver are conditioned on the current flow state and predict the corresponding timestep size $\Delta t$ and flow state $\Delta t$ ahead, respectively. *Right*: Inference pipeline. ShockCast autoregressively alternates between predicting the timestep size given the current flow state using the neural CFL model and evolving the flow state forward in time by the predicted timestep size using the neural solver model. Note that the example data are from the circular blast dataset we generated in this work.

to train neural solvers on time-adaptive meshes allows direct use of solutions produced from high-speed flow solvers without introducing error by interpolating to a uniform grid or modifying solver codes to save solutions at uniform timesteps.

Although well-motivated, the use of adaptive temporal meshes presents several challenges for neural solvers. Because the step size is determined by the solution, it is not known ahead of time and must instead be computed on-the-fly during autoregressive rollout. Importantly, the $\Delta t$ at inference time must be aligned with those from training to avoid a test-time distribution shift. While the CFL condition is used to determine the step size for classical solvers during data generation, it cannot produce $\Delta t$ matching those in the training data due to the use of coarsened computational meshes and only modeling a subset of the variables comprising the PDE. Specifically, while neural solvers are trained to advance time by hundreds of classical solver steps with a single forward pass, the CFL condition is used to compute the size of a single solver step, and will therefore suggest a step size orders of magnitude smaller than the neural solver encountered during training. Furthermore, because neural solvers learn on coarsened spatial meshes, the cell sizes $\Delta x$ and $\Delta y$ appearing in Equation (2) will not match those used to compute $\Delta t$ in the training data. Finally, Equation (2) is the condition for a single phase flow, whereas a multiphase setting that features, for example, a solid phase interacting with a liquid phase has a much more complicated form involving a large number of field variables. Direct use of this form would require the neural solver to learn to evolve all of these variables, thereby reducing the model's capacity to capture fields of interest accurately.

These challenges motivate us to develop ShockCast, a two-phase framework consisting of a timestep-conditioned neural solver and a neural CFL model which can emulate the timestep sizes in the training data on a coarsened space-time mesh using only a subset of the field variables. At inference time, each unrolling step utilizes each of the phases in turn. In the first phase, the neural CFL model predicts the timestep size which is used by the neural solver in the second phase to evolve the current flow field forward in time by the predicted timestep. We investigate approaches for better aligning the neural CFL model with the CFL condition, and introduce several novel timestep conditioning strategies for the neural solver.

## 3.2 A Two-Phase Framework

Our datasets $\mathcal{D} := \{U_i\}_i^N$ consist of $N$ numerical solutions to the compressible Navier-Stokes equations produced by a classical high-speed flow solver. Each solution $U := \{u_j\}_j^n$ consists of a series of $n$ snapshots on a temporal grid $\mathcal{T} := \{t_j\}_j^n$, where $u_j := u(t_j) \in \mathbb{R}^{D \times M}$ denotes the solution at time point $t_j$ sampled on a spatial discretization with $M$ mesh points and $D$ fields. Notably, $\mathcal{T}$ is coarsened relative to the grid used by the classical solver by selecting every $J$-th solution from the solver for a coarsening factor $J \geq 100$.

For the first phase of our framework, we train a neural CFL model $\psi$ to minimize

$$\mathbb{E}_{j\sim\mathcal{T},\boldsymbol{U}\sim\mathcal{D}}\left[\mathcal{L}_c\left(\psi(\boldsymbol{u}_j),\Delta_j\right)\right]$$

for the loss $\mathcal{L}_c$, where we take $\mathcal{L}_c$ to be the MAE. In the second phase, we train a neural solver $\phi$ to map the solution at the current timestep $\boldsymbol{u}_j$ and the timestep size $\Delta_j$ to the subsequent solution $\boldsymbol{u}_{j+1}$ by optimizing the one-step objective given by

$$\mathbb{E}_{j\sim\mathcal{T},\boldsymbol{U}\sim\mathcal{D}}\left[\mathcal{L}_s\left(\phi(\boldsymbol{u}_j,\Delta_j),\boldsymbol{u}_{j+1}\right)\right],$$

where we take $\mathcal{L}_s$ to be the relative error averaged over fields. While the neural solver is trained to emulate the behavior of the classical solver used to generate the data, the neural CFL model is trained to emulate the process by which the timestep sizes are chosen while generating data. It is important to again emphasize that due to the use of a coarsened computational mesh and a reduced number of states being modeled, it is not possible to directly use the CFL condition to deterministically predict the timestep size. At inference time, ShockCast predicts the full solution given the initial condition $\boldsymbol{u}_0$ by alternating between the two phases autoregressively as

$$\hat{\Delta}t := \psi(\hat{\boldsymbol{u}}(t)) \qquad\qquad \hat{\boldsymbol{u}}(t+\hat{\Delta}t) = \phi\left(\hat{\boldsymbol{u}}(t),\hat{\Delta}t\right),$$

where $\hat{\boldsymbol{u}}(t)$ denotes the predicted state up until time $t$. This process, shown in Figure 1, is repeated until a pre-specified stopping time is reached.

## 3.3 NEURAL CFL

Inspired by the classical CFL condition, we experiment with several modifications to the input features and internal structure of our neural CFL model. As adaptive time-stepping schemes adjust $\Delta t$ according to the sharpness of the gradients of $\boldsymbol{u}(t)$, we include the spatial gradients $\nabla\boldsymbol{u}$ computed using finite differences for all fields in $\boldsymbol{u}$ as inputs. From Equation (2), we furthermore observe the dependence of the CFL condition on the max wave speed, computed by taking the maximum over the local wave speed $\lambda(x,y)$ in all computational cells. This operation can be viewed as a functional mapping $\lambda$ to the scalar value $\lambda_{\max}$ via max pooling. This motivates us to employ max pooling as our spatial downsampling function. Finally, as previously discussed, the classical CFL condition is not directly applicable due to the use of mesh coarsening and modeling only a subset of the field variables. However, it is possible that the functions comprising the condition can be used to learn a surrogate condition. We therefore add *CFL features* to inputs as the local wave speed $\lambda(x,y)$, the velocity magnitudes $|u(x,y)|$ and $|v(x,y)|$, and the local sound speed $a(x,y)$.

## 3.4 TIMESTEP CONDITIONING FOR NEURAL SOLVERS

We now discuss our approaches for timestep conditioning for the neural solver phase.

**Time-Conditioned Layer Norm.** Several prior works have considered training models to advance time by multiples of a uniform step size (Gupta & Brandstetter, 2023; Herde et al., 2024). These models have utilized *time-conditioned layer norm*, a technique originally introduced for conditioning diffusion models on the diffusion time (Nichol & Dhariwal, 2021; Dhariwal & Nichol, 2021). Prior to each layer, the timestep size $\Delta t$ is embedded into two vectors $\boldsymbol{a}$ and $\boldsymbol{b}$ with sizes matching the hidden dimension $d_{\text{model}}$ of the feature map $\boldsymbol{z}$. These are then applied as a scale and shift following each normalization layer as $\text{LN}(\boldsymbol{z})(1+\boldsymbol{a})+\boldsymbol{b}$.

**Spatial-Spectral Conditioning.** Many neural solvers perform convolutions in Fourier space and may not use normalization layers by default (Li et al., 2021; Tran et al., 2023). For these models, the Spatial-Spectral conditioning strategy introduced by Gupta & Brandstetter (2023) can be used to perform timestep conditioning in the frequency domain. Under this scheme, the Fourier transform of the feature map is point-wise multiplied with a complex-valued embedding $\boldsymbol{\xi}$ of $\Delta t$ as $\mathcal{F}(\boldsymbol{z})\boldsymbol{\xi}$. $\boldsymbol{\xi}$ has different entries for each frequency of $\mathcal{F}(\boldsymbol{z})$, and so to maintain parameter-efficiency, Gupta & Brandstetter (2023) share $\boldsymbol{\xi}$ across all channels of $\boldsymbol{z}$.

**Euler Residuals.** Neural solvers often employ residual connections (He et al., 2016) as $\boldsymbol{z}_{l+1} = \boldsymbol{z}_l + F_l(\boldsymbol{z}_l)$, where the $l$-th solver layer $F_l$ includes spatial integration operations such as convolution

or attention. Many works have studied the relationship between residual connections and Euler integration (Lu et al., 2018; Haber & Ruthotto, 2017; Ruthotto & Haber, 2020), and even extended it to more general classes of integrators (Chen et al., 2018; Kidger, 2022). For a function of time $\boldsymbol{v}$, Euler integrators approximate time integration as

$$\boldsymbol{v}(t + \Delta t) = \boldsymbol{v}(t) + \int_t^{t+\Delta t} \partial_t \boldsymbol{v}(\tau) \mathrm{d}\tau \approx \boldsymbol{v}(t) + \Delta t \partial_t \boldsymbol{v}(t).$$

When viewing the evolution of the latent features $\boldsymbol{z}_l \mapsto \boldsymbol{z}_{l+1}$ from layer-to-layer as the evolution of some latent map $\boldsymbol{z}(t) \in \mathbb{R}^{d_{\mathrm{model}} \times M}$ in time, the time integration of $\boldsymbol{z}$ carried out numerically by the Euler integrator corresponds exactly to residual connections. In our scenario, we interpret the $l$-th layer representation $\boldsymbol{z}_l$ corresponding to the input field $\boldsymbol{u}(t)$ and timestep size $\Delta t$ as the latent form of some intermediate state $\boldsymbol{u}(t + \alpha \Delta t)$, where $\alpha \in [0, 1]$ increases monotonically with depth in the network. This interpretation leads to *Euler Residuals*, in which the period of time integration executed by $F_l$ is related to the timestep size $\Delta t$ as $\boldsymbol{z}_{l+1} = \boldsymbol{z}_l + \boldsymbol{a} F_l(\boldsymbol{z}_l)$, where $\boldsymbol{a}$ is an affine transformation of $\Delta t$ as $\boldsymbol{a} = \boldsymbol{W} \Delta t + \boldsymbol{c}$ for $\boldsymbol{W}, \boldsymbol{c} \in \mathbb{R}^{d_{\mathrm{model}}}$.

**Mixture of Experts.** Mixture of Experts (Shazeer et al., 2017; Jacobs et al., 1991; Fedus et al., 2022) has emerged as an effective approach to scaling up models and has been applied in PDE modeling tasks (Hao et al., 2023). The Mixture of Experts (MoE) layer consists of a gating network $G_l$ and $K$ experts $F_{l,1}, \ldots, F_{l,k}$. The gating network is often a simple MLP, while the experts can have more complex architectures. Based on layer inputs $\boldsymbol{z}_l$, the gating network weighs the outputs of the experts according to the gate $G_l(\boldsymbol{z}_l) \in \mathbb{R}^K$ as $\boldsymbol{z}_{l+1} = \boldsymbol{z}_l + \sum_k^K G_l(\boldsymbol{z}_l)_k F_{l,k}(\boldsymbol{z}_l)$. The gating network can then learn to partition the latent space such that each expert specializes in a particular area. As our task involves learning to evolve dynamics by variable time lengths, we gate experts according to the timestep $\Delta t$ as

$$\boldsymbol{z}_{l+1} = \boldsymbol{z}_l + \sum_k^K G_l(\Delta t)_k \left(\boldsymbol{a}_k F_{l,k}(\boldsymbol{z}_l)\right),$$

where $\boldsymbol{a}_k$ is an affine transformation of $\Delta t$ for the $k$-th expert, and plays the same role as for the Euler residuals. This enables each layer to have experts specializing in short time integration periods where $\boldsymbol{u}(t)$ contains sharp gradients, as well as experts for handling longer timesteps where the dynamics behave more smoothly. Because our router weights are dense, we do not leverage the conditional-computation efficiency of sparse MoE architectures (Shazeer et al., 2017; Fedus et al., 2022). As a result, the MoE conditioning mechanism trades increased model capacity for greater computational cost.

## 4 RELATED WORK

Learnable spatial re-meshing for PDEs has been an active area of research, with advances made in supervised Pfaff et al. (2021); Song et al. (2022); Zhang et al. (2024a) and reinforcement learning Wu et al. (2022a); Freymuth et al. (2023); Yang et al. (2023) frameworks. However, to the best of our knowledge, we are the first to consider learning to temporally re-mesh by utilizing data with adaptable temporal resolution, which is vital for developing models for high-speed flows. Works in this direction have instead developed various schemes using temporally uniform data. Wu et al. (2025) recently introduced a pipeline wherein a timestep prediction model is trained using an unsupervised loss designed to avoid timesteps that are too small. A second module then predicts temporal derivatives of various orders such that the solution can be queried using a Taylor expansion at any point up until the predicted timestep, enabling supervision using the temporally-uniform ground truth data. Following a similar continuous-time strategy, Janny et al. (2024) propose to learn an interpolator such that arbitrary time points between temporally uniform training data can be queried. Hagnberger et al. (2024) employ a conditional neural field to map from initial conditions to arbitrary query times. Other works learn to map forward in time by various multiples of a uniform step size, with Liu et al. (2022a); Hamid et al. (2024) training different models for each step size and Gupta & Brandstetter (2023); Herde et al. (2024) training one model to be shared across step sizes using time-conditioned layer norm. Importantly, Gupta & Brandstetter (2023); Herde et al. (2024) consider the timestep size to be known *a priori*, with Gupta & Brandstetter (2023) treating this as a benchmark for probing the ability of neural solvers to respond to timestep conditioning and Herde et al. (2024)

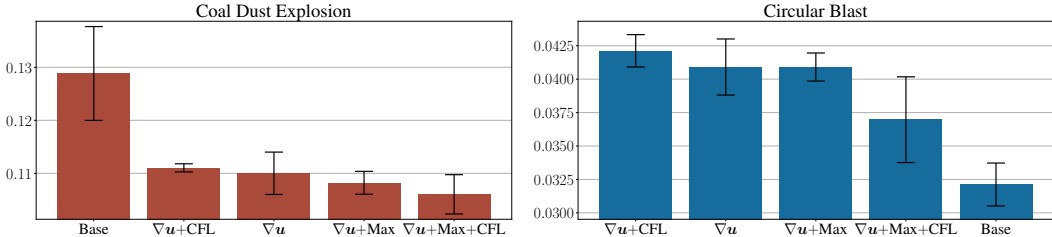

Figure 2: One-step MAE of Neural CFL models on $\Delta t$ averaged over 3 training runs, where $\Delta t$ is normalized to have standard deviation 1. Error bars are $\pm 2$ standard errors.

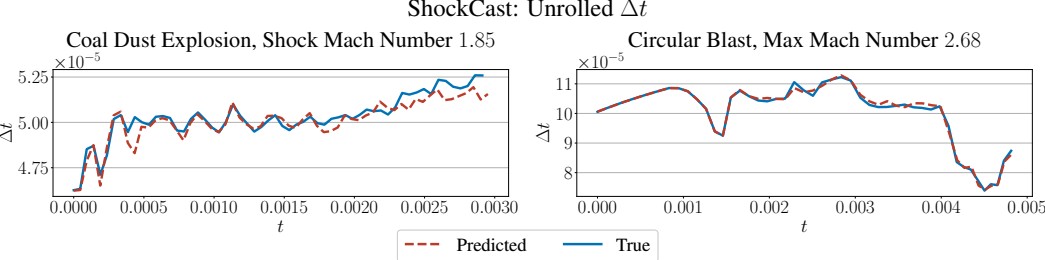

Figure 3: $\Delta t$ predicted by autoregressive unrolling of ShockCast with F-FNO+Euler conditioning neural solver backbone for a selected solution.

using it as a pre-training task. As detailed in Section 3.1, this assumption is not realistic in the setting we consider. Adaptive time-stepping extends beyond PDE modeling applications in ML research, with connections to adaptive learning rates (Duchi et al., 2011; Tieleman, 2012; Kingma & Ba, 2015) and continuous-time generative models (Song et al., 2021; Lipman et al., 2023).

## 5 EXPERIMENTS

### 5.1 DATASETS

We consider three settings of high-speed flows in the supersonic regime with two spatial dimensions, one of which we extend to three spatial dimensions. We discuss the generation of these cases in Appendix A and visualize solutions in Appendix J.

**Coal Dust Explosion.** The first setting we consider is a multiphase problem containing both gaseous air and granular coal particles. The simulation begins with a thin, uniform layer of coal dust settled on the bottom of a channel. Near the left boundary, we initialize a normal shock. We vary the initial strength of the shock between Mach 1.2 and 2.1 along with the particle diameter from case to case for a total of 100 cases, with 90 for training and 10 for evaluation. Once the simulation starts, the normal shock travels to the right as shown in Figure 10, where it interacts with the dust layer, first compressing it and later generating instabilities at the gas-dust layer interface. These instabilities further grow with time into turbulent vortical structures, which raise the dust in the channel and mix them with the air. The amount of mixing depends both on the initial shock strength and the particle diameter. We train our models to predict the velocity and temperature of the gas and the volume fraction describing the percentage of coal dust at each point.

**Circular Blast.** The second setting we consider is a two-dimensional circular blast case, which represents a two-dimensional version of the Sod's shock tube problem (Sod, 1978). We initialize a circular region of high pressure such that the pressure inside the circle is substantially higher than its surroundings as shown in Figure 11. We vary the ratio of these initial pressures from 1.99 to 50 to produce a set of 99 cases split into 90 training cases and 9 evaluation cases. Once the simulation starts, a circular shock travels radially outward, while an expansion wave travels in the opposite direction. This continues until the outward moving shock reflects from the boundaries

Circular Blast: Density

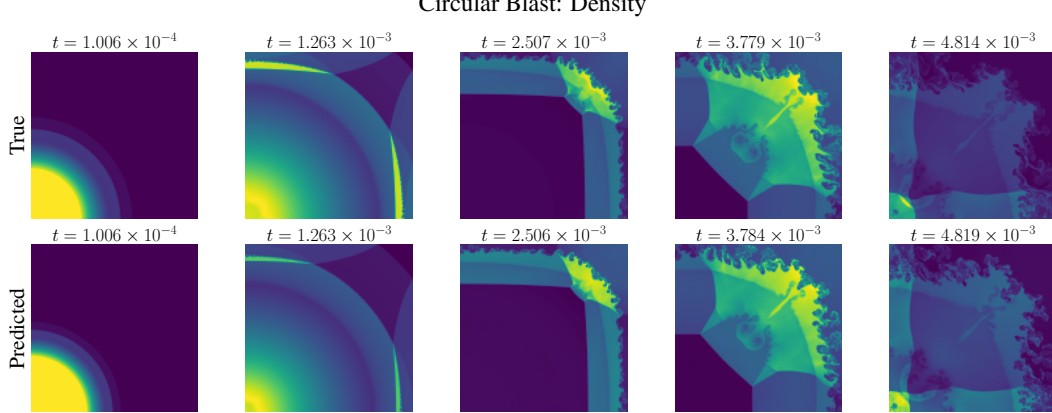

Figure 4: Comparison of the ground truth (top) and predicted (bottom) density fields for the circular blast data. We obtain predictions with autoregressive unrolling of ShockCast.

ShockCast Results: Coal Dust Explosion

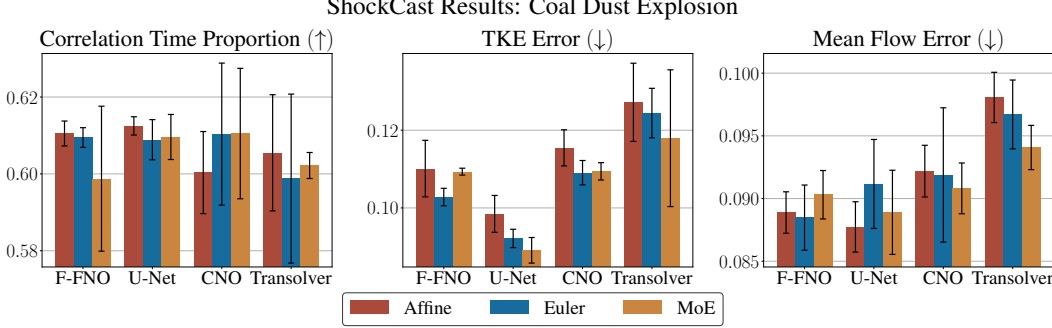

Figure 5: Coal dust explosion results averaged over three neural solver training runs. Error bars are $\pm 2$ standard errors.

and travels inwards toward the origin. The interaction of the reflected shocks with the post-shock gas generates instabilities which grow into turbulent structures. Once these reflected shocks reach the origin, they reflect again, thus propagating radially outward. This continues repeatedly, while with each reflection, the shocks lose strength. The maximum Mach number, which correlates with the initial pressure ratio, varied from 0.49 to 2.97 across cases. We train our models to predict the velocity, temperature, and density fields. To assess the scalability of ShockCast to three spatial dimensions, we additionally generate and evaluate on an extension of this setting to a **Spherical Blast**. We additionally assess the stability of ShockCast across many time integrations by generating an alternative extension of this setting to **Long Circular Blast** by more than doubling the length of the simulation, resulting in trajectories with more than 100 unrolling steps.

**Airfoil Shock.** The third setting we consider is a shock–airfoil interaction problem involving a NACA 0012 airfoil immersed in a compressible flow. We initialize a normal shock inside the domain and vary its initial strength between Mach numbers 1.2 and 2.1. In addition, we vary the angle of attack of the airfoil between $\pm 8°$, yielding a total of 100 cases with 90 used for training and 10 for evaluation. Once the simulation begins, the shock travels from left to right and impinges on the airfoil, generating strong compression on the windward side followed by shock reflections and downstream vortical structures. The complexity of the flow increases with both the shock strength and the angle of attack. The irregularity of the airfoil geometry allows us to evaluate ShockCast beyond regular domains with uniform spatial discretizations.

## 5.2 SHOCKCAST BACKBONES

In the setting with regular domain and with uniform spacing, we use the ConvNeXt architecture (Liu et al., 2022b) as the backbone architecture for our Neural CFL model, while in the Airfoil Shock settings, we use a GNN with spatial downsampling between each message passing layer. Both are

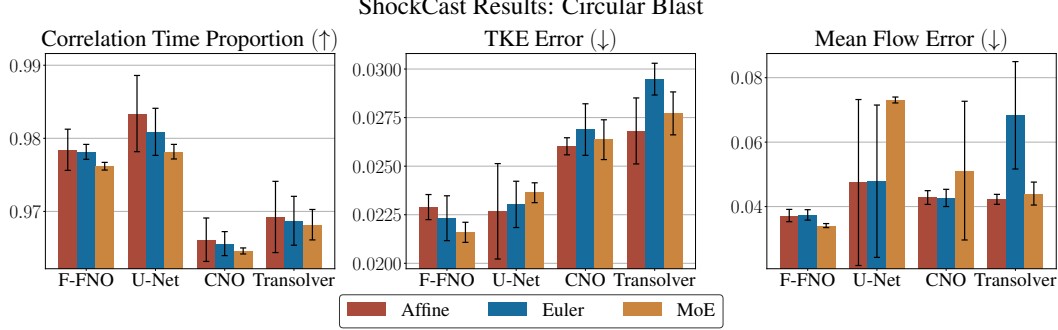

Figure 6: Circular blast results averaged over three neural solver training runs. Error bars are $\pm$ 2 standard errors.

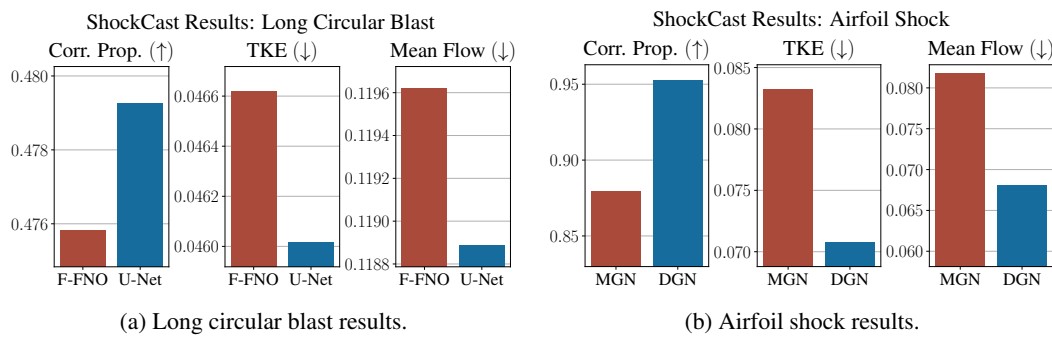

(a) Long circular blast results.    (b) Airfoil shock results.

trained with the noise injection strategy from Sanchez-Gonzalez et al. (2020). We experiment with a variety of neural solver architectures. For the Coal Dust Explosion and Circular Blast settings, we experiment with the U-Net from Gupta & Brandstetter (2023), Convolutional Neural Operator (CNO) (Raonic et al., 2024), Factorized FNO (Tran et al., 2023) (F-FNO), and Transolver (Wu et al., 2024). We refer to the Affine version of models as the one using either spatial-spectral conditioning in the case of F-FNO or time-conditioned layer norm in the case of the remaining architectures. The remaining model variants add onto the Affine versions by additionally employing Euler conditioning, as well as MoE conditioning, both of which use Euler residuals. Finally, for the Airfoil Shock settings, we experiment with MeshGraphNets (MGN) (Pfaff et al., 2021) and Diffusion Graph Nets (DGN) from Valencia et al. (2025), which is based on the Graph U-Net (Lino et al., 2022; Gao & Ji, 2019). Both GNN architectures use affine timestep conditioning. We discuss these architectures and training procedures in greater detail in Appendices B and C, respectively.

## 5.3 METRICS

To evaluate predicted solutions, we compute the Pearson's correlation coefficient for each field and at each timestep with the ground truth data. This requires the predicted fields to be on the same temporal grid as the ground truth data, for which we use linear interpolation, described in Appendix D. The correlation time (Kochkov et al., 2021; Lippe et al., 2023; Alkin et al., 2024) for a given field is defined as the last time $t$ before the correlation sinks below a threshold, which we take to be 0.9 here. We then average this time across all fields and report it as a percentage of the full simulation

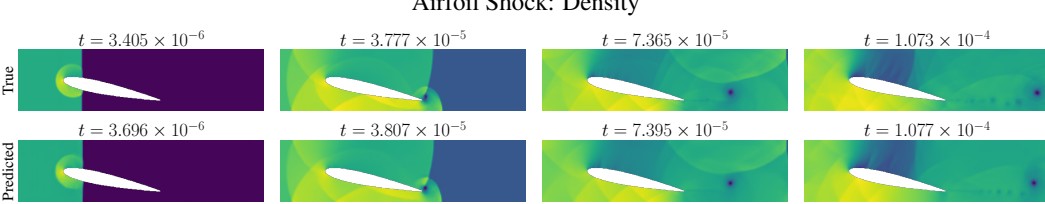

Figure 8: Comparison of the ground truth (top) and predicted (bottom) density fields for the Airfoil Shock data. We obtain predictions with autoregressive unrolling of ShockCast.

time. We additionally use the predicted fields to compute several of the primary physical quantities of interest to practitioners. The mean flow is computed by averaging each flow field over time, while the Turbulence Kinetic Energy (TKE) is calculated as the sum of the variances of the fluctuating part of the velocity field. These quantities are given by

$$\bar{\boldsymbol{u}} := \frac{1}{T} \int_0^T \boldsymbol{u}(t)\mathrm{d}t \qquad \mathrm{TKE} := \frac{1}{2T} \int_0^T (u(t) - \bar{u})^2 + (v(t) - \bar{v})^2 \mathrm{d}t, \qquad (4)$$

respectively. We report the relative error averaged over each variable for the mean flow, and relative error of the TKE field, where the integrals are approximated using the trapezoidal rule.

### 5.4 RESULTS

We examine the one-step MAE of several variants of the neural CFL model in Figure 2. For the circular blast, a single-phase problem where the CFL condition is determined entirely by the velocity and temperature fields, the base model yields the best performance, as the modeled variables by themselves are sufficient to accurately predict the timestep size. In contrast, the coal dust explosion has a more complicated form of the CFL condition to account for both the solid phase describing the coal dust and the gas phase. The modeled variables primarily focus on the gas phase, with the exception of the volume fraction. In this more challenging setting, we observe substantial benefits from our physically-motivated enhancements to the neural CFL model. Our best results are achieved when using max-pooling, with the spatial gradient of the flow state $\nabla \boldsymbol{u}$ and CFL features added to inputs. In Figure 3, we examine the predicted $\Delta t$ obtained through autoregressive unrolling of the ShockCast framework in each setting and observe a close match with the ground truth. We use the best neural CFL model in each of the settings for the full ShockCast models discussed next.

We visualize unrolled predictions on the circular blast density field in Figure 4. In Figures 5 and 6, we examine the performance of ShockCast with each of the neural solver architectures and timestep conditioning strategies. For both the coal dust explosion and circular blast settings, ShockCast achieves the strongest performance in terms of correlation time when leveraging a U-Net backbone with time-conditioned layer norm. On the coal dust explosion cases, MoE conditioning and Euler conditioning with a U-Net backbone achieve the first and second best performance in terms of TKE error, respectively. Similarly, the TKE error for the circular blast is lowest for Euler and MoE conditioning strategies with a F-FNO backbone. Finally, while U-Net with time-conditioned layer norm has the best mean flow error for the coal dust explosion setting, the F-FNO with MoE variant of ShockCast has the lowest circular blast mean flow error.

In Figure 9, we visualize unrolled predictions for a spherical blast case using a 3D U-Net neural solver backbone with time-conditioned layer norm and 3D ConvNeXt neural CFL backbone. In Figure 7a, we present results for the long circular blast setting for U-Net and F-FNO neural solver backbones with time-conditioned layer norm. The solutions remain correlated about 0.9 for approximately half of the simulation time, and the low TKE and mean flow errors demonstrate that the predicted solutions remain stable and in-distribution, which is further supported by the rollout visualizations shown in Appendix G. Finally, we compare ShockCast with MGN and DGN neural solver backbones in Figure 7b, where we find DGN to be more performant across metrics. We visualize ShockCast predictions for a test airfoil case in Figure 8. We present extended results in Appendix K.

## 6 CONCLUSION

In this work, we develop machine learning methods for modeling high-speed flows with adaptive time-stepping. To this end, we propose ShockCast, a two-phase framework that learns timestep sizes in the first phase and evolves fluid fields by the predicted step size in the second phase. To evaluate ShockCast, we generate three new supersonic flow datasets. Results show that ShockCast is effective at learning to temporally re-mesh and evolve fluid fields. Our work takes steps towards developing machine learning models for high-speed flows, where there is great potential for neural acceleration due to the immense computational requirements of classical methods.

ACKNOWLEDGMENTS

This work was supported in part by the National Science Foundation under grant IIS-2243850. The authors express their gratitude to Professor Ryan Houim of the University of Florida for providing access to HyBurn, the computational fluid dynamics (CFD) code utilized for the simulations presented in this study. The authors are grateful to Fernando Guzman for his assistance generating CFD data. The CFD calculations presented in this work were partly performed on the Texas A&M high-performance computing cluster Grace.

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

# Appendix

## A  CFD PROBLEM DESCRIPTION AND NUMERICAL MODELS

We conduct the numerical simulations in this work using the HyBurn code, which implements high-order Godunov schemes to solve the governing equations. HyBurn is a finite volume solver that employs parallelization and adaptive mesh refinement (AMR) via the AMReX library (Zhang et al., 2019). It uses a low-dissipation, WENO-based high-order Godunov method (Houim & Kuo, 2011; Houim & Oran, 2016; Balsara & Shu, 2000; Martín et al., 2006; Shen et al., 2016; Harten et al., 1983) and models turbulence using Implicit Large Eddy Simulation (Grinstein et al., 2007; Oran et al., 2001; Thornber et al., 2008; Drikakis et al., 2009; Thornber et al., 2007). HyBurn solves compressible Navier-Stokes equations for Eulerian-Eulerian granular multiphase reactive flows, employing advanced numerical techniques developed by Houim & Oran (2016). It uses an explicit third-order three-stage Runge-Kutta time-stepping algorithm along with two independent CFL numbers – one for the hyperbolic and the other for the parabolic terms of the compressible Navier-Stokes equations. The actual timestep size is determined by whichever is the smaller of the two. HyBurn has been extensively validated and verified against various compressible multiphase and reactive flow problems (Houim & Kuo, 2011; Houim & Oran, 2016). More details about the algorithms and

Spherical Blast: Density

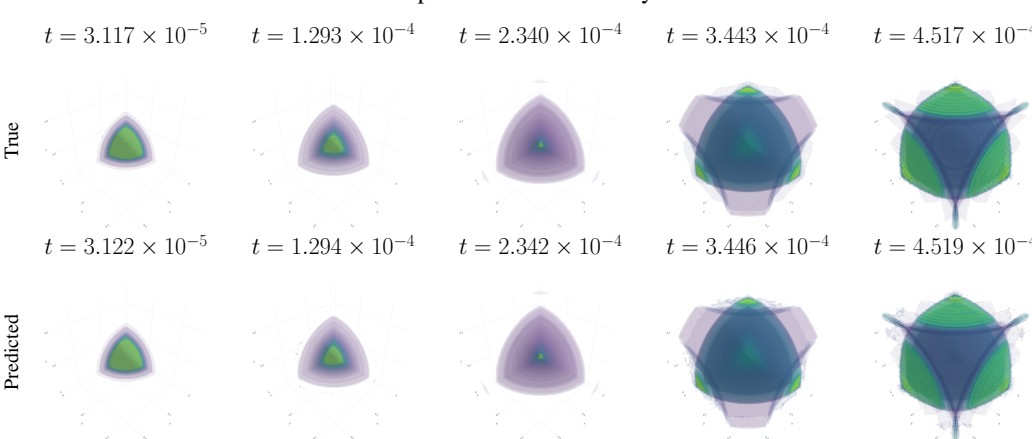

Figure 9: Comparison of the ground truth (top) and predicted (bottom) density fields for the spherical blast data. We obtain predictions with autoregressive unrolling of ShockCast.

| Coal Dust Explosion: Grid Independence | |
| --- | --- |
| Refinement Levels | Shock speed $(\mathrm{m\,s^{-1}})$ |
| 1 | 1,102.52 |
| 2 | 1,033.99 |
| 3 | 999.70 |

| Circular Blast: Grid Independence | |
| --- | --- |
| Refinement Levels | Time $(\mathrm{s} \times 10^{-3})$ |
| 1 | 1.0178 |
| 2 | 1.0179 |
| 3 | 1.0196 |

(a) Coal dust explosion grid independence tests. We choose an arbitrary point in the domain and measure the shock speed at this point for each of the configurations.

(b) Circular blast grid independence tests. We choose an arbitrary point in the domain and measure the time taken for the shock to reach this point for each of the three configurations.

Table 1: Grid independence tests. In each setting, we re-ran a selected case for varying levels of mesh refinement.

the applications of HyBurn can be found in previous work (Guhathakurta & Houim, 2023a; 2021; Guhathakurta, 2021; Guhathakurta & Houim, 2023b; Posey et al., 2021; Li & Houim, 2024; Li, 2022; Hargis et al., 2024; Egeln et al., 2023; Farrukh et al., 2025).

### A.1 COAL DUST EXPLOSION

The first setting we consider is loosely based on the coal dust explosion simulations in Guhathakurta & Houim (2023a; 2021); Guhathakurta (2021); Guhathakurta & Houim (2023b). The major differences are that in this study we use a normal shock instead of a detonation to mimic the primary explosion in a coal mine, and there are no chemical reactions involved. The computational geometry we use is a $25\,\mathrm{cm}$ by $5\,\mathrm{cm}$ two-dimensional rectangular channel containing air, with a thin uniform layer of coal dust settled on the bottom of the channel. We keep the left and right boundaries open, while the top and bottom boundaries are symmetry. This is a multiphase problem containing both gaseous air and granular coal particles. Near the left boundary, we initialize a normal shock using post-shock conditions to the left of the shock and quiescent air ($300\,\mathrm{K}$ temperature, $1\,\mathrm{atm}$ pressure) ahead of it. We vary the initial strength of the shock between Mach 1.2 and 2.1 along with the particle diameter between $1\,\mathrm{\mu m}$ and $150\,\mathrm{\mu m}$ from case to case for a total of 100 cases. We set the initial volume fraction of the dust layer to $47\%$ and the particles to be monodisperse. We model the coal particles as being composed of inert ash only, with a solid-phase density of $1330\,\mathrm{kg\,m^{-3}}$. We use two AMR levels to give an effective resolution of $\sim0.12\,\mathrm{mm}$ at the finest level. We use a fifth-order WENO interpolation scheme and the HLL (Toro, 2013) flux reconstruction method. We set the CFL numbers for both the hyperbolic and parabolic terms to 0.6.

Once the simulation starts, the normal shock travels to the right as shown in Figure 10, where it interacts with the dust layer, first compressing it and later generating instabilities at the gas-dust layer interface. These instabilities further grow with time into turbulent vortical structures, which

Coal Dust Explosion: Gas Velocity $x$-component, Shock Mach Number $1.85$

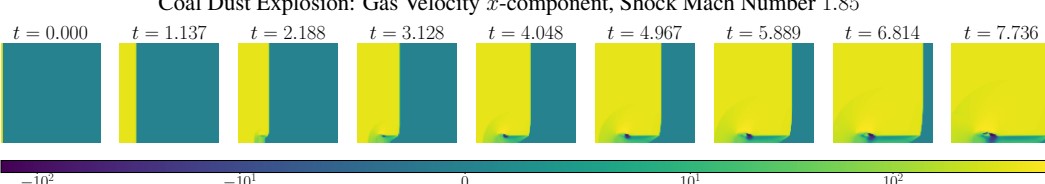

Figure 10: Initial gas velocity $x$-component for a selected coal dust explosion case. Times are in units of $10^{-5}$ seconds and the downsampling factor relative to the classical solver solution is $100\times$ compared to $500\times$ used for training ShockCast. The initial shock can be seen to be moving from left to right.

Circular Blast: Initial Densities

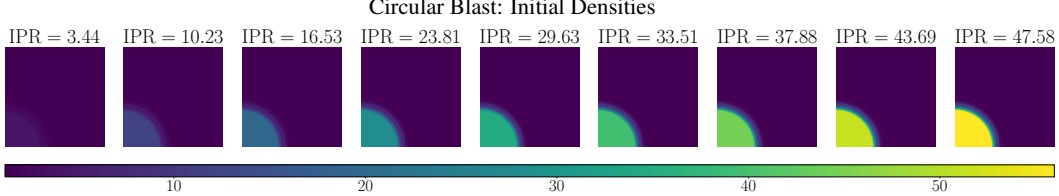

Figure 11: Initial density field for the circular blast evaluation cases with varying Initial Pressure Ratios (IPR). Due to the ideal gas law, increasing the pressure also increases the density.

raise the dust in the channel and mix them with the air. The amount of mixing depends both on the initial shock strength and the particle diameter. We cap the total simulation time at $3\,\text{ms}$ for all cases.

We used a case with initial strength of the shock Mach 3 and particle diameter $30\,\mu\text{m}$ to conduct a grid independence analysis. We used a base grid of $520 \times 104$ and conducted the analysis by varying the AMReX maximum levels of refinement between 1 and 3. In Table 1a, we compare the shock speed at an arbitrary point in the domain for each of the three configurations. Because the shock speed differed least between 2 and 3 levels of refinement (3.4%), we used 2 levels of refinement as the grid configuration for data generation.

## A.2 CIRCULAR BLAST

The second setting we consider is a two-dimensional circular blast case, which represents a two-dimensional version of the Sod's shock tube problem (Sod, 1978). We initialize a circular region of high pressure such that the pressure inside the circle is substantially higher than its surroundings as shown in Figure 11. We vary the ratio of these initial pressures from 1.99 to 50 to produce a set of 99 cases. To reduce the computational cost, we only simulate a quarter of the circle. We use symmetry boundary conditions for all boundaries to allow for the generated shocks and expansion waves to reflect when incident on them. We model a single gaseous phase throughout the computational domain. We set the initial temperatures to $300\,\text{K}$ everywhere, with the gas initially at rest. We use two Adaptive Mesh Refinement (AMR) levels – triggered by predefined pressure and density ratio thresholds between any two adjacent computational cells – to give an effective resolution of $\sim 0.98\,\text{mm}$ at the finest level, which is sufficient to resolve the shocks. We use a fifth-order MUSCL (Van Leer, 1979) interpolation scheme and the HLLC-LM (Fleischmann et al., 2020) flux reconstruction method. We set the CFL numbers for both the hyperbolic and parabolic terms to $0.8$.

Once the simulation starts, a circular shock travels radially outward, while an expansion wave travels in the opposite direction. This continues until the outward moving shock reflects from the boundaries and travels inwards toward the origin. The interaction of the reflected shocks with the post-shock gas generates instabilities which grow into turbulent structures. Once these reflected shocks reach the origin, they reflect again, thus propagating radially outward. This continues repeatedly, while with each reflection, the shocks lose strength. We cap the total simulation time at $5\,\text{ms}$ for all cases.

We used a case with blast pressure of $50\,\text{atm}$ to conduct the grid independence test. We used a base grid of $256 \times 256$ and performed the grid independence test by varying the AMREX maximum levels of refinement between 1 and 3. In Table 1b, we compare the time taken by the shock to reach an arbitrary point in the domain for each of the three grid configurations. As the difference in the

shock speeds was minor across the three configurations, we chose the grid with 2 levels of maximum refinement for generating the cases.

### A.3 SPHERICAL BLAST

The spherical blast setting extends the circular blast configuration to three spatial dimensions. As in the 2D case, we initialize a region of high pressure embedded in quiescent ambient air. The ambient state is set to $300\,\text{K}$ and $1\,\text{atm}$, with the gas initially at rest throughout the domain. Inside the spherical region, we increase the pressure to obtain initial pressure ratios ranging from 2 to 34 across cases while keeping the temperature fixed at $300\,\text{K}$, leading to correspondingly higher initial densities through the ideal gas law. We model a single gaseous phase governed by a constant-$\gamma$ ideal-gas equation of state with the same thermodynamic parameters as in the circular blast setting.

To reduce computational cost, we exploit symmetry and simulate only one octant of the spherical blast. The computational domain is a cubic box of side length $25\,\text{cm}$, and we place the blast center at the corner of the cube so that the coordinate planes coincide with symmetry boundaries. We employ symmetry boundary conditions on all faces, allowing the outward-propagating spherical shock and inward-moving expansion waves to reflect in a physically consistent manner. We use a base grid of $64 \times 64 \times 64$ cells with two Adaptive Mesh Refinement (AMR) levels, triggered by pressure and density gradients as in the circular blast case. This yields an effective finest-level cell size of approximately $0.98\,\text{mm}$, which is sufficient to resolve the shock structure. We use the MUSCL (Van Leer, 1979) reconstruction scheme with the HLLC-LM flux method (Fleischmann et al., 2020), and set the CFL numbers for both hyperbolic and parabolic terms to $0.8$. We run each simulation until $2\,\text{ms}$, capturing the initial blast, the formation of the outgoing spherical shock and inward rarefaction, and their early reflections from the symmetry planes across the full range of pressure ratios.

### A.4 AIRFOIL SHOCK

The third setting we consider is a shock–airfoil interaction problem involving a NACA 0012 airfoil immersed in a compressible flow being exposed to a moving normal shock of varying strengths. This case evaluates ShockCast's ability to model high-speed flows over curved solid boundaries using the immersed boundary method (IBM) implemented in HyBurn. The computational domain is a two-dimensional Cartesian rectangle of size $5\,\text{cm} \times 1.25\,\text{cm}$, and the airfoil geometry is imported from an ASCII STL file and scaled by a factor of $0.02$, yielding an effective chord length of approximately $2\,\text{cm}$. The geometry is then translated so that the lower-left corner of its axis-aligned bounding box lies at $(x, y) = (1\,\text{cm}, 0.5\,\text{cm})$. The airfoil surface is treated as an adiabatic no-slip wall, with IBM guard cells and halo layers used to enforce the boundary conditions and maintain surface refinement during regridding.

We initialize a normal shock inside the domain at $x = 5\,\text{mm}$, with post-shock conditions to the left of the shock and quiescent ambient air ($300\,\text{K}$, $1\,\text{atm}$) ahead of it. Across the dataset, we vary the initial shock strength between Mach numbers $1.2$ and $2.1$, and we vary the angle of attack of the airfoil between $-8°$ and $+8°$ by rotating the underlying STL geometry. This yields a total of 100 cases, of which 90 are used for training and 10 for evaluation.

We employ Adaptive Mesh Refinement (AMR) with a base grid of $64 \times 16$ cells and up to four levels of refinement, with a refinement ratio of $2\times$ between successive levels. This gives an effective finest resolution of approximately $50\,\mu\text{m}$ to resolve the shock adequately. Refinement is triggered using shock sensors based on pressure rise and IBM surface proximity, ensuring that both the shock front and the airfoil geometry remain well-resolved throughout the simulation. We use a fifth-order WENO reconstruction scheme with Rusanov flux and a low-Mach correction to the Riemann solver.

Once the simulation begins, the normal shock propagates from left to right and impinges on the airfoil leading edge. The interaction generates strong compression on the windward side of the airfoil, followed by shock reflections and possible separation depending on the angle of attack. The flow downstream of the airfoil develops complex vortical structures whose strength increases with both shock Mach number and angle of attack. We set the CFL numbers for the hyperbolic and parabolic terms to $0.9$ and integrate the compressible Navier–Stokes equations using a three-stage, third-order SSP Runge–Kutta method. All cases are simulated until a final time of $1.2 \times 10^{-4}\,\text{s}$,

| Model | Initial Learning Rate | Latent Dimension | Key Hyperparameters |
|---|---|---|---|
| U-Net | $2 \times 10^{-4}$ | 64 | 3 downsampling/upsampling levels |
| CNO | $2 \times 10^{-4}$ | 27 | 4 downsampling/upsampling levels, 6 residual blocks per level |
| F-FNO | $1 \times 10^{-3}$ | 96 | 32 modes, 12 layers |
| Transolver | $6 \times 10^{-4}$ | 192 | 8 layers, 8 attention heads, 8 slices |
| ConvNeXT | $2 \times 10^{-4}$ | 96 | ConvNeXt-T (See Section 3 of Liu et al. (2022b)) |
| MGN | $2 \times 10^{-4}$ | 128 | 16 message passing layers |
| DGN | $2 \times 10^{-4}$ | 96 | 4 downsampling/upsampling levels, 2 message passing blocks per level |
| DGN-CFL | $2 \times 10^{-5}$ | 128 | 4 downsampling levels, 2 message passing blocks per level |
| U-Net-3D | $2 \times 10^{-4}$ | 32 | 3 downsampling/upsampling levels |
| F-FNO-3D | $1 \times 10^{-3}$ | 96 | 16 modes, 12 layers |
| ConvNeXT-3D | $2 \times 10^{-4}$ | 96 | ConvNeXt-T (See Section 3 of Liu et al. (2022b)) |

Table 2: Hyperparameters for neural solver and neural CFL models.

which is sufficient for the shock to traverse the chord length and for the primary reflection patterns to form.

# B  NEURAL SOLVERS  AND NEURAL CFL MODELS

Various neural models for solving PDEs have emerged over the last decade, including Physics-Informed Neural Networks (Raissi et al., 2019; Biswas & Anand, 2023; 2024; Cho et al., 2024; Shah & Anand, 2024) and operator learning (Lu et al., 2021; Gupta et al., 2021; Li et al., 2022b;a; 2023b; Poli et al., 2022; Seidman et al., 2022; Kovachki et al., 2023). Neural solvers have been tailored to a variety of applications of PDE modeling, including subsurface modeling (Deng et al., 2022; Wu et al., 2022b), climate and weather modeling (Bi et al., 2022; Lam et al., 2022; Pathak et al., 2022; Price et al., 2023; Nguyen et al., 2023), airfoil design (Bonnet et al., 2022; Helwig et al., 2024), and density functional theory (Zhang et al., 2026). Neural solvers span a diverse array of architectures, including convolutional models (Li et al., 2021; Tran et al., 2023; Wen et al., 2022; Helwig et al., 2023; Wen et al., 2023; Bonev et al., 2023; Zhang et al., 2024b; Raonic et al., 2024), transformers (Cao, 2021; Li et al., 2023a; Janny et al., 2023; Hao et al., 2024; Alkin et al., 2024), and graph neural networks (Li et al., 2020a;b; Brandstetter et al., 2022; Horie & Mitsume, 2022).

Previous works have highlighted the importance of multiscale processing mechanisms in designing effective neural solvers Gupta & Brandstetter (2023). This is at least in part due to the temporal coarsening approach taken by neural solvers. While differential operators defining PDEs are primarily local in nature, their effects become increasingly global as the timestep size is increased beyond that of the classical solver. It is therefore vital for neural solvers to incorporate spatial processing mechanisms that enable modeling of phenomena on both local and global scales. Thus, we explore a variety of neural solver backbones in ShockCast spanning both hierarchical and parallel multi-scale processing mechanisms. Within the parallel framework, we consider both convolution-based and attention-based mechanisms.

**U-Net.** The U-Net Ronneberger et al. (2015) is one of the most prominent examples of a hierarchical mechanism. The U-Net is composed of a downsampling path and an upsampling path. To achieve a global receptive field, the input features sampled on the original solution mesh are first sequentially downsampled using pooling operations or strided convolutions. At each resolution, the downsampled features are convolved and point-wise activations are applied. Following the downsampling path, an inverse upsampling path is applied, consisting of upsampling operations which are either transposed convolutions or interpolation, with convolution and non-linearities again ap-

| Model | Parameters (M) | GigaFLOPs | Peak Train Memory (GiB) |
|---|---|---|---|
| U-Net | | | |
|     Affine | 140.861 | 94.328 | 13.465 |
|     Euler | 140.861 | 94.328 | 14.152 |
|     MoE | 131.129 | 85.839 | 21.307 |
| CNO | | | |
|     Affine | 45.710 | 16.028 | 8.951 |
|     Euler | 45.704 | 16.028 | 9.629 |
|     MoE | 174.339 | 55.917 | 27.783 |
| F-FNO | | | |
|     Affine | 15.372 | 20.901 | 18.953 |
|     Euler | 15.374 | 20.901 | 20.164 |
|     MoE | 15.697 | 20.762 | 37.242 |
| Transolver | | | |
|     Affine | 11.139 | 191.112 | 41.895 |
|     Euler | 10.336 | 191.110 | 41.881 |
|     MoE | 9.911 | 182.924 | 62.432 |
| ConvNeXT | 27.822 | 1.736 | 11.117 |
| MGN | 4.719 | 382.129 | 39.383 |
| DGN | 4.443 | 124.484 | 17.445 |
| DGN-CFL | 0.795 | 15.577 | 4.305 |

Table 3: Model parameter counts, GigaFLOPs per forward pass, and peak GPU memory usage during training. For U-Net, CNO, F-FNO, Transolver, and ConvNeXT, FLOPs were computed with a batch size of 1 on the coal dust explosion dataset, while GPU memory was computed for a batch size of 32 on a single A100 GPU. For MGN, DGN, and DGN-CFL, FLOPs were computed with a batch size of 1 on the airfoil shock dataset, while GPU memory was computed for a batch size of 4 on a single RTX 6000 Ada GPU.

plied at each resolution. To restore high-frequency details that were lost along the downsampling path, skip connections from respective resolutions along the downsampling path concatenate feature maps at each stage of the upsampling path. The U-Net architecture we employ in our framework is the "modern U-Net" architecture from Gupta & Brandstetter (2023), which closely resembles architectures used by diffusion models (Ho et al., 2020).

**CNO.** The Convolutional Neural Operator (Raonic et al., 2024) adapts the U-Net into the neural operator framework. Neural operators (Kovachki et al., 2023; Seidman et al., 2022) are a class of neural solver which aim to maintain the continuous nature of the underlying PDE solution despite the discretization of training data. In doing so, they enable trained neural solvers to be evaluated on discretizations differing from the training data. CNO extends these properties to the U-Net architecture using anti-aliasing techniques from Karras et al. (2021).

**F-FNO.** While the previous architectures take a hierarchical approach to multi-scale processing, Fourier Neural Operators (Li et al., 2021) process information on multiple scales in parallel using global Fourier convolutions. Convolution kernels are parameterized in the frequency domain such that the complex-valued weights to be learned represent the coefficients of the kernel function in the Fourier basis. Due to the convolution theorem, convolutions in the frequency domain are carried out via point-wise multiplication of frequency modes. Furthermore, because Fourier basis functions have global support, with high frequency modes describing fine details and low frequencies describing the "background" of the function, information on multiple spatial scales is processed in parallel. To execute these convolutions more efficiently, FNOs truncate the number of non-zero modes in each kernel to a threshold such that only the lowest frequencies are present in the Fourier expansion of the kernel function. Building on this efficiency, Factorized Fourier Neural Operators (Tran et al., 2023) perform convolutions one spatial dimension at a time such that kernels are a function of only one spatial dimension. This enables deeper F-FNOs, enhancing the expressive capacity of the architecture.

**Transolver.** As an alternative to parallel multi-scale processing with Fourier convolutions, attention enables mesh points both distant and local to share information with one another. However, due to its quadratic complexity, adapting Transformers to PDE modeling tasks, where the number of mesh points can be on the order of thousands and above, presents computational challenges (Li et al., 2023a; Cao, 2021; Hao et al., 2023). Transolver (Wu et al., 2024) reduces this complexity by performing attention on a coarsened mesh. The coarsening is achieved by a learnable soft pooling operation, where the soft assignments are not entirely based on clustering local points together.

**ConvNeXT.** For domains discretized on regular Cartesian grids, we employ a ConvNeXT neural CFL backbone (Liu et al., 2022b) to predict the timestep size. A convolutional stem with stride-4 downsampling followed by a sequence of ConvNeXT stages progressively reduces the spatial resolution while expanding the channel dimension. Each ConvNeXT stage is composed of residual blocks with depthwise convolutions and pointwise MLPs interleaved with layer normalization. After the final stage, global pooling over the spatial dimensions yields a single feature vector, which is normalized and passed through a linear head to produce the scalar timestep prediction.

**MeshGraphNets.** The MeshGraphNets (Pfaff et al., 2021) (MGN) architecture is a GNN model capable of solving PDEs on arbitrary geometries with possibly irregular domains, such as the airfoil shock setting we consider. Each computational cell is treated as a node whose features collect the local flow variables together, while edges connect nearby nodes and carry relative geometric information. In our airfoil shock setting, node features additionally include distance to the airfoil, while edge features are pairwise displacements and distances. We construct a $k$-nearest-neighbor graph with $k = 8$ using the mesh coordinates, and encode node and edge features with separate multilayer perceptrons. The encoded graph is then processed by a stack of message-passing blocks that alternate between edge updates and node updates, where edge blocks aggregate information from source and target nodes and update edge features, and node blocks aggregate incoming edge messages via a permutation-invariant reduction such as summation to update node states. Finally, a node decoder maps the processed node features back to the predicted flow variables on the mesh.

**Diffusion Graph Net.** The Diffusion Graph Net (DGN) (Valencia et al., 2025) combines a multi-scale graph neural network (Lino et al., 2022; Gao & Ji, 2019) with timestep conditioning to model flow dynamics on arbitrary geometries. While the original architecture was used as a denoising diffusion model (Valencia et al., 2025), we instead utilize it as a timestep-conditioned neural solver. We use the same input node features, edge features, and graph construction process as described for MGN. Starting from the input graph, a sequence of mesh coarsening operations constructs progressively coarser graphs by clustering nearby nodes and pooling their incident edges. At each resolution, interaction-network blocks update node and edge features by aggregating messages over local neighborhoods. Information is then propagated back to the finest scale via graph unpooling layers with skip connections, yielding a U-Net–style encoder–decoder hierarchy in graph space. A final linear decoder maps the latent node features at the finest scale.

**DGN Neural CFL.** For domains with non-uniform geometries, we reuse the encoder (downsampling) half of the DGN backbone as a neural CFL model to predict the timestep size. We use the same graph construction and node/edge features as in the DGN solver. The resulting graph is passed through the multi-scale message-passing hierarchy: at each scale, interaction-network blocks update node and edge features, followed by mesh coarsening layers that cluster nearby nodes and pool their incident edges into a progressively coarser sequence of graphs. On the final (coarsest) graph, we aggregate node features using global mean pooling over all nodes, and feed the resulting global embedding through a linear decoder to produce the predicted timestep size.

## C TRAINING DETAILS

### C.1 DATASETS

We coarsen the coal dust explosion cases in time by saving every 100 steps, and applied further coarsening to the saved steps by a factor of 5 for an overall coarsening factor of $500\times$ relative to the CFD solver. The coarsest AMR level gave a spatial resolution of $104 \times 520$. We only use the leftmost fifth of the domain along the horizontal axis such that the training resolution was $104 \times 104$. We train models on the velocity and temperature fields of the gas, as well as the volume fraction describing the percentage of coal comprising each computational cell.

For the circular blast cases, we save solutions every 100 CFD steps. The coarsest AMR level for the circular blast setting gives a spatial resolution of $256 \times 256$, which we coarsened further to the training resolution of $128 \times 128$ using averaging. We train models on the velocity, temperature and density fields.

The initial shock strength and particle diameters for the Coal Dust Explosion are sampled randomly. Half of them are sampled using uniform sampling and the remaining half are sampled using Bridson's algorithm (Bridson, 2007) to fill the parameter space. The initial pressure ratios for the Spherical Blast dataset are uniformly spaced. All of the initial shock strength and angle of attacks for the airfoil shock cases are sampled uniformly. In all settings, the evaluation cases are selected so as to be as diverse as possible by first sampling points in the parameter space using Bridson's algorithm and then finding the corresponding case closest to the sampled point.

### C.2 DATASET STATISTICS

We visualize the distribution of dataset trajectory parameters in Figure 12. In Figure 13, we visualize timestep sizes, while in Figure 14, we visualize the distribution of the number of steps per solution.

### C.3 TRAINING PIPELINE

We implement our training pipeline in PyTorch (Paszke et al., 2019) using PyTorch Lightning (Falcon, 2019). Depending on model training memory requirements, we train models on between 1-2 80 GiB A100 GPUs or between 1-8 11 GiB RTX 2080 GPUs. We optimize all models using the Adam optimizer (Kingma & Ba, 2015) with a cosine learning rate scheduler (Loshchilov & Hutter, 2017). We train all neural solver models for 400 epochs using a batch size of 32, resulting in over 75K training updates for the coal dust explosion dataset and over 50K training updates for both the circular blast dataset and the airfoil shock dataset. On the coal dust explosion and circular blast

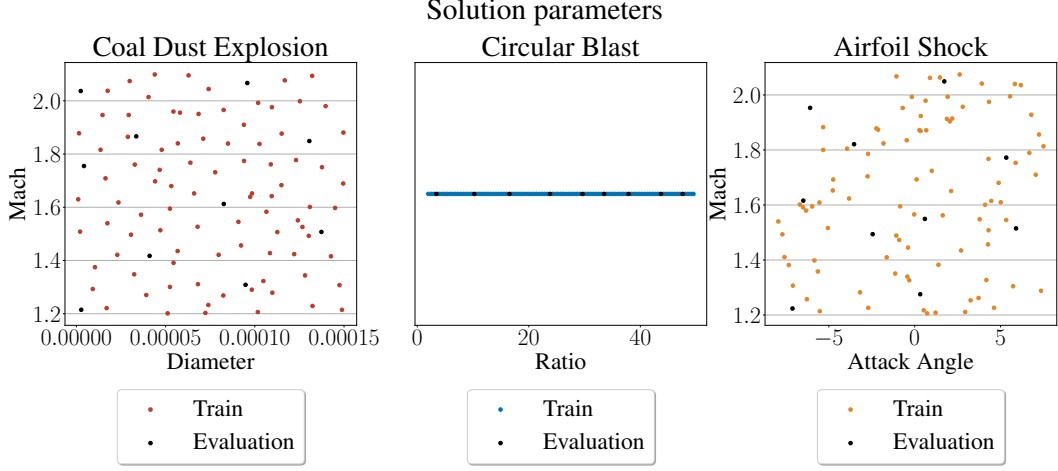

Figure 12: Distribution of solution parameters.

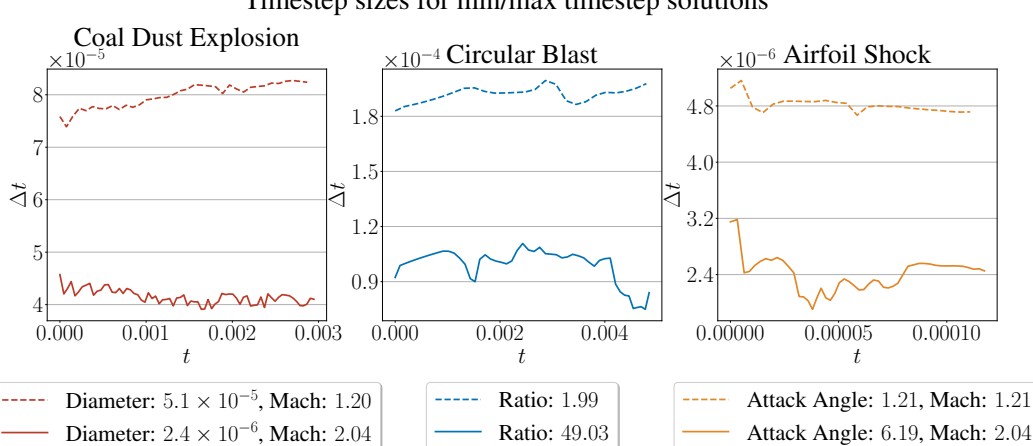

Figure 13: Timestep size as a function of time for the cases with the least and most number of steps in each setting.

datasets, we train neural CFL models for 800 epochs using a batch size of 320 using training noise with a level of 0.01 (Sanchez-Gonzalez et al., 2020). On the coal dust explosion dataset, this results in over 12K training updates, while for the circular blast dataset, this results in over 6K training updates. For the airfoil shock dataset, we train the neural CFL model for 400 epochs using a batch

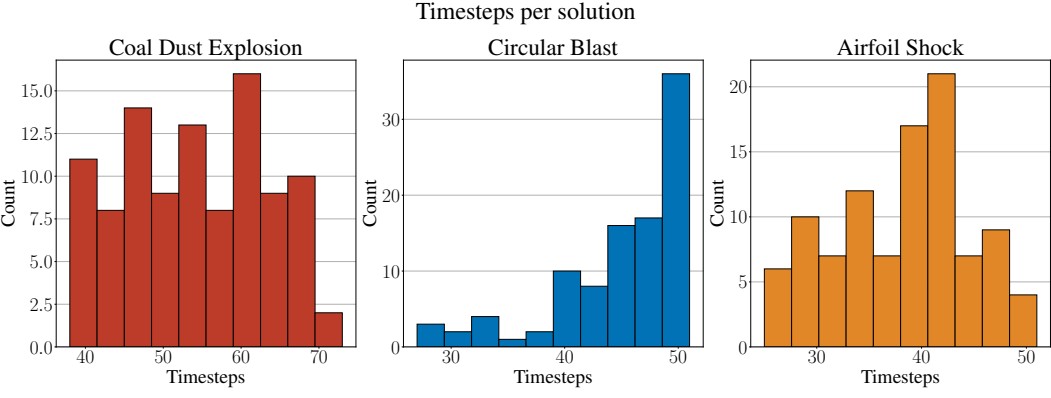

Figure 14: Number of timesteps per solution.

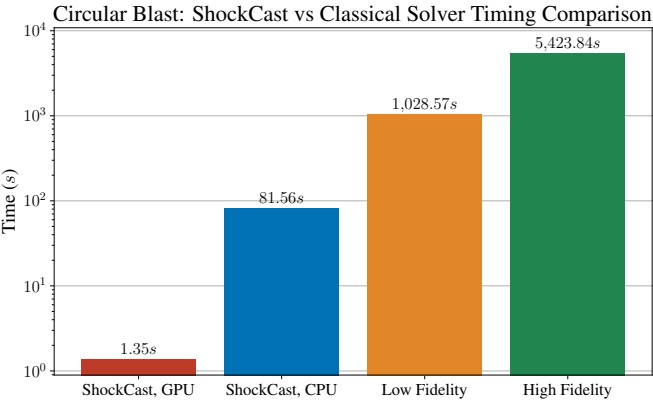

Figure 15: Timing comparison for ShockCast on CPU and GPU versus low and high fidelity classical solver for a circular blast case.

size of 32 with training noise with a level of $0.01$, resulting in over 50K training updates. We present initial learning rates and other key hyperparameters used for all models in Table 2.

## C.4 PARAMETER COUNTS, FLOPS AND PEAK GPU MEMORY

In Table 3, we present parameter counts, forward FLOPs, and peak training memory for the coal dust explosion setting and the airfoil shock setting. We compute FLOPs with a batch size of one using `FlopCounterMode` from the `torch.utils.flop_counter` module, and report training memory observed on a single A100 GPU using a batch size of 32 in the coal dust explosion setting and on a single RTX 6000 Ada GPU with a batch size of 4 in the airfoil shock setting.

In our experiments, we offset increased computation when using the MoE timestep conditioning strategy by reducing the latent dimension of all models except CNO according to the square root of the number of experts used. We use four experts for all models except for Transolver, where we use two experts due to high memory consumption. For CNO, we ran into stability issues when attempting to scale the number of parameters beyond those reported in (Herde et al., 2024). As can be seen in Table 3, this led to a reduced amount of computation for the CNO variants of ShockCast relative to the other neural solver backbones. However, when using MoE timestep conditioning, we found that we could stably train CNO with 4 experts with no reduction in the embedding dimension.

## C.5 SOLUTION RUNTIME ANALYSIS

We compare the runtime of the classical solver to ShockCast with the F-FNO neural solver backbone using affine timestep conditioning following the best practices described in McGreivy & Hakim (2024). Specifically, we report both GPU and CPU runtimes for ShockCast, and additionally report both high-fidelity and low-fidelity runtimes for the classical solver. The high-fidelity classical solver refers to the use of the original solver settings, while low-fidelity uses a setting that trades off a degree of accuracy comparable to the neural solver error for increased efficiency. We compare using a circular blast case from the test set with initial pressure ratio $47.575$. The low fidelity solver is achieved by removing the finest level of the mesh. Times are shown in Figure 15, while per-step errors relative to the high-fidelity solution are shown in Figure 16. The high fidelity, low fidelity, and neural solutions are visualized in Figure 17.

In Table 4, we present the time to compute a solution for ShockCast across all neural solvers backbones and conditioning methods in both the Coal Dust Explosion and Circular Blast settings, timed on both GPU and CPU-only. We report runtimes for the high fidelity classical solver in the Coal Dust Explosion setting in Table 5.

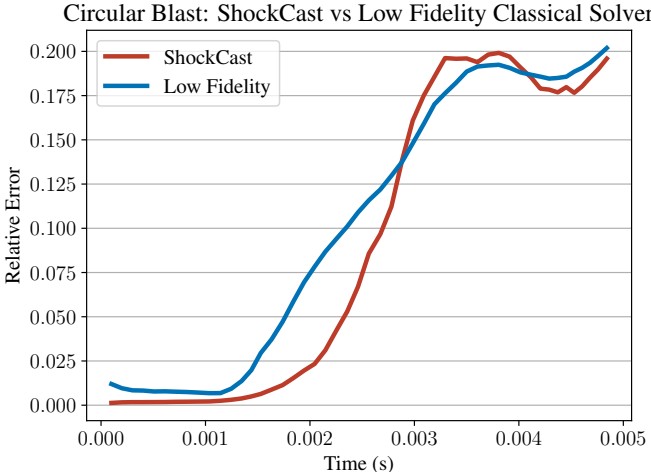

Figure 16: Time versus Relative error for ShockCast solution and Low Fidelity classical solver solution for the circular blast setting.

| Coal Dust Explosion: Runtime (s) | | | Circular Blast: Runtime (s) | | |
|---|---|---|---|---|---|
| Model | GPU | CPU | Model | GPU | CPU |
| CNO | | | CNO | | |
| Affine | 2.15 (0.07) | 42.51 (0.15) | Affine | 1.67 (0.03) | 40.58 (0.47) |
| Euler | 2.30 (0.03) | 42.57 (0.08) | Euler | 1.94 (0.01) | 41.86 (0.54) |
| MoE | 7.14 (0.01) | 108.51 (0.28) | MoE | 6.12 (0.10) | 117.45 (2.34) |
| F-FNO | | | F-FNO | | |
| Affine | 1.41 (0.03) | 58.05 (0.41) | Affine | 1.15 (0.01) | 56.32 (2.14) |
| Euler | 1.42 (0.01) | 56.38 (0.04) | Euler | 1.18 (0.01) | 56.45 (2.23) |
| MoE | 3.75 (0.04) | 82.49 (0.53) | MoE | 3.12 (0.03) | 79.91 (2.56) |
| Transolver | | | Transolver | | |
| Affine | 2.22 (0.03) | 159.82 (1.10) | Affine | 2.34 (0.01) | 194.56 (4.49) |
| Euler | 2.21 (0.04) | 167.54 (6.15) | Euler | 2.33 (0.03) | 198.53 (5.82) |
| MoE | 3.39 (0.03) | 175.19 (0.56) | MoE | 3.76 (0.04) | 226.63 (4.59) |
| U-Net | | | U-Net | | |
| Affine | 1.66 (0.01) | 85.44 (0.69) | Affine | 1.61 (0.01) | 94.76 (2.43) |
| Euler | 1.66 (0.01) | 87.36 (0.33) | Euler | 1.64 (0.01) | 95.49 (2.16) |
| MoE | 2.78 (0.01) | 79.29 (0.67) | MoE | 2.51 (0.04) | 84.96 (1.06) |

Table 4: ShockCast runtime to compute a solution via autoregressive unrolling in both settings on CPU and GPU, presented as *mean (standard error)*.

| Minimum | Mean | Maximum |
|---|---|---|
| 15,592 | 67,441 | 128,675 |

Table 5: Classical solver runtime to compute a solution on 16 CPU cores in seconds for the Coal Dust Explosion setting.

Circular Blast, Low Fidelity Solver: Density

Figure 17: Density for circular blast, low fidelity classical solver comparison.

## D   METRICS

**Correlation time proportion.**     We compute Pearson's correlation coefficient for each field and at each timestep with the ground truth data. This requires the predicted fields to be on the same temporal grid as the ground truth data, for which we use linear interpolation. Specifically, for the function $\boldsymbol{f} : \mathbb{R} \to \mathbb{R}^d$ sampled on the grid $\{t_j\}_j^n$ where $t_j \leq t_{j+1}$, we interpolate $\boldsymbol{f}$ to the query point $t^\star \in [t_1, t_n]$ as

$$\boldsymbol{f}(t^\star) \approx \sum_{j}^{n-1} \mathbb{1}_{[t_j, t_{j+1}]}(t^\star) \left( \frac{\boldsymbol{f}(t_j)(t_{j+1} - t^\star) + \boldsymbol{f}(t_{j+1})(t^\star - t_j)}{t_{j+1} - t_j} \right),$$

where $\mathbb{1}_{[a,b]}$ is the characteristic function given by

$$\mathbb{1}_{[a,b]}(x) = \begin{cases} 1 & x \in [a, b] \\ 0 & o.w. \end{cases}.$$

The correlation time (Kochkov et al., 2021; Lippe et al., 2023; Alkin et al., 2024) for a given field is defined as the last time $t$ before the correlation sinks below a threshold, which we take to be $0.9$ here. We then average this time across all fields and report it as a proportion of the full simulation time such that a perfect prediction would have correlation time proportion of $1$.

**Mean flow and Turbulence Kinetic Energy.**     The mean flow is defined as the flow states averaged over time, while the turbulence kinetic energy is the sum of the variances of the fluctuating part of the velocity field. These quantities are given by

$$\bar{\boldsymbol{u}} := \frac{1}{T} \int_0^T \boldsymbol{u}(t) \mathrm{d}t \qquad \text{TKE} := \frac{1}{2T} \int_0^T (u(t) - \bar{u})^2 + (v(t) - \bar{v})^2 \mathrm{d}t. \qquad (5)$$

We approximate the integrals in Equation (5) using the trapezoidal rule as

$$\int_0^T f(t) \mathrm{d}t \approx \frac{1}{2} \sum_{j}^{n-1} \Delta_j \left( f(t_{j+1}) + f(t_j) \right),$$

where $\Delta_j := t_{j+1} - t_j$.

## E   HYPERSONIC AIRFOIL SHOCK

To evaluate the ability of ShockCast to handle hypersonic (Mach $> 5$) settings, we extend the Airfoil Shock setting by varying the initial shock strength from Mach 5 to Mach 6 at a fixed angle of attack of $0°$. We found that despite having good performance on the airfoil shock setting, a moderately-sized MGN model (16 message passing layers, embedding dimension of 128) did not have sufficient capacity for the hypersonic dynamics despite having a large GPU footprint. In particular, we found that the $y$-velocity field was especially challenging to model and thus, we only model the $x$-velocity, density, and temperature fields in this setting. Additionally, we employed a stronger U-Net model adapted to handle the non-uniform mesh. Specifically, we interpolated the non-uniform domain to a fine, uniformly spaced $256 \times 1024$ grid. The U-Net projected input fields to latent space through a point-wise linear projection applied to the mesh points away from the airfoil surface, and a learned embedding vector for mesh points on and inside of the airfoil. The U-Net then processed the data as usual. Following the final layer of the U-Net, we masked mesh points on and inside of the airfoil out of the loss. We visualize rollout predictions for a held-out test case in Figures 18 to 20.

## F   MASS CONSERVATION ANALYSIS

We analyze the physical consistency of predictions made by ShockCast by examining mass conservation in the circular blast setting. As the boundaries for the blast case do not allow any fluid to enter or exit, the total mass should remain constant across time up to numerical precision. That is, the mass function $M(t) : \mathbb{R} \mapsto \mathbb{R}$ defined as

$$M(t) = \int_\Omega \rho(\boldsymbol{x}, t) d\boldsymbol{x}$$

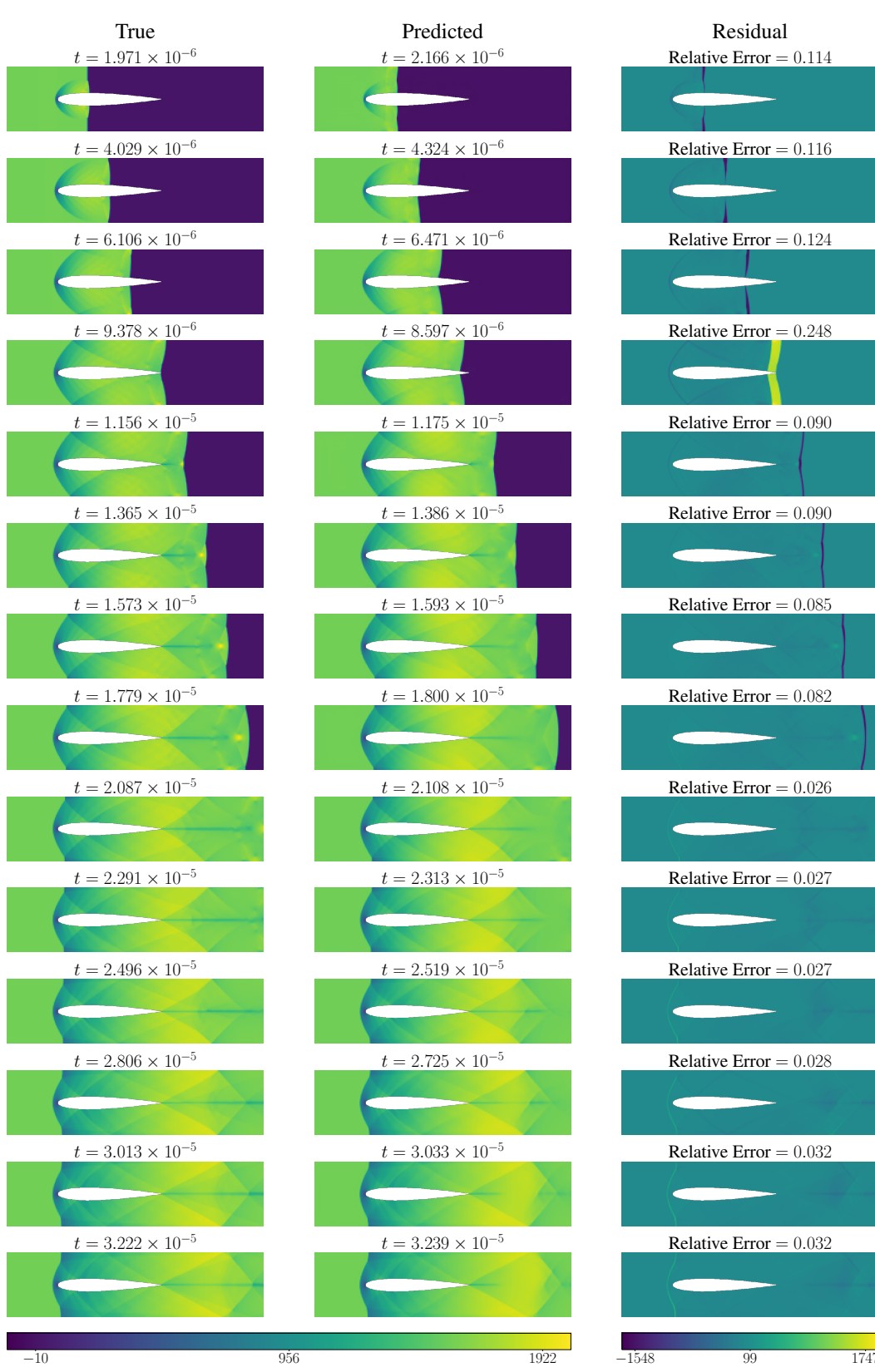

Figure 18: Gas velocity $x$-component for hypersonic airfoil shock.

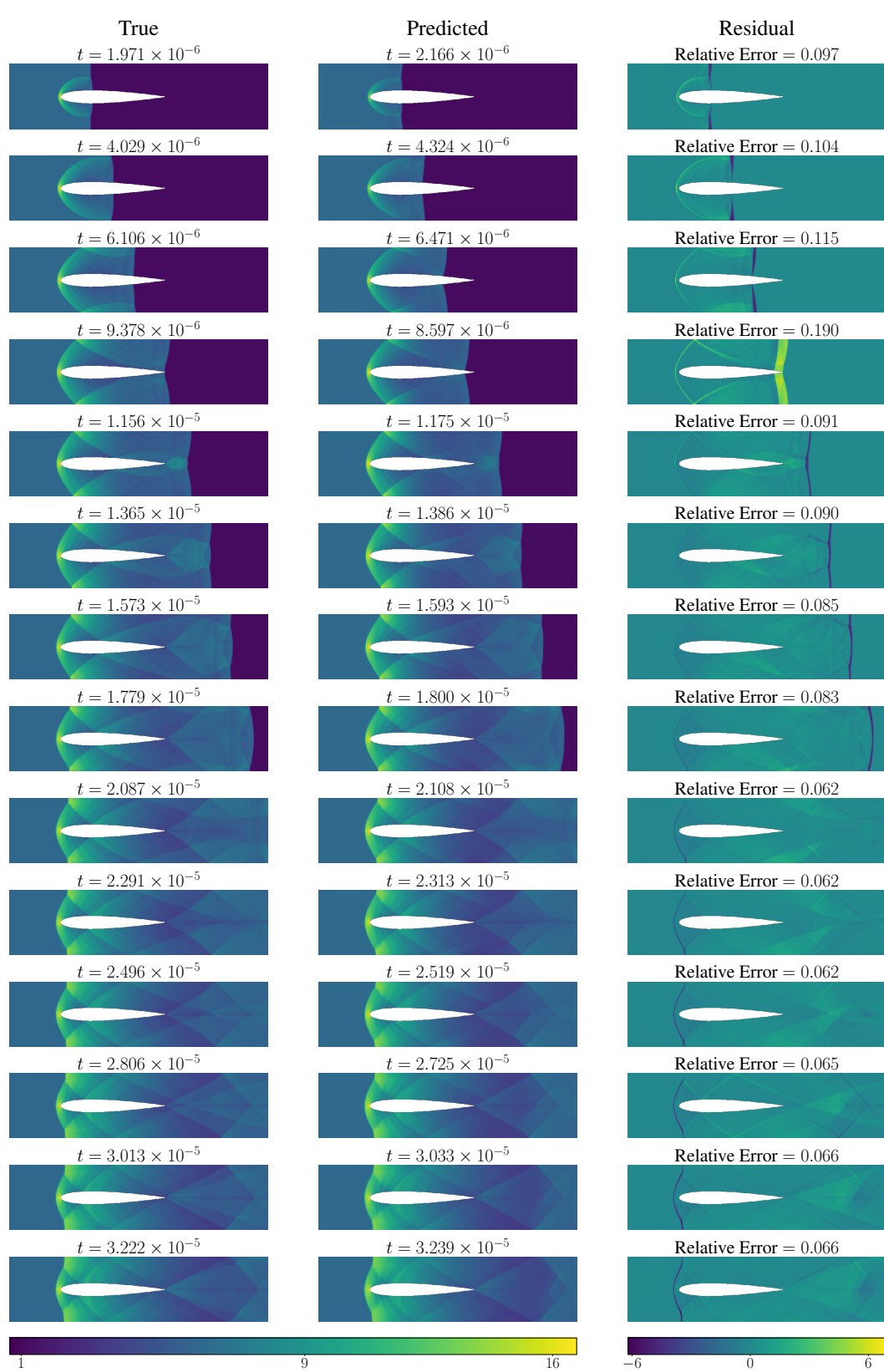

Figure 19: Gas density for hypersonic airfoil shock.

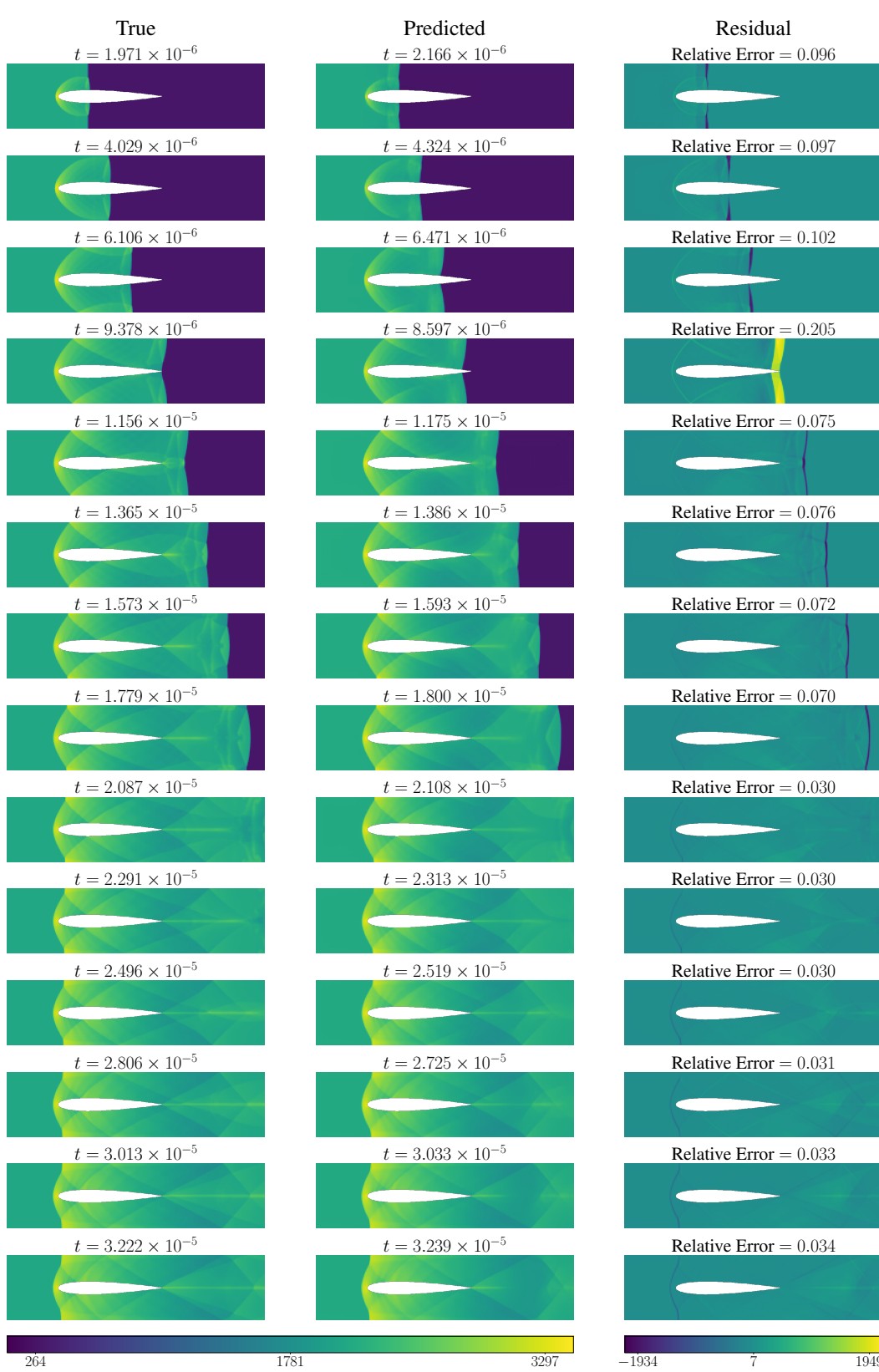

Figure 20: Gas temperature for hypersonic airfoil shock.

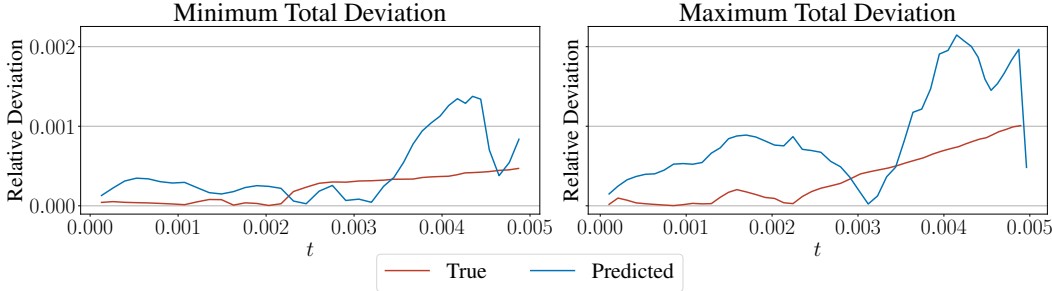

Figure 21: Relative mass deviation for the two cases with the minimum and maximum prediction total deviation, where the total is computed by summing over time.

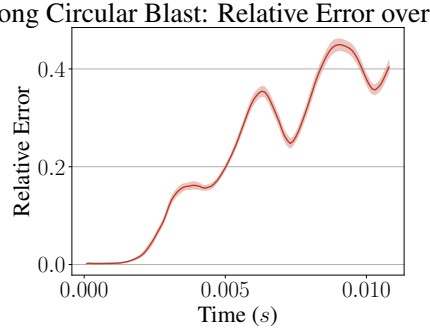

Figure 22: Visualization of per-step relative rollout error for the long circular blast cases. The error is averaged over fields. The solid line is the mean over the test set, and the shaded region is two times the standard error.

should remain constant, where $\Omega$ is the spatial domain and $\rho$ is the density. To analyze the evolution of the mass function over time, we define the relative mass deviation as

$$\frac{|M(t) - M(0)|}{|M(0)|},$$

which is positive for $t > 0$ if the mass has changed since the simulation started. As can be seen in Figure 21, ShockCast's predicted total mass stays within $0.2\%$ of $M(0)$.

## G  LONG CIRCULAR BLAST ANALYSIS

In this section, we analyze performance of ShockCast in the long circular blast setting with a U-Net backbone. The rollout error over time is visualized in Figure 22. Although the solution predicted by ShockCast has a larger relative error as more steps are taken, the rollout visualizations in Figures 23 to 26 show that predictions remain stable and in the distribution of solutions.

## H  NEURAL CFL ABLATION

In this section, we evaluate and ablate the neural CFL module in ShockCast by replacing it with two alternatives: an oracle model and a mean prediction model. The oracle conditions the neural solver on the ground-truth $\Delta t$ for each timestep. As discussed in Section 3.1, this is not a tractable approach, as the ground truth $\Delta t$ are determined by the full solution and therefore are not known before the solution is computed. However, comparing the performance of ShockCast to the oracle model is of interest, as the neural CFL model is trained using the oracle $\Delta t$ values. Therefore, the oracle represents an upper bound on achievable performance. The mean prediction model is more viable than the oracle and a simpler alternative to the neural CFL model. At each timestep, the mean prediction model predicts the mean $\Delta t$ computed from the training data, independent of the current flow state $\boldsymbol{u}(t)$.

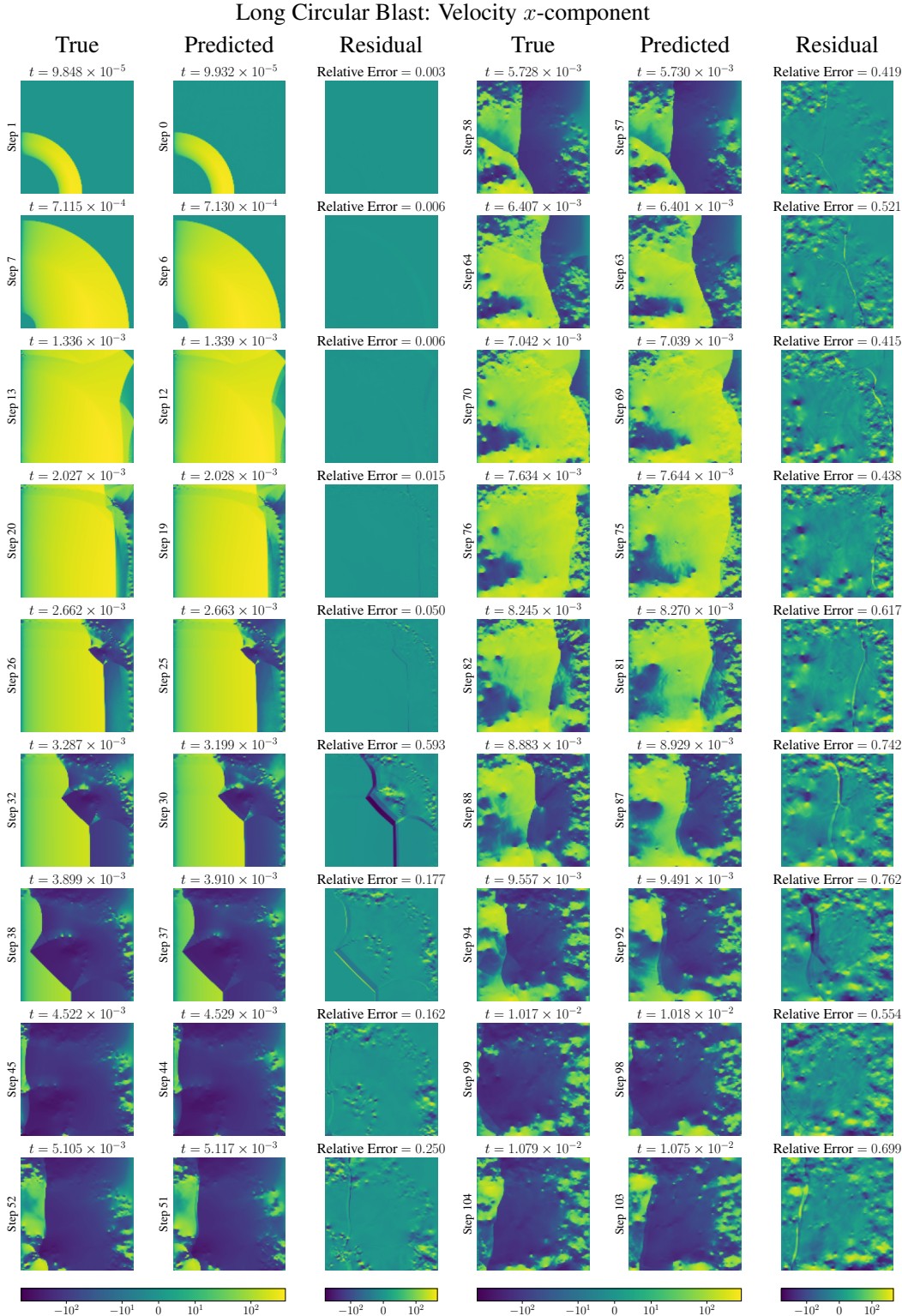

Figure 23: Velocity $x$-component for long circular blast.

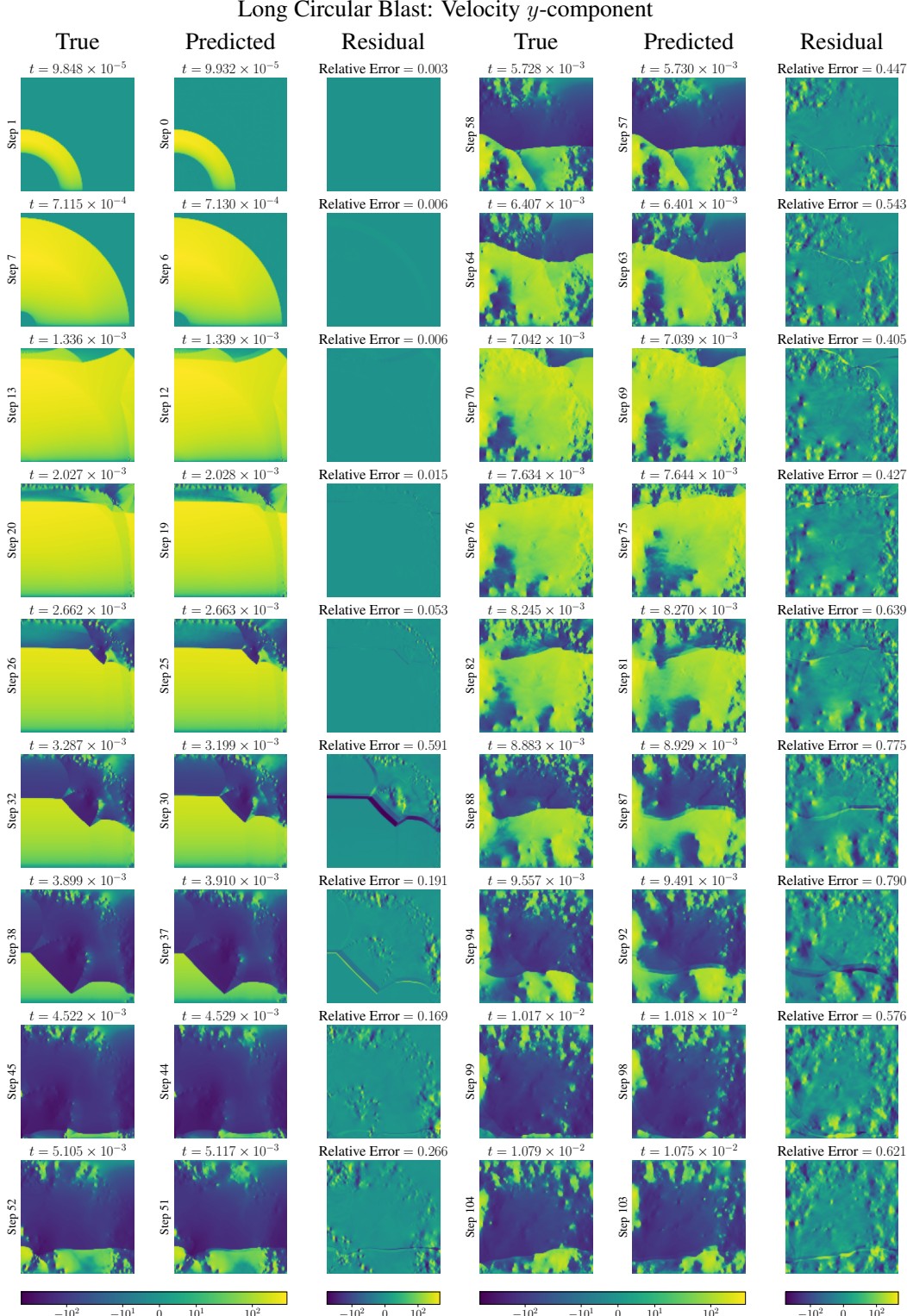

Figure 24: Velocity $y$-component for long circular blast.

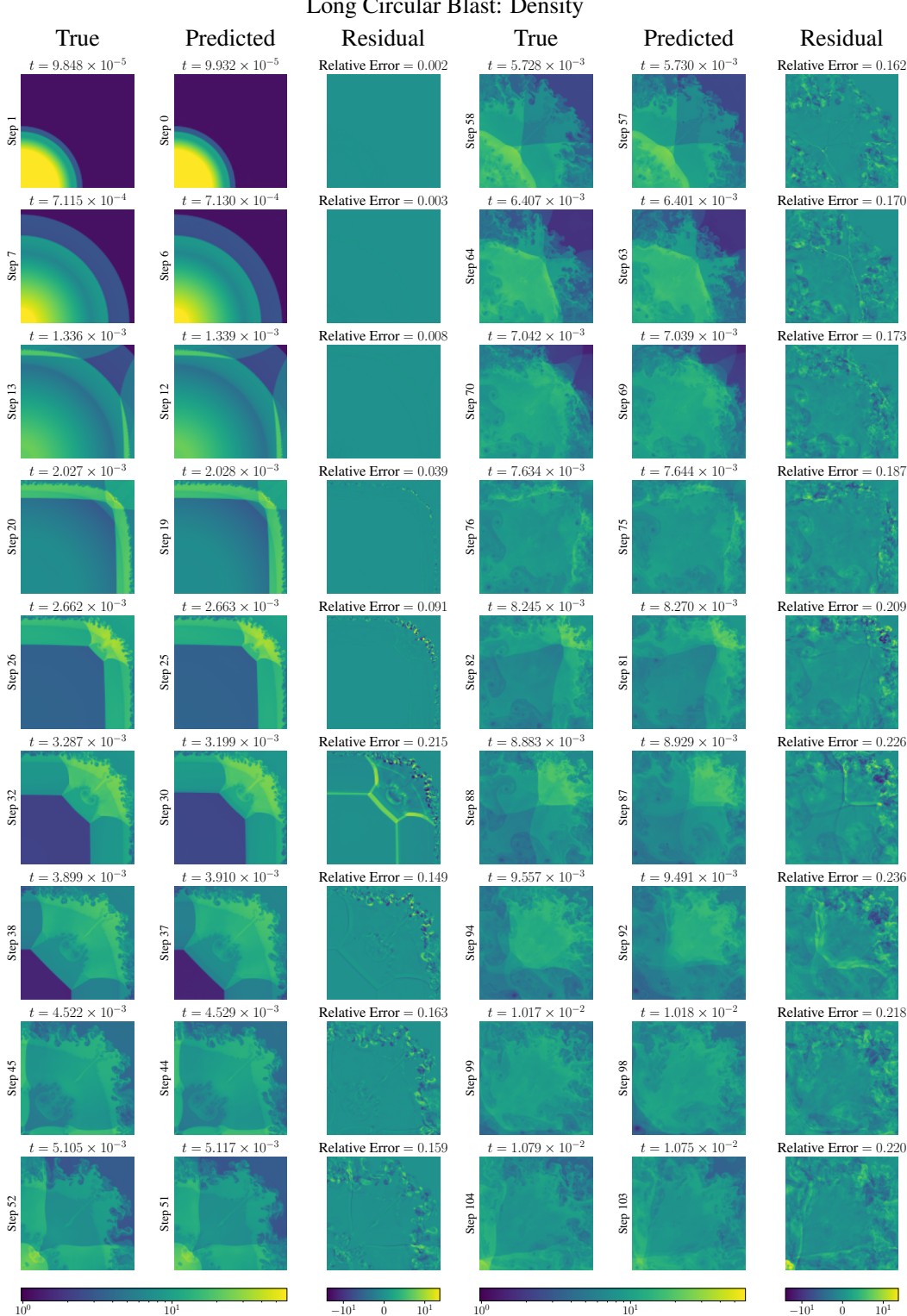

Figure 25: Density for long circular blast.

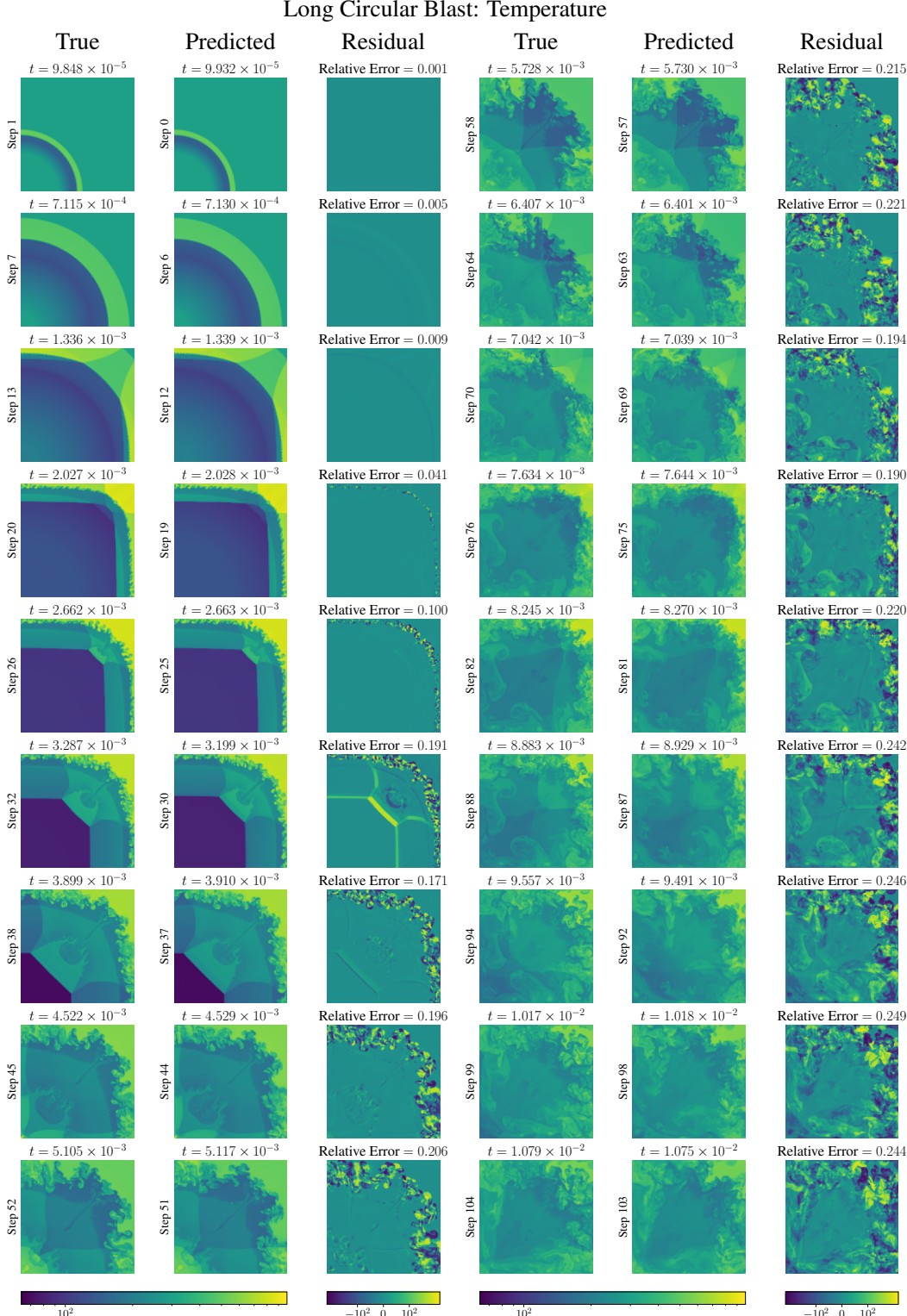

Figure 26: Temperature for long circular blast.

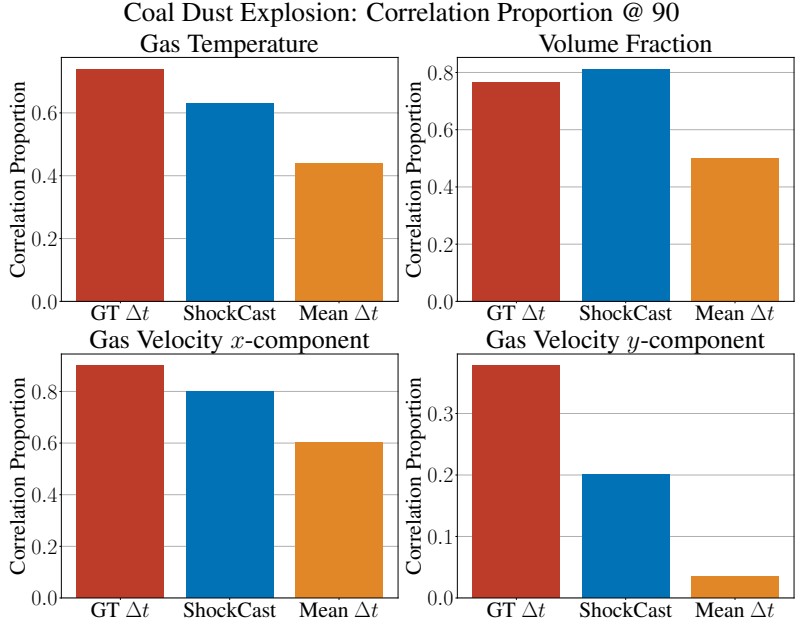

Figure 27: Ablation on the neural CFL model in the coal dust explosion setting.

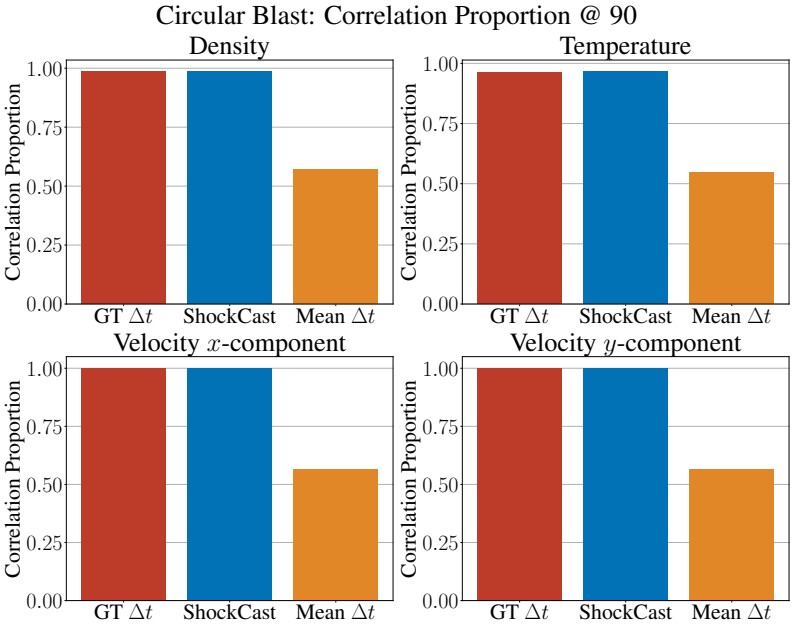

Figure 28: Ablation on the neural CFL model in the circular blast setting.

| | Coal Dust Explosion: Correlation Proportion Difference $\times 10^{-2}$ ($\downarrow$) | |
|---|---|---|
| Model | Mean Correlation Proportion | Mean Correlation Proportion Difference |
| CNO | | |
| Affine | 60.03 (0.53) | −9.24 (0.11) |
| Euler | 61.04 (0.92) | −12.00 (2.08) |
| MoE | 61.05 (0.85) | −11.58 (2.60) |
| F-FNO | | |
| Affine | 61.05 (0.16) | −14.70 (0.51) |
| Euler | 60.95 (0.13) | −15.76 (0.56) |
| MoE | 59.88 (0.94) | −17.42 (0.94) |
| Transolver | | |
| Affine | 60.55 (0.76) | −11.61 (2.27) |
| Euler | 59.89 (1.10) | −10.47 (0.39) |
| MoE | 60.22 (0.17) | −8.98 (0.74) |
| U-Net | | |
| Affine | 61.25 (0.12) | −9.84 (0.72) |
| Euler | 60.89 (0.26) | −9.66 (0.52) |
| MoE | 60.96 (0.29) | −10.57 (0.64) |

Table 6: Correlation proportion averaged over fields for ShockCast and difference between the ShockCast correlation proportion and the correlation proportion for the neural solver component using the ground truth $\Delta t$ in the circular blast setting.

Figure 29: Feature importance analysis for the Neural CFL model in the coal dust explosion setting.

We present results for the oracle (GT $\Delta t$) and mean prediction (Mean $\Delta t$) models compared to ShockCast using a U-Net backbone with time-conditioned layer norm in Figures 27 and 28. In both settings, we observe that the mean model performs substantially worse than ShockCast. This is likely because the predicted $\Delta t$ for the mean prediction model is independent of the current flow state $\boldsymbol{u}(t)$, despite the dependence between these variables observed in Figure 13. This results in the mean timestep size $\Delta t$ and current flow state $\boldsymbol{u}(t)$ being an out-of-distribution input pair for the neural solver. We additionally observe that ShockCast achieves performance close to its upper bound in the oracle model. We provide comprehensive comparisons of ShockCast to the oracle model with various neural solver backbones and conditioning methods in Tables 6 and 7.

## I FEATURE IMPORTANCE ANALYSIS

To assess the degree to which each of the input features to the neural CFL model influence predictions, we conduct a feature importance analysis by progressively corrupting each feature in the coal dust explosion setting. We analyze the effect this perturbation has by measuring the Mean Absolute

| Circular Blast: Correlation Proportion Difference $\times 10^{-2}$ ($\downarrow$) | | |
|---|---|---|
| Model | Mean Correlation Proportion | Mean Correlation Proportion Difference |
| CNO | | |
| Affine | 96.61 (0.15) | 0.16 (0.11) |
| Euler | 96.56 (0.08) | 0.07 (0.10) |
| MoE | 96.46 (0.02) | 0.05 (0.03) |
| F-FNO | | |
| Affine | 97.84 (0.14) | 0.10 (0.08) |
| Euler | 97.81 (0.05) | 0.01 (0.02) |
| MoE | 97.62 (0.03) | 0.12 (0.06) |
| Transolver | | |
| Affine | 96.92 (0.24) | 0.18 (0.09) |
| Euler | 96.87 (0.17) | 0.17 (0.09) |
| MoE | 96.82 (0.10) | 0.30 (0.05) |
| U-Net | | |
| Affine | 98.34 (0.26) | 0.04 (0.04) |
| Euler | 98.09 (0.16) | 0.22 (0.09) |
| MoE | 97.82 (0.05) | 0.08 (0.02) |

Table 7: Correlation proportion averaged over fields for ShockCast and difference between the ShockCast correlation proportion and the correlation proportion for the neural solver component using the ground truth $\Delta t$ in the circular blast setting.

Normalized Error, defined as

$$\frac{|\Delta - \hat{\Delta}|}{\sigma_\Delta},$$

where $\Delta$ and $\hat{\Delta}$ are the ground truth and predicted timesteps, and $\sigma_\Delta$ is the standard deviation of the timestep sizes over the training data. We corrupt inputs by applying dropout of increasing levels one field at a time. Results are shown in Figure 29. As can be seen, the flow state $\boldsymbol{u}(t)$ has the largest influence on predictions, followed by the partial derivative with respect to $y$. This is consistent with the physics of the coal dust explosion setting, in which the shock repeatedly reflects between the upper and lower channel boundaries, generating strong vertical gradients that require small timesteps to resolve as the shock approaches the upper or lower boundary.

## J    SOLUTION VISUALIZATIONS

In this section, we visualize true and predicted fields for ShockCast on selected solutions from the evaluation datasets. For the coal dust explosion and circular blast setting, ShockCast predictions are made using the F-FNO with Euler conditioning neural solver backbone, while for the airfoil shock setting, ShockCast predictions are made with the DGN neural solver. The selected solution has a shock Mach number of $1.85$ in the coal dust explosion setting, a max Mach number of $2.68$ in the circular blast setting, and a shock Mach number of $1.51$ in the airfoil shock setting. We visualize TKE fields in Figures 30 to 32 and mean flow fields in Figures 33 to 35. When visualizing the instantaneous fields, we subsample the true solution in time to reduce the number of snapshots. For each of the snapshots $\boldsymbol{u}(t)$ in the subsampled solution, we find the snapshot from the solution $\hat{\boldsymbol{u}}(\hat{t})$ autoregressively unrolled by ShockCast with the closest predicted time $\hat{t}$ to $t$. We then visualize the true snapshots $\boldsymbol{u}(t)$ alongside the corresponding closest-in-time predicted snapshots $\hat{\boldsymbol{u}}(\hat{t})$, as well as the residual between each pair $|\boldsymbol{u}(t) - \hat{\boldsymbol{u}}(\hat{t})|$. We present these visualizations for each of the fields in Figures 36 to 39 for the coal dust explosion case, Figures 40 to 43 for the circular blast case, and Figures 44 to 47 for the airfoil shock case.

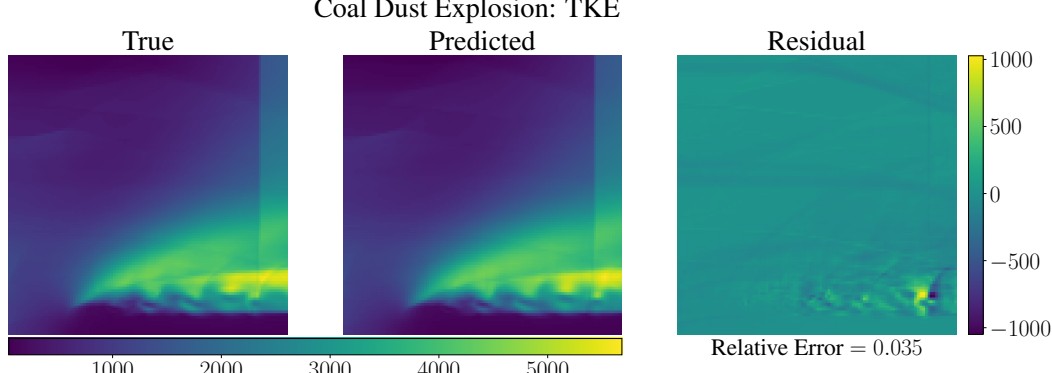

Figure 30: TKE for coal dust explosion.

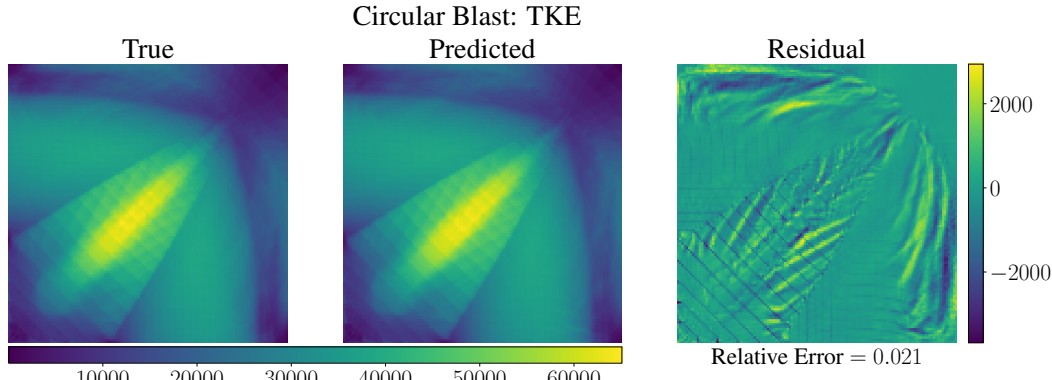

Figure 31: TKE for circular blast.

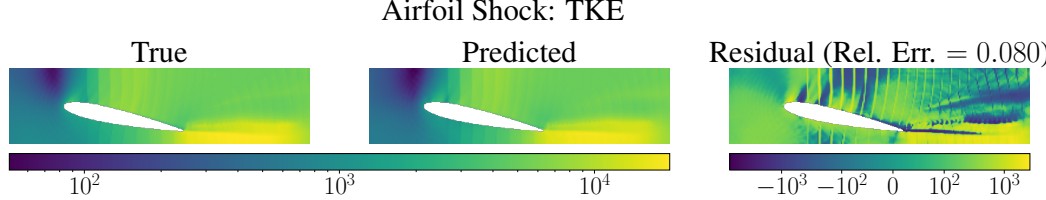

Figure 32: TKE for airfoil shock.

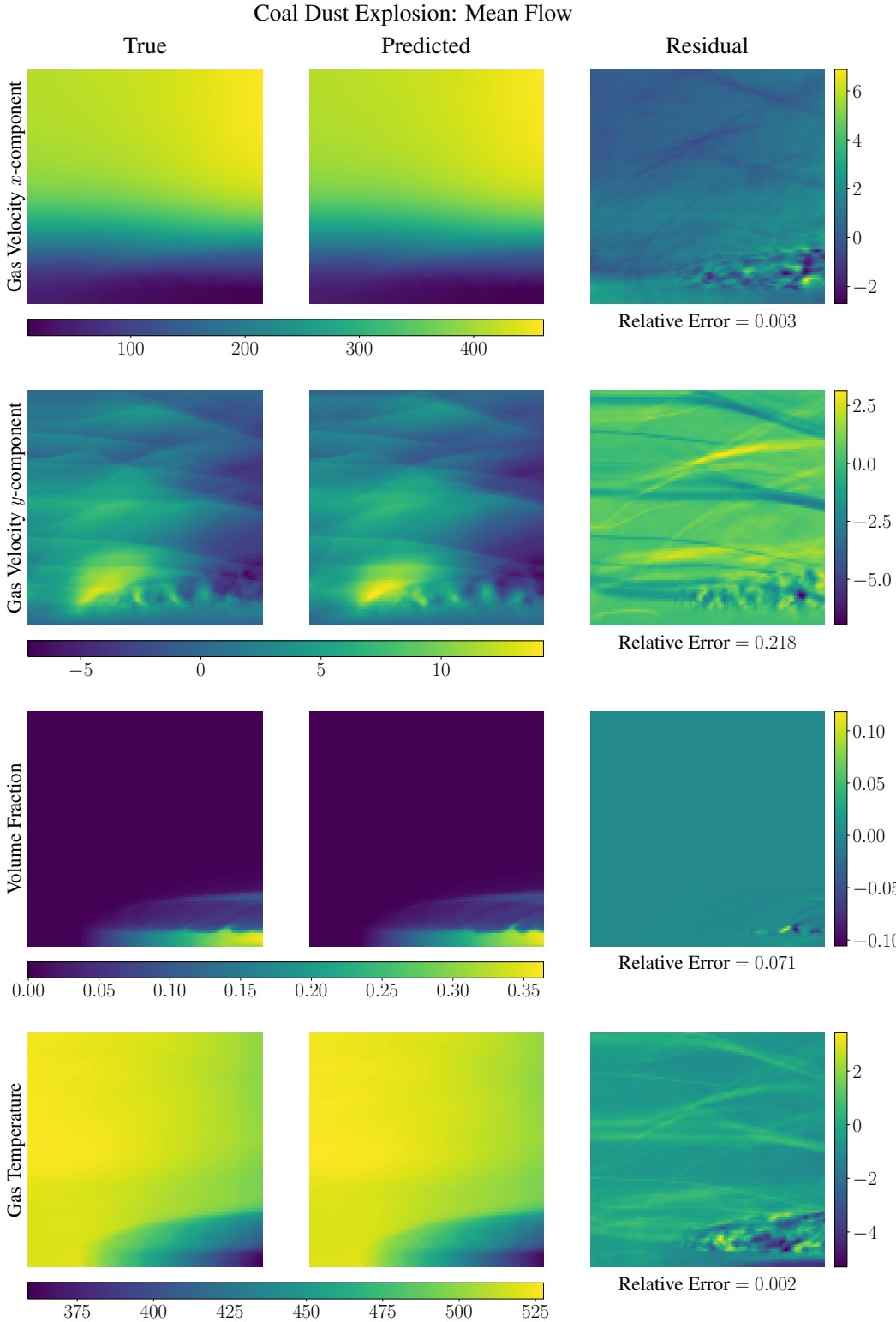

Figure 33: Mean flow for coal dust explosion.

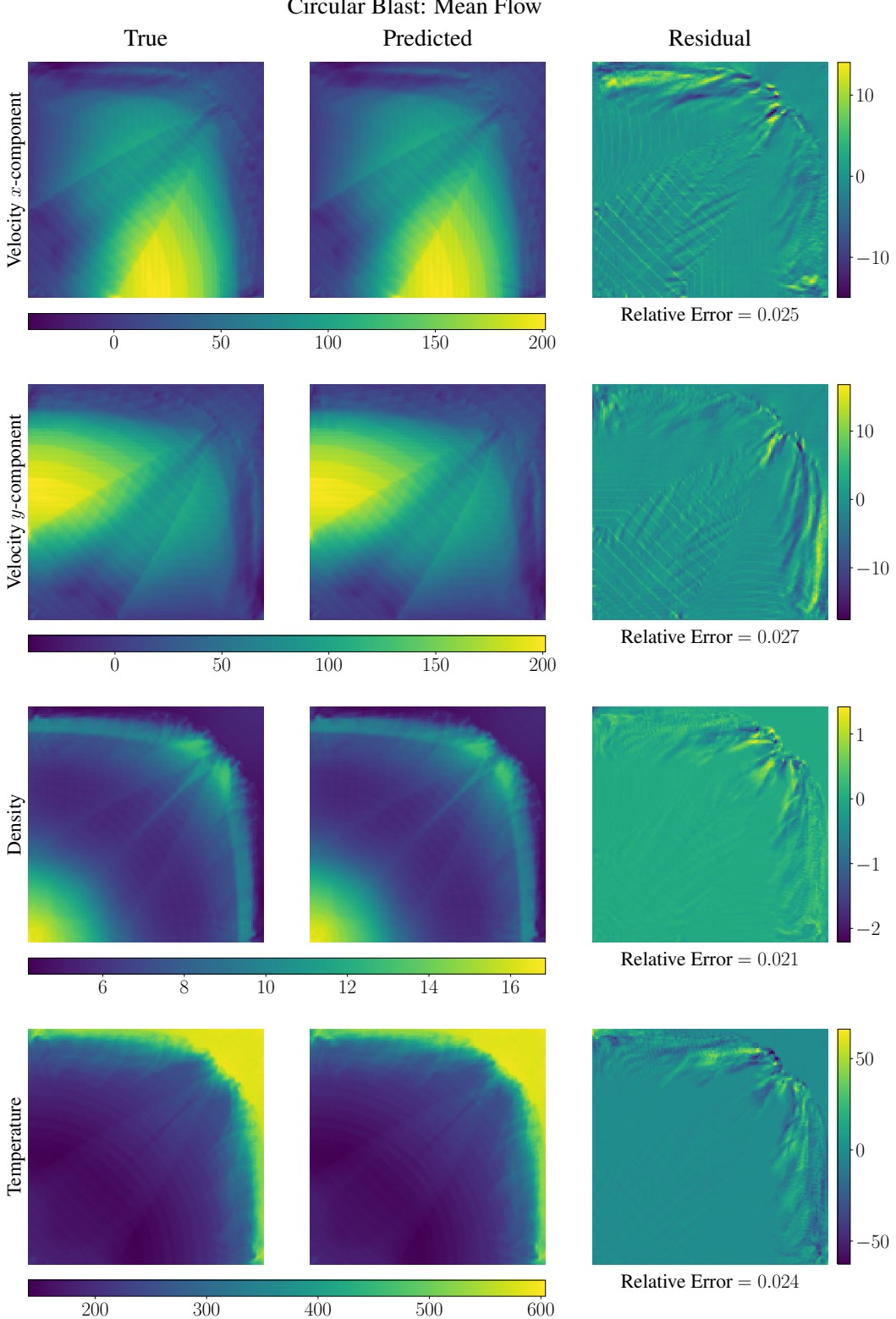

Figure 34: Mean flow for circular blast.

Airfoil Shock: Mean Flow

Figure 35: Mean flow for airfoil shock.

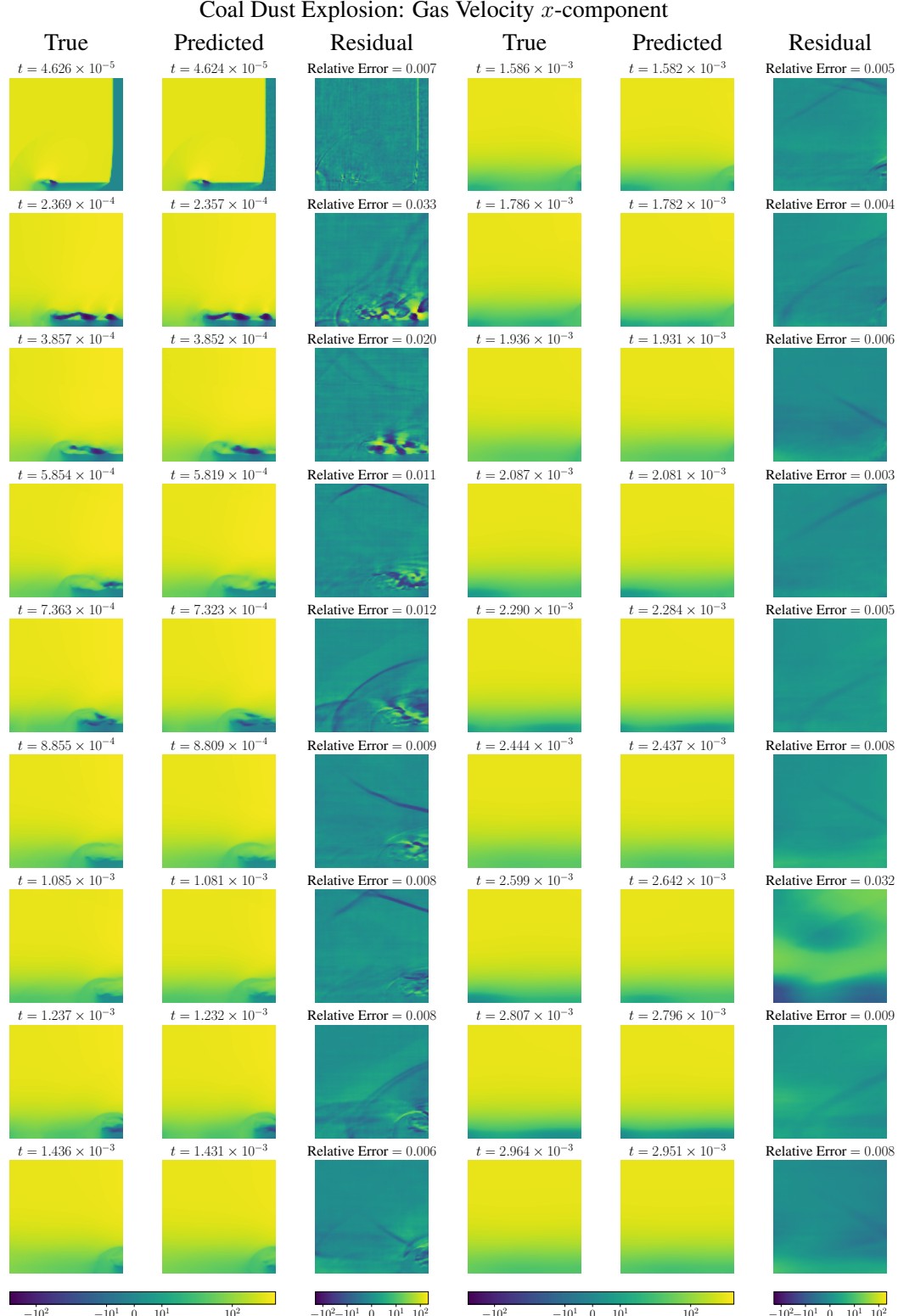

Figure 36: Gas velocity $x$-component for coal dust explosion.

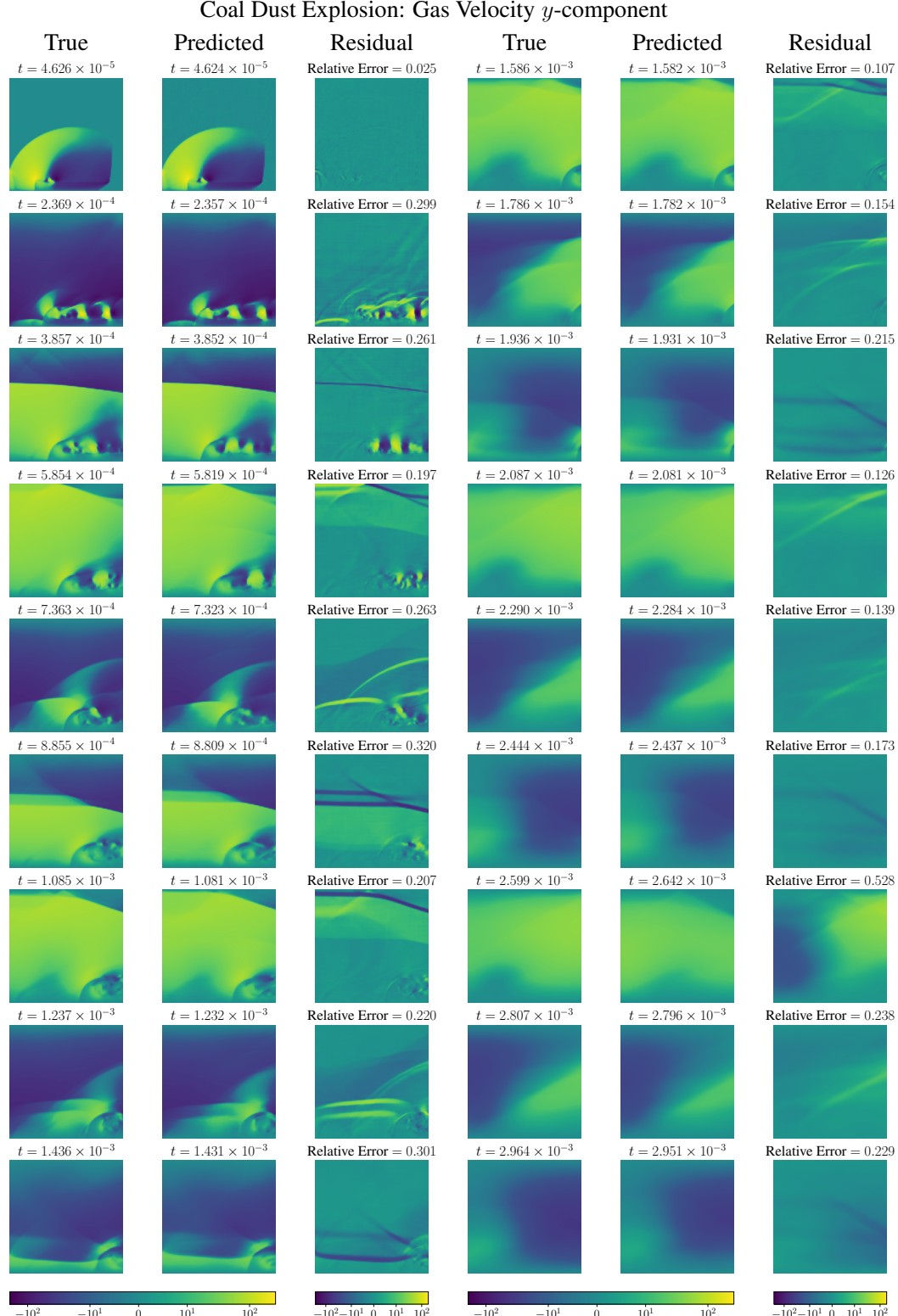

Figure 37: Gas velocity $y$-component for coal dust explosion.

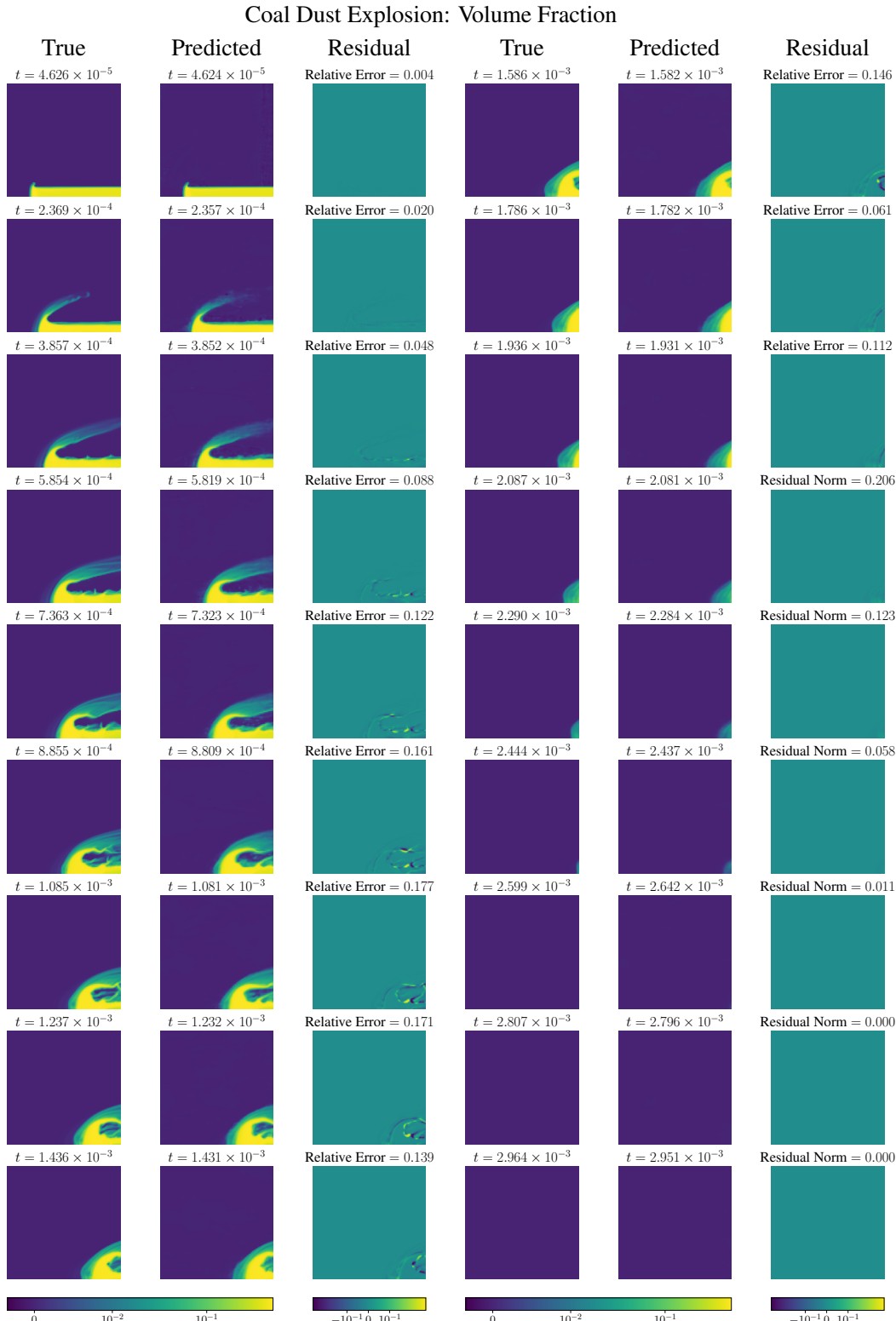

Figure 38: Volume fraction for coal dust explosion.

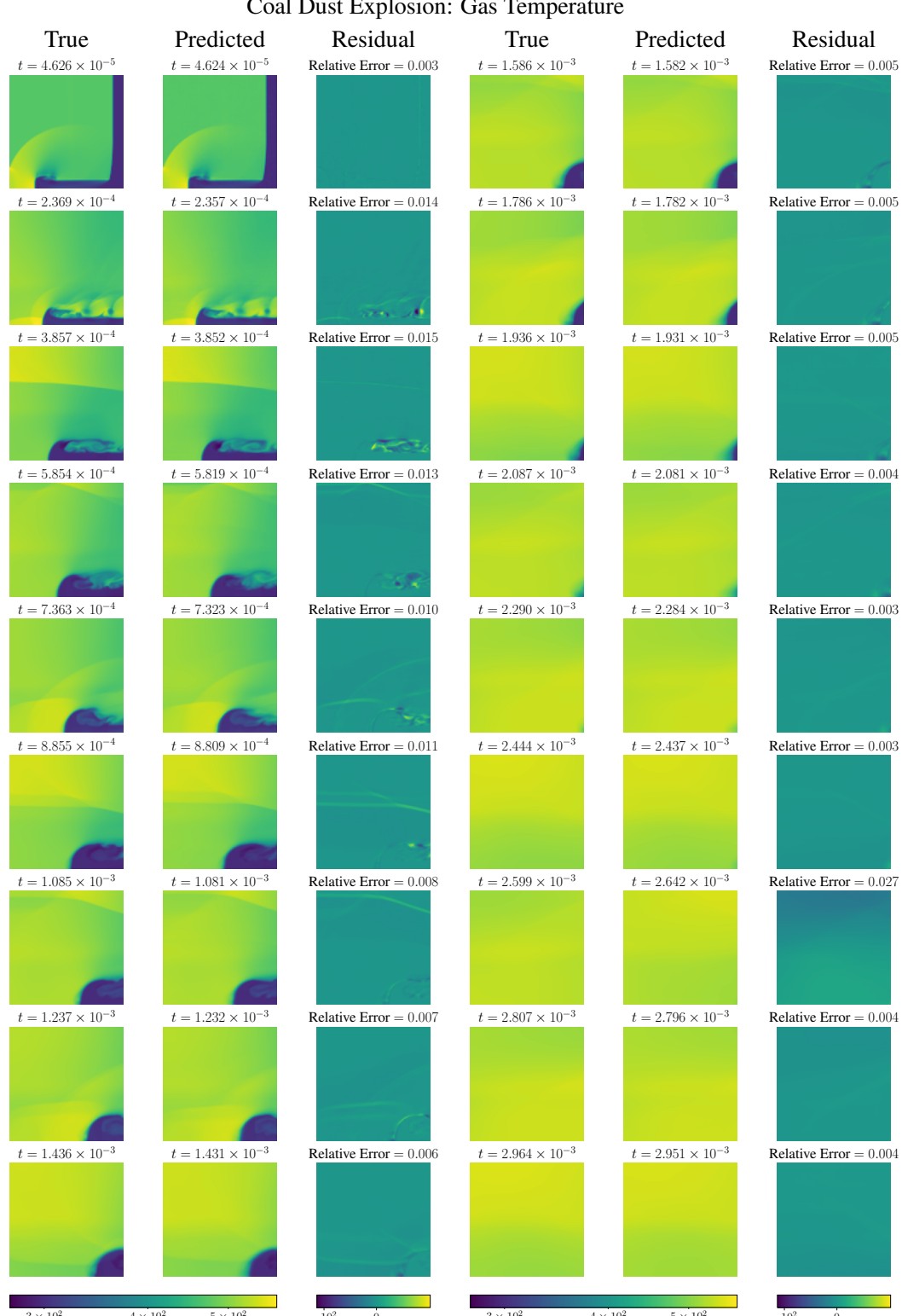

Figure 39: Gas temperature for coal dust explosion.

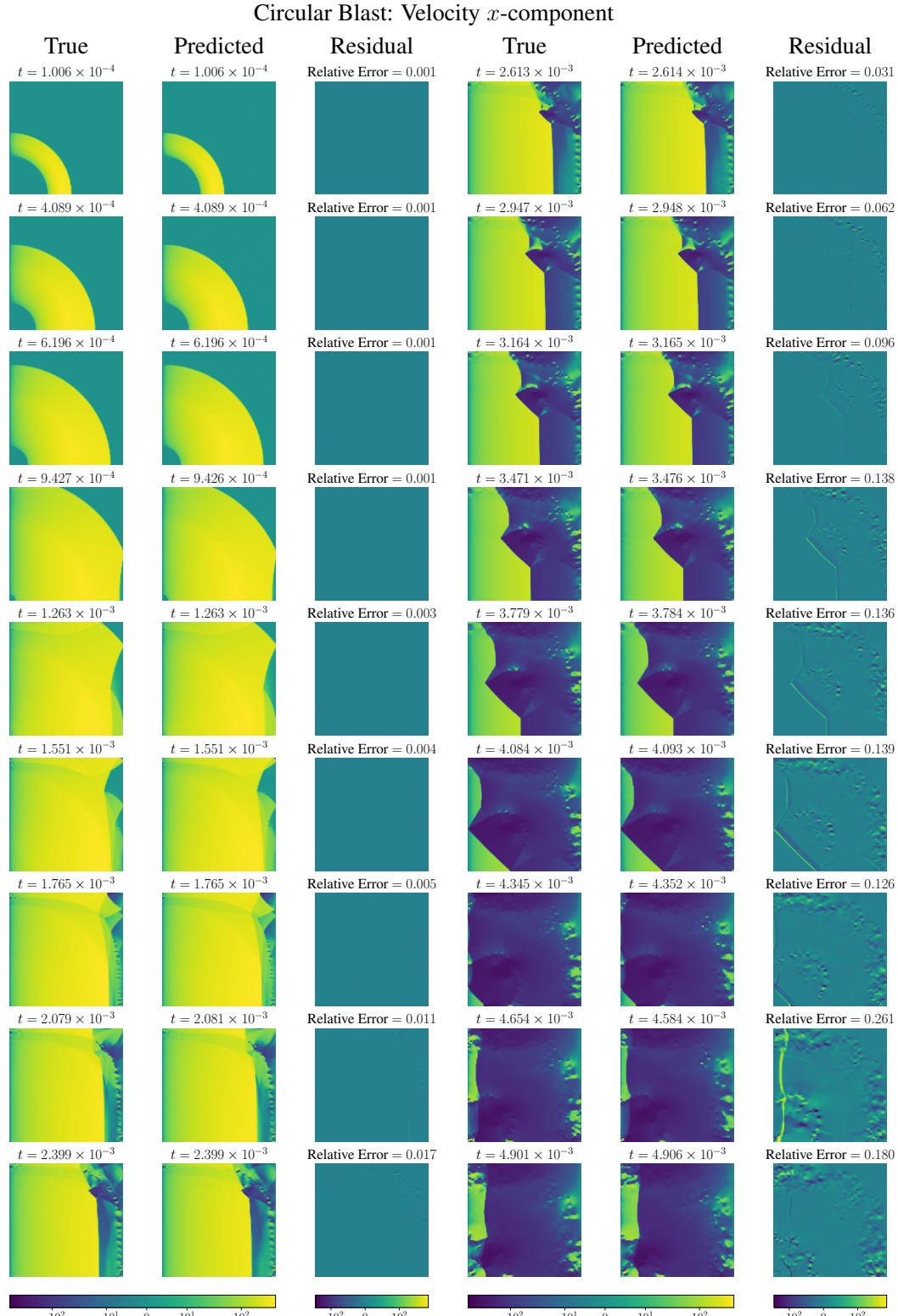

Figure 40: Velocity $x$-component for circular blast.

Circular Blast: Velocity $y$-component

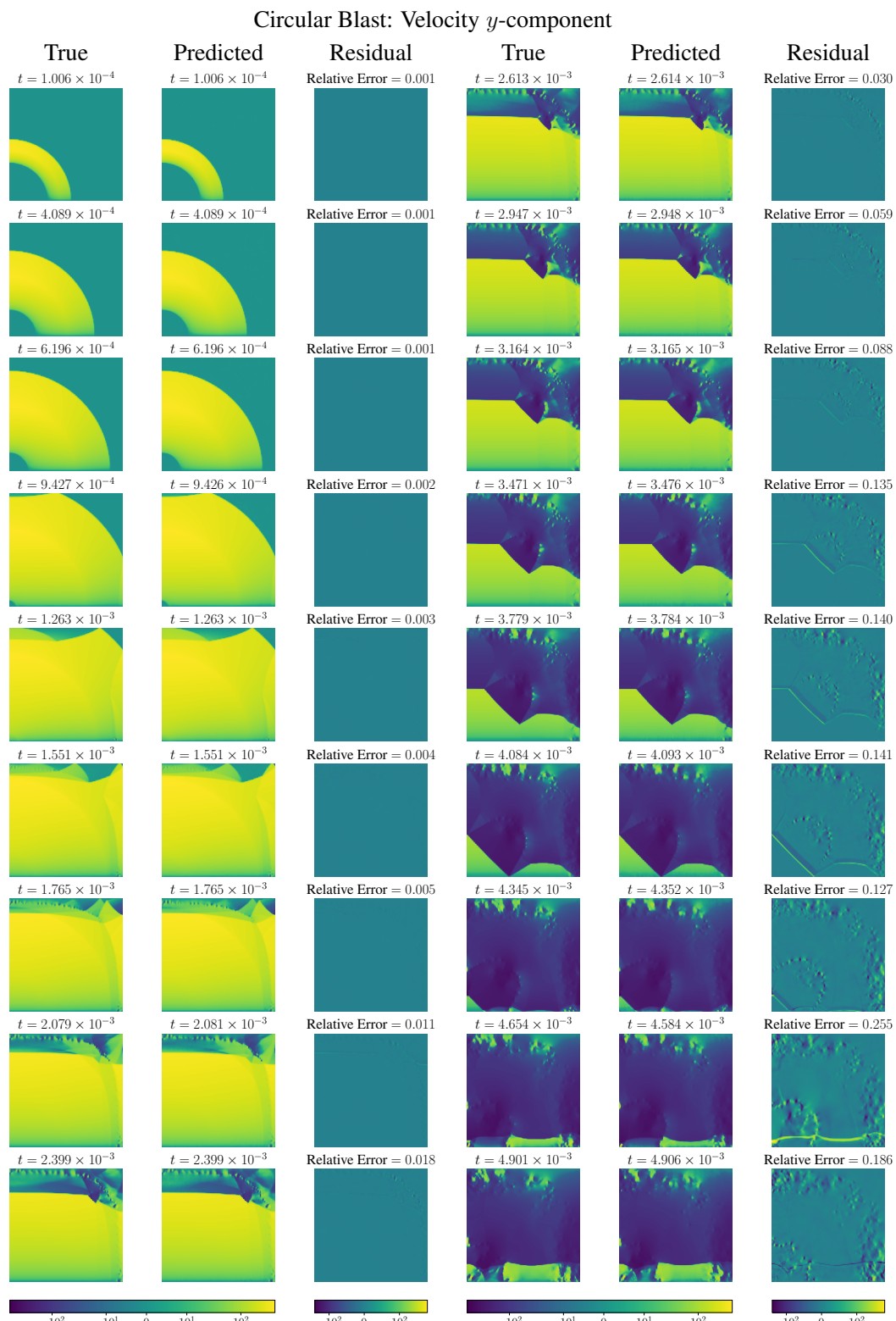

Figure 41: Velocity $y$-component for circular blast.

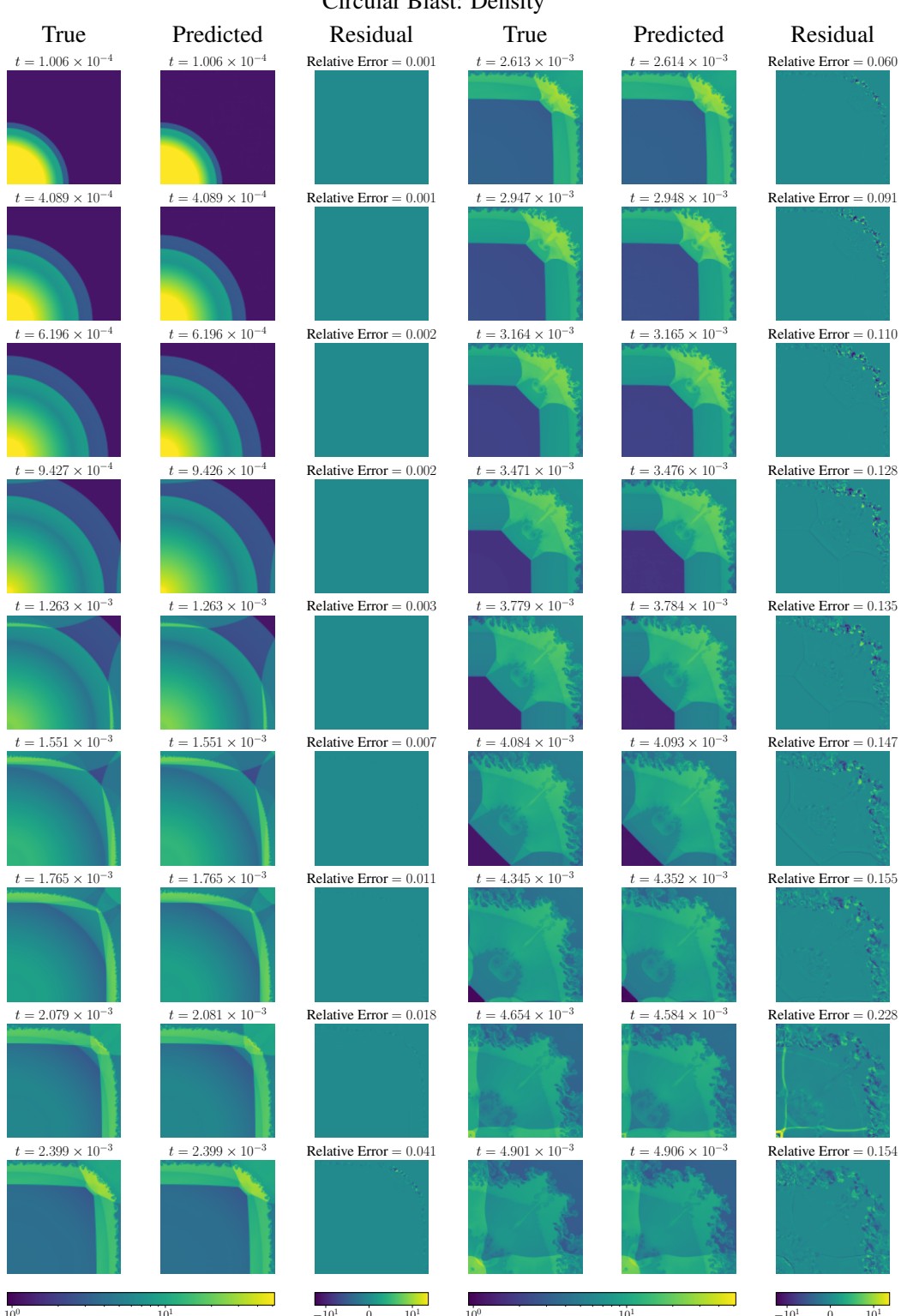

Figure 42: Density for circular blast.

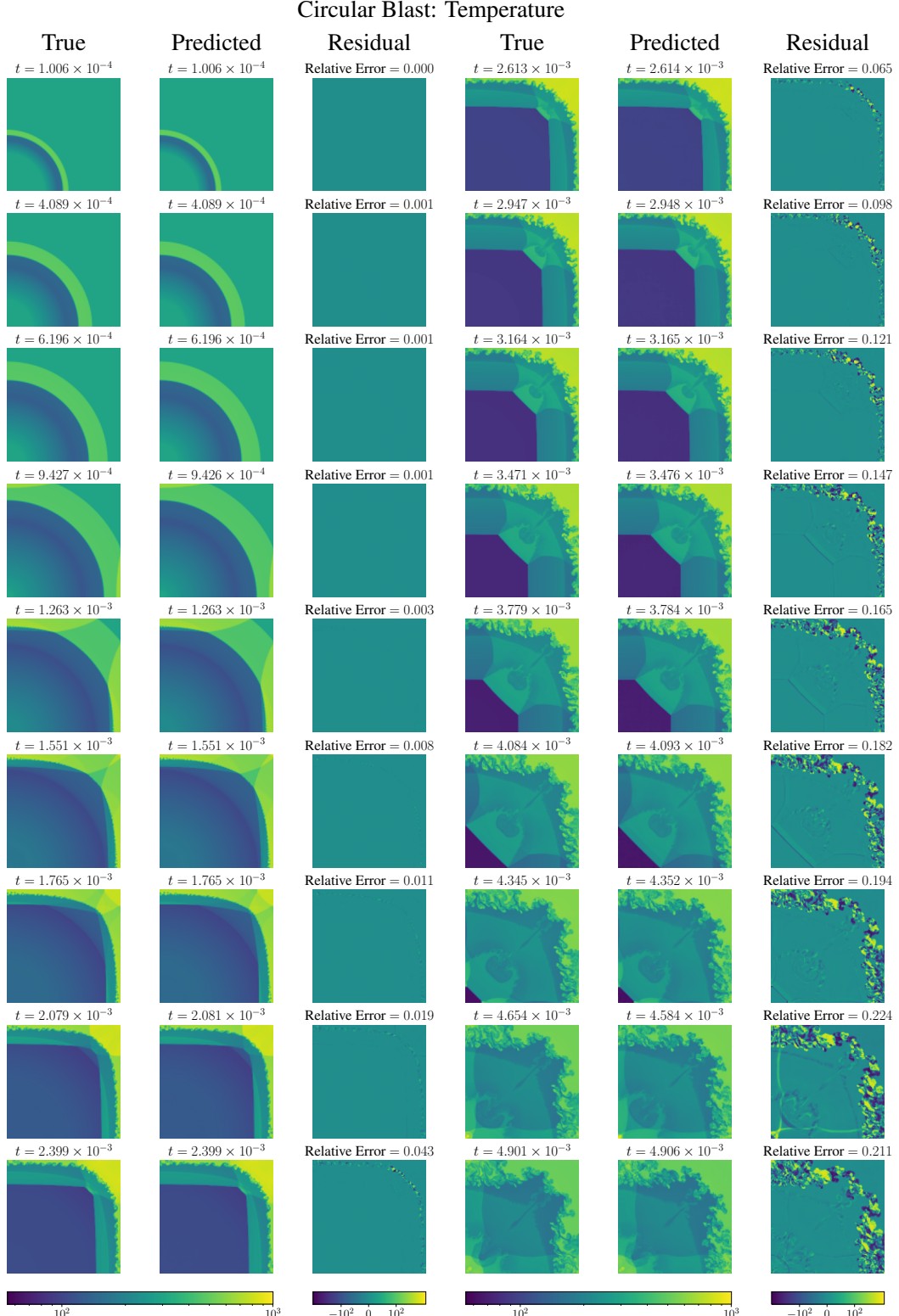

Figure 43: Temperature for circular blast.

Airfoil Shock: Velocity $x$-component

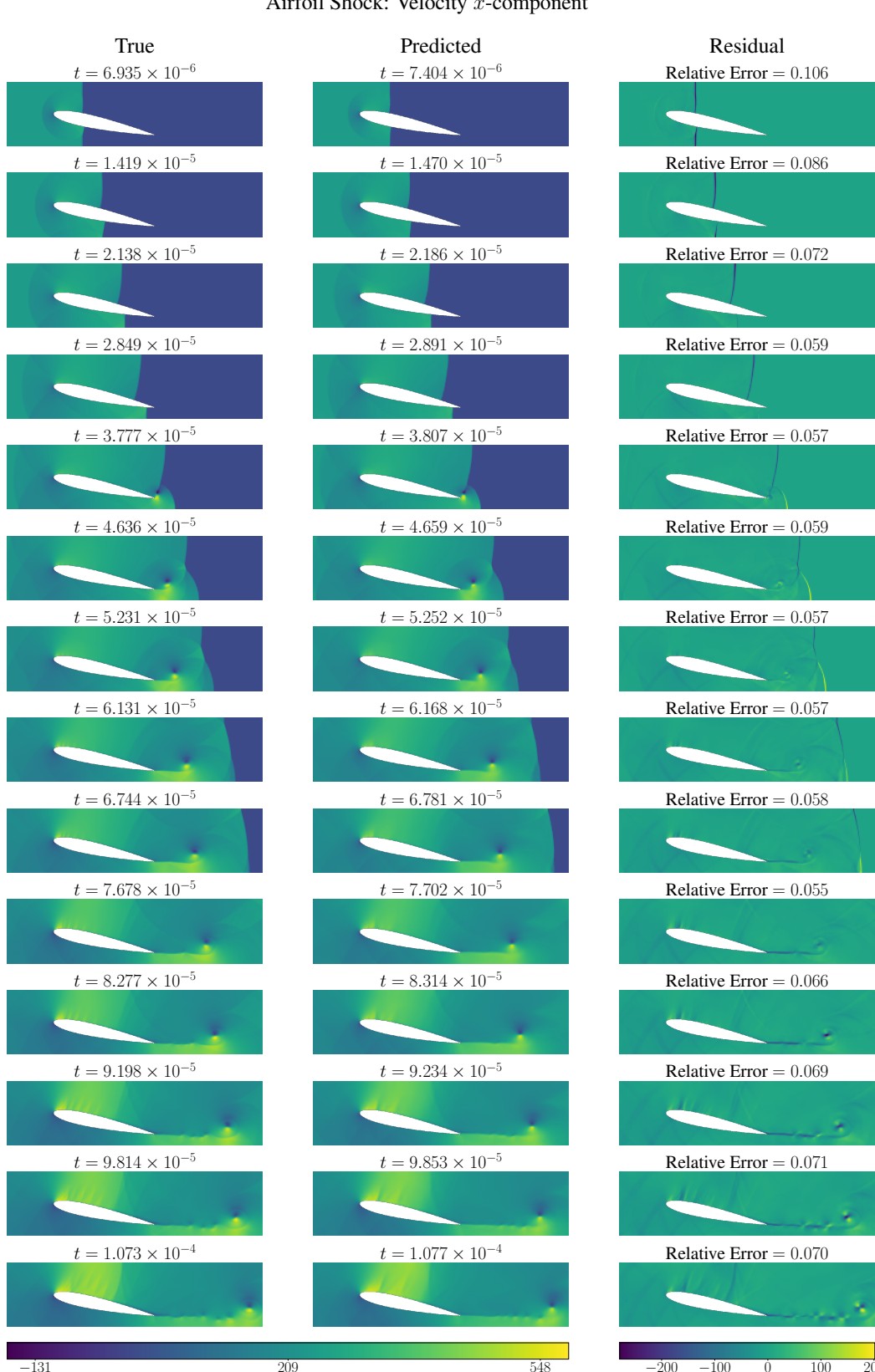

Figure 44: Velocity $x$-component for airfoil shock.

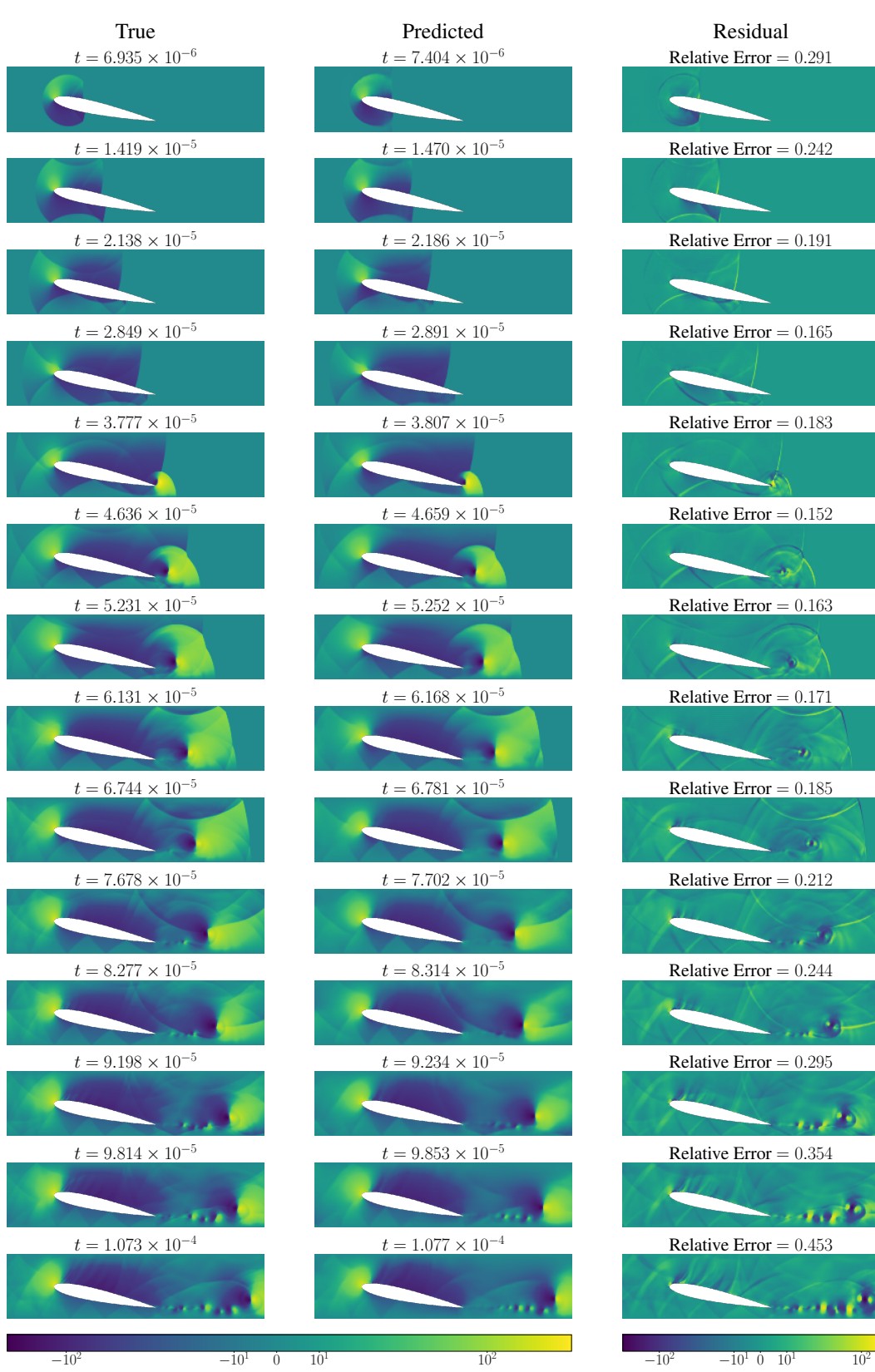

Figure 45: Velocity $y$-component for airfoil shock.

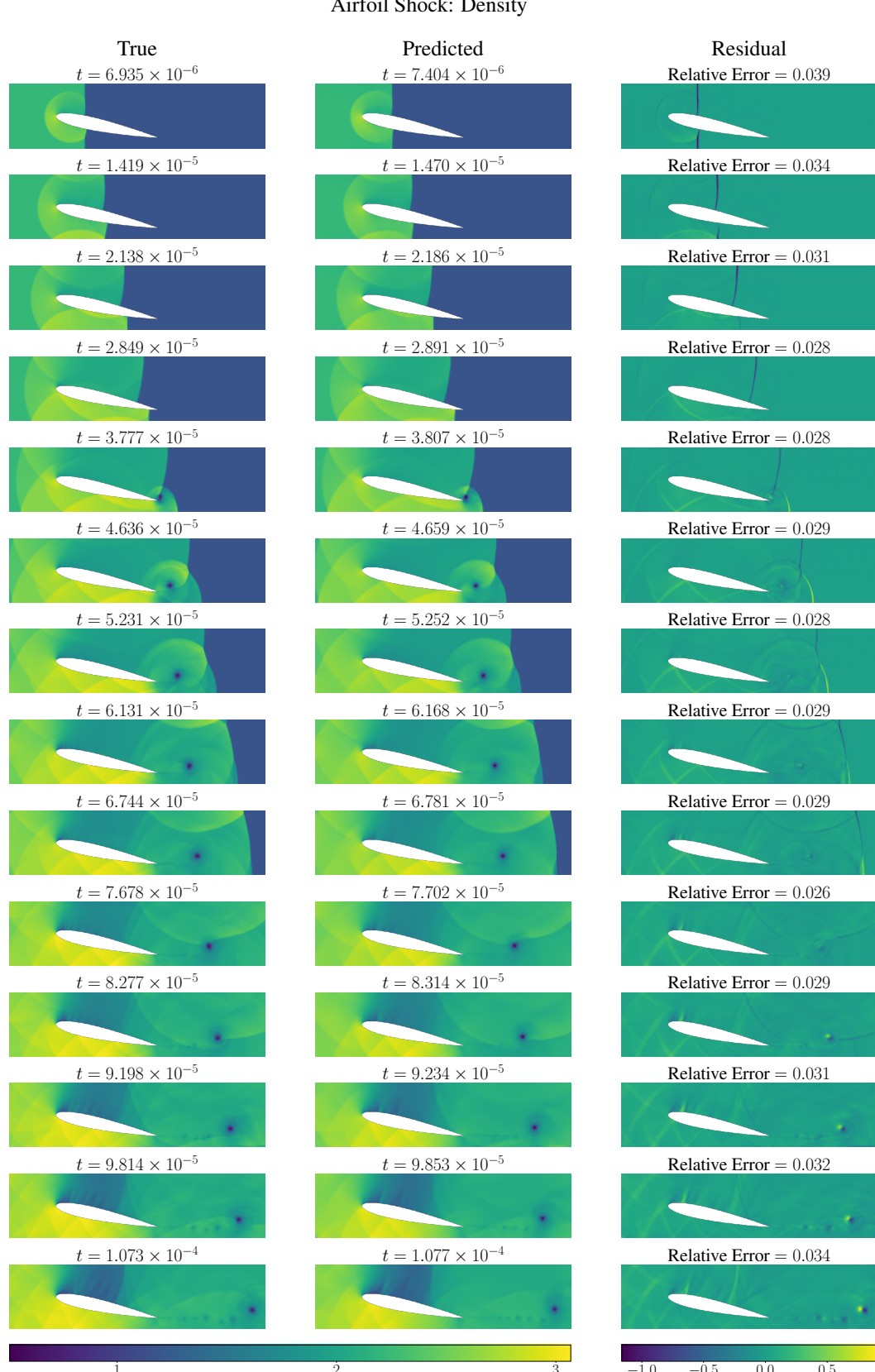

Figure 46: Density for airfoil shock.

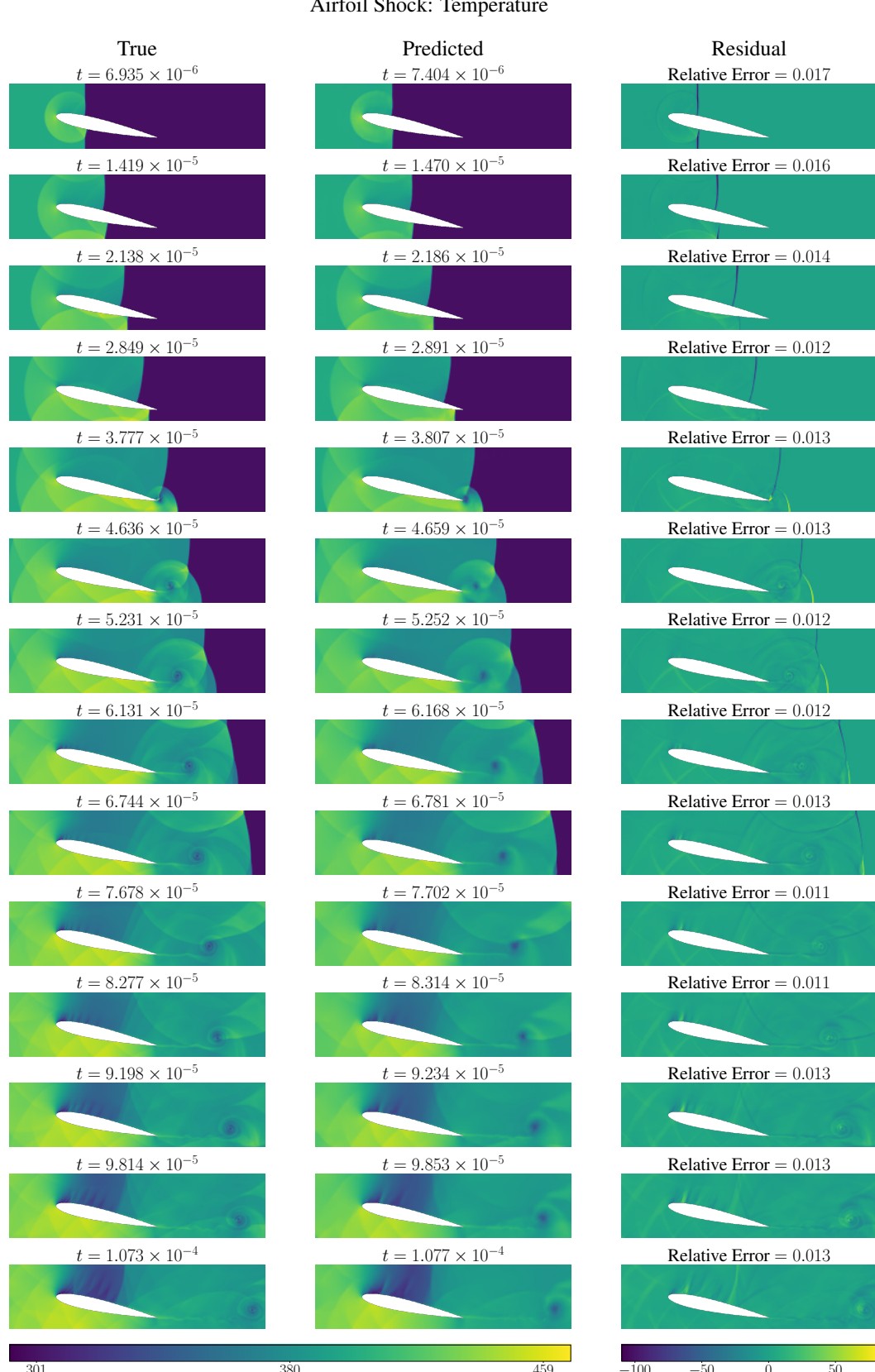

Figure 47: Temperature for airfoil shock.

| Coal Dust Explosion: One-Step Prediction Relative Error $\times 10^{-2}$ ($\downarrow$) | | | | | |
|---|---|---|---|---|---|
| Model | Gas Velocity $x$-component | Gas Velocity $y$-component | Volume Fraction | Gas Temperature | Mean |
| CNO | | | | | |
|     Affine | 0.94 (0.01) | 9.94 (0.02) | 3.03 (0.01) | 0.38 (0.00) | 3.57 (0.00) |
|     Euler | 0.97 (0.01) | 10.27 (0.08) | 3.15 (0.04) | 0.38 (0.00) | 3.69 (0.02) |
|     MoE | 0.92 (0.00) | 9.93 (0.04) | 2.86 (0.02) | 0.36 (0.00) | 3.52 (0.01) |
| F-FNO | | | | | |
|     Affine | 0.93 (0.00) | 10.27 (0.06) | 3.04 (0.01) | 0.36 (0.00) | 3.65 (0.02) |
|     Euler | 0.93 (0.00) | 10.20 (0.01) | 3.01 (0.03) | 0.36 (0.00) | 3.62 (0.01) |
|     MoE | 0.93 (0.00) | 10.33 (0.03) | 3.05 (0.01) | 0.36 (0.00) | 3.67 (0.01) |
| Transolver | | | | | |
|     Affine | 1.22 (0.02) | 12.98 (0.19) | 2.62 (0.01) | 0.44 (0.01) | 4.32 (0.05) |
|     Euler | 1.18 (0.02) | 12.83 (0.22) | 2.60 (0.04) | 0.44 (0.01) | 4.26 (0.07) |
|     MoE | 1.20 (0.01) | 12.83 (0.07) | 2.53 (0.02) | 0.43 (0.00) | 4.25 (0.02) |
| U-Net | | | | | |
|     Affine | 0.92 (0.01) | 10.32 (0.06) | 2.80 (0.02) | 0.35 (0.00) | 3.59 (0.02) |
|     Euler | 0.91 (0.01) | 10.27 (0.05) | 2.82 (0.02) | 0.35 (0.00) | 3.59 (0.02) |
|     MoE | 0.93 (0.01) | 10.36 (0.07) | 2.88 (0.01) | 0.35 (0.00) | 3.63 (0.02) |

Table 8: Relative error for one-step predictions on evaluation split of coal dust explosion cases.

## K EXTENDED RESULTS

In this section, we present the numerical values of average evaluation errors and their corresponding standard errors as *mean (standard error)*. In Tables 8 and 9, we present one-step errors for Shock-Cast. We note that the timestep predicted by the neural CFL model will not perfectly match the ground truth timestep such that the prediction from the neural solver model will be for a time which differs from the ground truth. To compute the unrolled errors in Tables 10 and 11 and correlation time proportions shown in Tables 12 and 13, we linearly interpolate ShockCast predictions in time to be sampled on the same temporal grid as the ground truth data. As can be seen in Figure 38, the volume fraction field at later timesteps can be sparse, and so we clamp the norm of the ground truth field in the denominator of the relative error to have a minimum value of 1. For the mean flow results, which we show in Tables 14 and 15, and TKE results that we present in Table 16, the target quantities involve integrating the instantaneous fields with respect to time, and thus, no interpolation is required.

## L LIMITATIONS AND FUTURE DIRECTIONS

As previously discussed, neural solvers can benefit from time-adaptive schemes, as varying the timestep size according to the rate of change can lead to more balanced one-step objectives across flow states with varying gradient sharpness. Here, we have supervised our neural CFL model using timesteps resulting from coarsening a temporal mesh computed using the CFL condition. However, approaches that learn to adapt timestep sizes based on a policy that balances solution accuracy with computational cost, as is done by Wu et al. (2022a) for spatial remeshing, may lead to further improvements. While the settings here contain dynamics comprising some of the most prevalent phenomena in high-speed flows, including shocks, blasts, and a fluid-solid interaction, future works should look to study other phenomena such as detonations and boundary layers. As discussed in Section 3.1, the use of adaptive time-stepping results in more balanced training objectives, which often results in improved generalization due to variance reduction (Duchi & Namkoong, 2019). Nevertheless, neural solvers in general do not include the same theoretical convergence guarantees enjoyed by classical methods. Future works should look to extend these results from the classical setting to machine learning methods. As discussed in Section 3.4, the MoE timestep-conditioning strategy does not leverage the efficiency advantages of sparse MoE architectures, which apply conditional computation over experts. While this design allows for a simpler implementation without load-balancing losses (Shazeer et al., 2017; Fedus et al., 2022), future work should explore incorporating

| Circular Blast: One-Step Prediction Relative Error $\times 10^{-2}$ ($\downarrow$) | | | | | |
|---|---|---|---|---|---|
| Model | Velocity $x$-component | Velocity $y$-component | Density | Temperature | Mean |
| CNO | | | | | |
|     Affine | 2.08 (0.01) | 2.08 (0.01) | 1.82 (0.01) | 1.73 (0.01) | 1.93 (0.01) |
|     Euler | 2.05 (0.01) | 2.08 (0.02) | 1.79 (0.01) | 1.69 (0.01) | 1.90 (0.01) |
|     MoE | 1.75 (0.03) | 1.75 (0.04) | 1.49 (0.01) | 1.42 (0.01) | 1.60 (0.01) |
| F-FNO | | | | | |
|     Affine | 1.87 (0.00) | 1.87 (0.00) | 1.93 (0.00) | 2.05 (0.00) | 1.93 (0.00) |
|     Euler | 1.85 (0.01) | 1.84 (0.01) | 1.93 (0.00) | 2.04 (0.00) | 1.92 (0.01) |
|     MoE | 1.87 (0.00) | 1.86 (0.00) | 2.00 (0.01) | 2.14 (0.00) | 1.97 (0.00) |
| Transolver | | | | | |
|     Affine | 1.21 (0.01) | 1.21 (0.01) | 0.95 (0.01) | 0.93 (0.01) | 1.07 (0.01) |
|     Euler | 1.27 (0.02) | 1.27 (0.02) | 1.01 (0.01) | 0.99 (0.01) | 1.14 (0.01) |
|     MoE | 1.28 (0.01) | 1.29 (0.01) | 1.02 (0.01) | 0.99 (0.01) | 1.15 (0.01) |
| U-Net | | | | | |
|     Affine | 1.52 (0.00) | 1.51 (0.01) | 1.34 (0.00) | 1.37 (0.00) | 1.44 (0.00) |
|     Euler | 1.52 (0.00) | 1.52 (0.00) | 1.34 (0.00) | 1.38 (0.01) | 1.44 (0.00) |
|     MoE | 1.71 (0.01) | 1.70 (0.01) | 1.49 (0.01) | 1.52 (0.01) | 1.61 (0.01) |

Table 9: Relative error for one-step predictions on evaluation split of circular blast cases.

| Coal Dust Explosion: Unrolled Prediction Relative Error $\times 10^{-2}$ ($\downarrow$) | | | | | |
|---|---|---|---|---|---|
| Model | Gas Velocity $x$-component | Gas Velocity $y$-component | Volume Fraction | Gas Temperature | Mean |
| CNO | | | | | |
|     Affine | 3.09 (0.08) | 45.30 (1.25) | 18.02 (1.00) | 1.22 (0.05) | 16.91 (0.59) |
|     Euler | 3.07 (0.09) | 45.71 (1.15) | 17.53 (0.34) | 1.19 (0.03) | 16.88 (0.35) |
|     MoE | 3.04 (0.02) | 45.53 (0.17) | 17.58 (0.70) | 1.18 (0.01) | 16.83 (0.15) |
| F-FNO | | | | | |
|     Affine | 2.87 (0.08) | 42.70 (0.38) | 16.72 (0.09) | 1.10 (0.01) | 15.85 (0.13) |
|     Euler | 2.80 (0.00) | 42.51 (0.66) | 16.98 (0.27) | 1.09 (0.01) | 15.84 (0.20) |
|     MoE | 2.89 (0.06) | 43.27 (0.73) | 17.16 (0.32) | 1.13 (0.01) | 16.11 (0.14) |
| Transolver | | | | | |
|     Affine | 3.33 (0.07) | 42.59 (1.20) | 19.59 (0.21) | 1.20 (0.02) | 16.68 (0.26) |
|     Euler | 3.33 (0.02) | 43.95 (0.65) | 19.26 (0.70) | 1.22 (0.03) | 16.94 (0.30) |
|     MoE | 3.21 (0.11) | 42.61 (0.84) | 19.56 (0.60) | 1.22 (0.03) | 16.65 (0.38) |
| U-Net | | | | | |
|     Affine | 3.03 (0.05) | 44.66 (0.49) | 16.26 (0.23) | 1.12 (0.01) | 16.27 (0.16) |
|     Euler | 2.93 (0.03) | 44.82 (0.40) | 16.86 (0.25) | 1.12 (0.00) | 16.43 (0.08) |
|     MoE | 3.00 (0.04) | 45.02 (0.47) | 17.12 (0.10) | 1.14 (0.01) | 16.57 (0.15) |

Table 10: Relative error for unrolled predictions on evaluation split of coal dust explosion cases.

| Circular Blast: Unrolled Prediction Relative Error $\times 10^{-2}$ ($\downarrow$) | | | | | |
|---|---|---|---|---|---|
| Model | Velocity $x$-component | Velocity $y$-component | Density | Temperature | Mean |
| CNO | | | | | |
|    Affine | 6.93 (0.08) | 6.99 (0.12) | 7.63 (0.02) | 7.96 (0.03) | 7.37 (0.05) |
|    Euler | 6.98 (0.13) | 7.06 (0.12) | 7.58 (0.06) | 7.92 (0.06) | 7.38 (0.08) |
|    MoE | 6.74 (0.06) | 6.77 (0.05) | 7.48 (0.07) | 7.85 (0.05) | 7.21 (0.01) |
| F-FNO | | | | | |
|    Affine | 5.89 (0.03) | 5.86 (0.01) | 5.59 (0.01) | 5.89 (0.01) | 5.81 (0.01) |
|    Euler | 5.70 (0.04) | 5.73 (0.06) | 5.56 (0.03) | 5.87 (0.03) | 5.71 (0.04) |
|    MoE | 5.91 (0.08) | 5.95 (0.10) | 5.68 (0.01) | 6.00 (0.03) | 5.89 (0.04) |
| Transolver | | | | | |
|    Affine | 7.35 (0.31) | 7.35 (0.32) | 7.59 (0.05) | 8.07 (0.02) | 7.59 (0.17) |
|    Euler | 7.54 (0.16) | 7.31 (0.11) | 7.63 (0.09) | 8.11 (0.04) | 7.65 (0.09) |
|    MoE | 7.27 (0.17) | 7.23 (0.23) | 7.62 (0.10) | 8.08 (0.06) | 7.55 (0.14) |
| U-Net | | | | | |
|    Affine | 5.45 (0.07) | 5.47 (0.10) | 5.16 (0.03) | 5.44 (0.04) | 5.38 (0.05) |
|    Euler | 5.50 (0.01) | 5.44 (0.02) | 5.24 (0.06) | 5.55 (0.06) | 5.43 (0.03) |
|    MoE | 5.46 (0.02) | 5.50 (0.02) | 5.30 (0.04) | 5.56 (0.03) | 5.45 (0.01) |

Table 11: Relative error for unrolled predictions on evaluation split of circular blast cases.

| Coal Dust Explosion: Correlation Time Proportion $\times 10^{-2}$ ($\uparrow$) | | | | | |
|---|---|---|---|---|---|
| Model | Gas Velocity $x$-component | Gas Velocity $y$-component | Volume Fraction | Gas Temperature | Mean |
| CNO | | | | | |
|    Affine | 80.00 (0.00) | 19.05 (0.17) | 80.04 (1.10) | 61.05 (1.06) | 60.03 (0.53) |
|    Euler | 80.00 (0.00) | 21.40 (2.96) | 78.53 (1.59) | 64.21 (0.92) | 61.04 (0.92) |
|    MoE | 80.00 (0.00) | 19.48 (0.32) | 79.27 (1.07) | 65.45 (2.44) | 61.05 (0.85) |
| F-FNO | | | | | |
|    Affine | 80.00 (0.00) | 19.76 (0.28) | 78.74 (0.32) | 65.71 (0.30) | 61.05 (0.16) |
|    Euler | 80.00 (0.00) | 20.72 (1.46) | 78.21 (0.23) | 64.86 (0.94) | 60.95 (0.13) |
|    MoE | 80.00 (0.00) | 16.17 (2.99) | 77.64 (0.68) | 65.69 (0.25) | 59.88 (0.94) |
| Transolver | | | | | |
|    Affine | 80.00 (0.00) | 21.76 (2.71) | 75.25 (0.28) | 65.19 (0.48) | 60.55 (0.76) |
|    Euler | 80.00 (0.00) | 19.23 (4.55) | 75.80 (0.69) | 64.48 (0.48) | 59.88 (1.10) |
|    MoE | 80.00 (0.00) | 19.36 (0.34) | 76.46 (0.74) | 65.06 (0.05) | 60.22 (0.17) |
| U-Net | | | | | |
|    Affine | 80.00 (0.00) | 21.90 (1.68) | 79.92 (1.13) | 63.17 (0.10) | 61.25 (0.12) |
|    Euler | 80.00 (0.00) | 20.02 (0.15) | 79.27 (1.01) | 64.27 (0.56) | 60.89 (0.26) |
|    MoE | 80.00 (0.00) | 22.08 (1.50) | 78.65 (0.96) | 63.12 (0.11) | 60.96 (0.29) |

Table 12: Correlation time proportion for unrolled predictions on evaluation split of coal dust explosion cases.

| Circular Blast: Correlation Time Proportion $\times 10^{-2}$ ($\uparrow$) | | | | | |
|---|---|---|---|---|---|
| Model | Velocity $x$-component | Velocity $y$-component | Density | Temperature | Mean |
| CNO | | | | | |
|     Affine | 100.00 (0.00) | 100.00 (0.00) | 93.38 (0.39) | 93.07 (0.23) | 96.61 (0.15) |
|     Euler | 100.00 (0.00) | 100.00 (0.00) | 93.35 (0.12) | 92.89 (0.22) | 96.56 (0.08) |
|     MoE | 100.00 (0.00) | 100.00 (0.00) | 93.04 (0.09) | 92.79 (0.06) | 96.46 (0.02) |
| F-FNO | | | | | |
|     Affine | 100.00 (0.00) | 100.00 (0.00) | 96.07 (0.42) | 95.30 (0.15) | 97.84 (0.14) |
|     Euler | 100.00 (0.00) | 100.00 (0.00) | 95.77 (0.06) | 95.49 (0.14) | 97.81 (0.05) |
|     MoE | 100.00 (0.00) | 100.00 (0.00) | 95.52 (0.06) | 94.96 (0.05) | 97.62 (0.03) |
| Transolver | | | | | |
|     Affine | 100.00 (0.00) | 100.00 (0.00) | 94.11 (0.50) | 93.58 (0.47) | 96.92 (0.24) |
|     Euler | 100.00 (0.00) | 100.00 (0.00) | 94.11 (0.39) | 93.37 (0.39) | 96.87 (0.17) |
|     MoE | 100.00 (0.00) | 100.00 (0.00) | 94.18 (0.42) | 93.09 (0.01) | 96.82 (0.10) |
| U-Net | | | | | |
|     Affine | 100.00 (0.00) | 100.00 (0.00) | 97.22 (0.84) | 96.13 (0.21) | 98.34 (0.26) |
|     Euler | 100.00 (0.00) | 100.00 (0.00) | 96.25 (0.42) | 96.10 (0.23) | 98.09 (0.16) |
|     MoE | 100.00 (0.00) | 100.00 (0.00) | 95.70 (0.07) | 95.57 (0.13) | 97.82 (0.05) |

Table 13: Correlation time proportion for unrolled predictions on evaluation split of circular blast cases.

| Coal Dust Explosion: Mean Flow Relative Error $\times 10^{-2}$ ($\downarrow$) | | | | | |
|---|---|---|---|---|---|
| Model | Gas Velocity $x$-component | Gas Velocity $y$-component | Volume Fraction | Gas Temperature | Mean |
| CNO | | | | | |
|     Affine | 1.00 (0.02) | 23.03 (0.39) | 12.42 (0.17) | 0.42 (0.01) | 9.22 (0.10) |
|     Euler | 0.97 (0.05) | 22.68 (0.86) | 12.71 (0.30) | 0.39 (0.01) | 9.19 (0.27) |
|     MoE | 0.91 (0.04) | 22.79 (0.25) | 12.25 (0.28) | 0.37 (0.01) | 9.08 (0.10) |
| F-FNO | | | | | |
|     Affine | 0.82 (0.10) | 22.59 (0.12) | 11.80 (0.12) | 0.35 (0.01) | 8.89 (0.08) |
|     Euler | 0.79 (0.04) | 22.29 (0.25) | 11.99 (0.31) | 0.32 (0.01) | 8.85 (0.13) |
|     MoE | 0.80 (0.02) | 22.44 (0.13) | 12.55 (0.24) | 0.34 (0.00) | 9.03 (0.10) |
| Transolver | | | | | |
|     Affine | 1.44 (0.09) | 23.07 (0.60) | 14.32 (0.17) | 0.39 (0.03) | 9.81 (0.10) |
|     Euler | 1.37 (0.10) | 23.47 (0.70) | 13.46 (0.13) | 0.39 (0.01) | 9.67 (0.14) |
|     MoE | 1.25 (0.16) | 22.78 (0.11) | 13.20 (0.15) | 0.40 (0.02) | 9.41 (0.09) |
| U-Net | | | | | |
|     Affine | 1.06 (0.07) | 22.50 (0.24) | 11.17 (0.30) | 0.36 (0.02) | 8.77 (0.10) |
|     Euler | 1.03 (0.05) | 23.33 (0.30) | 11.74 (0.41) | 0.37 (0.02) | 9.12 (0.18) |
|     MoE | 1.06 (0.02) | 22.53 (0.48) | 11.62 (0.25) | 0.35 (0.01) | 8.89 (0.17) |

Table 14: Relative error for mean flow on evaluation split of coal dust explosion cases.

| Circular Blast: Mean Flow Relative Error $\times 10^{-2}$ ($\downarrow$) | | | | | |
|---|---|---|---|---|---|
| Model | Velocity $x$-component | Velocity $y$-component | Density | Temperature | Mean |
| CNO | | | | | |
|     Affine | 6.25 (0.12) | 6.20 (0.24) | 2.35 (0.06) | 2.33 (0.07) | 4.28 (0.11) |
|     Euler | 6.24 (0.23) | 6.25 (0.21) | 2.30 (0.05) | 2.28 (0.06) | 4.27 (0.13) |
|     MoE | 7.94 (2.10) | 7.98 (2.14) | 2.30 (0.06) | 2.24 (0.05) | 5.12 (1.08) |
| F-FNO | | | | | |
|     Affine | 5.77 (0.22) | 5.70 (0.15) | 1.73 (0.01) | 1.68 (0.02) | 3.72 (0.10) |
|     Euler | 5.77 (0.19) | 5.81 (0.12) | 1.72 (0.02) | 1.66 (0.03) | 3.74 (0.08) |
|     MoE | 5.12 (0.13) | 5.12 (0.07) | 1.71 (0.01) | 1.69 (0.02) | 3.41 (0.03) |
| Transolver | | | | | |
|     Affine | 6.21 (0.13) | 6.04 (0.23) | 2.27 (0.01) | 2.37 (0.01) | 4.22 (0.08) |
|     Euler | 11.59 (1.40) | 10.82 (1.95) | 2.43 (0.01) | 2.49 (0.01) | 6.83 (0.83) |
|     MoE | 6.52 (0.28) | 6.30 (0.42) | 2.34 (0.05) | 2.44 (0.02) | 4.40 (0.18) |
| U-Net | | | | | |
|     Affine | 7.94 (2.63) | 7.90 (2.45) | 1.62 (0.05) | 1.53 (0.03) | 4.75 (1.29) |
|     Euler | 7.97 (2.34) | 7.92 (2.32) | 1.67 (0.05) | 1.58 (0.02) | 4.79 (1.18) |
|     MoE | 12.89 (0.02) | 12.95 (0.12) | 1.76 (0.03) | 1.63 (0.02) | 7.31 (0.05) |

Table 15: Relative error for mean flow on evaluation split of circular blast cases.

| Coal Dust Explosion: TKE Relative Error $\times 10^{-2}$ ($\downarrow$) | | Circular Blast: TKE Relative Error $\times 10^{-2}$ ($\downarrow$) | |
|---|---|---|---|
| Model | TKE | Model | TKE |
| CNO | | CNO | |
|     Affine | 11.55 (0.23) |     Affine | 2.60 (0.02) |
|     Euler | 10.91 (0.16) |     Euler | 2.69 (0.07) |
|     MoE | 10.94 (0.11) |     MoE | 2.64 (0.05) |
| F-FNO | | F-FNO | |
|     Affine | 11.01 (0.36) |     Affine | 2.29 (0.03) |
|     Euler | 10.28 (0.11) |     Euler | 2.23 (0.06) |
|     MoE | 10.93 (0.04) |     MoE | 2.16 (0.03) |
| Transolver | | Transolver | |
|     Affine | 12.72 (0.50) |     Affine | 2.68 (0.08) |
|     Euler | 12.44 (0.32) |     Euler | 2.95 (0.04) |
|     MoE | 11.79 (0.88) |     MoE | 2.77 (0.06) |
| U-Net | | U-Net | |
|     Affine | 9.85 (0.24) |     Affine | 2.27 (0.12) |
|     Euler | 9.21 (0.12) |     Euler | 2.30 (0.06) |
|     MoE | 8.91 (0.16) |     MoE | 2.36 (0.03) |

Table 16: Relative error for TKE on evaluation splits.

sparsity into MoE-based timestep conditioning. Such an approach holds strong potential for scaling time-adaptive neural solvers while balancing computational and memory cost.

## M  BROADER IMPACTS

Neural PDE solvers have been applied in accelerating dynamics simulations across a variety of real-world applications of PDE modeling, including weather and climate forecasting, aerodynamics modeling, and subsurface modeling. As neural solvers often do not include guarantees on generalization or stability over long time-integration periods, it is vital to perform rigorous validation before relying on predictions in applications. Here, we have explored the potential of neural solvers to accelerate modeling of high-speed flows. High-speed flows play an important role in the design of a variety of applications with potential for societal impact, including spacecraft, missiles, and atmospheric reentry vehicles. It is therefore important to closely monitor the development of works along this direction.

