# OpenReview forum: "A Two-Phase Deep Learning Framework for Adaptive Time-Stepping in High-Speed Flow Modeling"
_ICLR.cc/2026/Conference — ICLR 2026 Poster_

### Official Review · Reviewer_b48r · 2025-10-25

**Soundness:** 3
**Presentation:** 2
**Contribution:** 3
**Rating:** 6
**Confidence:** 4

**Summary:**

This paper proposes using an ML model to predict time steps for fluid simulations that use adaptive time steps.  It then uses those timesteps in an autoregressive fashion to solve compressible flow problems with a neural solver.

**Strengths:**

- The paper makes an interesting observation that when using a neural PDE solver, you can't necessarily rely on the CFL number computed on the fine computational mesh used for the simulation, because the neural model will effectively downsample the mesh inside the network, and so the adaptive time step chosen should be on that coarser scale.
- The numerical experiments seem convincing to me of the utility of the proposed method.

**Weaknesses:**

- I think "Our work represents the first steps towards developing machine learning models for high-speed flows" is a bit of an inflated statement that could be toned down.
- Although it's a fair point that a neural solver will have a different internal grid resolution than what is used for the PDE outside the network, presumably one knows what is the architecture they're dealing with and what that coarse grid resolution is.  Why not just compute an adaptive time step - classically - using that coarser scale (which should be known from looking at the network architecture you choose) and eschewing the proposed neural CFL predictor?
- If the answer to the above is "you could do either, but neural CFL predictor is faster while still just as accurate," can we see a study on that?

**Questions:**

- It's probably worth making the connection in the into that adaptive time stepping is not some CFD-specific thing - it is core to ML as well, e.g. "adaptive learning rates" with SGD/Adam/etc.
- Probably also worth noting in the intro that the problem with adaptive time stepping is more extreme than what you describe - if you don't have good timesteps, not only will you miss fine-scale flow phenomena, but your simulation will just crash due to predicting negative pressures etc. - before the solution ever has a chance to diverge - see e.g. Patkar et al., "Towards positivity preservation for monolithic two-way solid–fluid coupling" (in particular the issue of positivity preservation that is highlighted - even for relatively low-Mach flows)
- In 2.3, since you are interested in CFD/engineering, it may be worth noting that methods like PINNs are not convergent (they are, in their vanilla versions at least, one-shot predictors of the solution of a PDE, which can't be convergent).  Recent hybrid solver approaches like that of Kaneda et al., "A deep conjugate direction method for iteratively solving linear systems" use a neural network for preconditioning but use a classical solver loop to ensure convergence (that paper happens to be for incompressible flow, but same ideas for compressible flow systems).

---

> ### Author Response · Authors · 2025-11-28
>
> We thank the reviewer for their thoughtful suggestions and constructive feedback, which have helped strengthen the paper.
>
> ## Prior Works
>
> > *I think "Our work represents the first steps towards developing machine learning models for high-speed flows" is a bit of an inflated statement that could be toned down.*
>
> We revised the following statements:
>
> Abstract:
>
> > ~~As ShockCast is the first framework for learning high-speed flows, we~~*We* evaluate our methods by generating three supersonic flow datasets.
>
> Section 1:
>
> > We therefore develop ShockCast, ~~the first~~ *a* machine learning framework ~~(to the best of our knowledge)~~ for temporally-adaptive modeling of high-speed flows.
>
> > Our work ~~represents the first~~ *takes* steps towards developing machine learning models for high-speed flows, where there is great potential for neural acceleration due to the immense computational requirements of classical methods.
>
> Section 4:
>
> > However, to the best of our knowledge, we are the first to consider learning to temporally re-mesh ~~and utilize~~ *by utilizing* data with adaptable temporal resolution, which ~~are both~~*is* vital for developing models for high-speed flows.
>
> Section 6:
>
> > Our work ~~represents the first~~ *takes* steps towards developing machine learning models for high-speed flows, where there is great potential for neural acceleration due to the immense computational requirements of classical methods.
>
> ## Neural CFL Ablation (1/2)
>
> > *The paper makes an interesting observation that when using a neural PDE solver, you can't necessarily rely on the CFL number computed on the fine computational mesh used for the simulation, because the neural model will effectively downsample the mesh inside the network, and so the adaptive time step chosen should be on that coarser scale.*
>
> We thank the reviewer for acknowledging the novelty of our insight that the CFL condition cannot be used with temporally-adaptive neural solvers. As a clarification, this motivation is driven by downsampling in the PDE solution space as a preprocessing step, not from any internal downsampling inside the neural network. From **Section 2.3**:
>
> > Speedups over classical methods are primarily achieved by the ability of neural solvers to learn solution mappings on coarsened grids in space-time (Stachenfeld et al., 2021; Kochkov et al., 2021). To maintain stability, classical methods require computational grids to be sufficiently fine in time, as specified by the CFL condition, as well as in space. On the other hand, neural methods can learn to map between solutions spaced hundreds of classical solver steps apart on much lower-dimensional spatial discretizations, thereby realizing substantial speedups.
>
> More formally, suppose there is a PDE solution function $u(x,t):\mathbb{R}^d\times \mathbb{R}^{+}\mapsto\mathbb{R}^D$, and we have used a classical solver to sample the solution on the space time grid $\mathcal X\times\mathcal T\subset \mathbb{R}^d\times \mathbb{R}^{+}$, where for simplicity in exposition, we have assumed that the spatial grid $\mathcal X$ is constant in time (no adaptive re-meshing). Neural solvers achieve a speedup by learning on the numerical solution interpolated and/or downsampled to a lower-dimensional grid. For example, we may select a new spatial grid $\mathcal X'$ with $|\mathcal X'|\ll|\mathcal X|$ and interpolate $u(\mathcal X,t)\to u(\mathcal X',t)$ for all $t\in\mathcal T$. Similarly, we might select every 100th timestep in $\mathcal T$ to form $\mathcal T'$. Due to the ability of neural solvers to learn direct mappings without reliance on classical computation of numerical derivatives and integrals, they are much more robust to such coarsenings, and can realize substantial speedup compared to the classical solver through operating in this lower dimensional space.
>
> We are now prepared to address the reviewer’s question:

---

> ### Author Response · Authors · 2025-11-28
>
> ## Neural CFL Ablation (2/2)
>
> > *Although it's a fair point that a neural solver will have a different internal grid resolution than what is used for the PDE outside the network, presumably one knows what is the architecture they're dealing with and what that coarse grid resolution is. Why not just compute an adaptive time step - classically - using that coarser scale (which should be known from looking at the network architecture you choose) and eschewing the proposed neural CFL predictor?*
>
> > *If the answer to the above is "you could do either, but neural CFL predictor is faster while still just as accurate," can we see a study on that?*
>
> As we have discussed above, it is not an issue with the model's internal representation precluding the use of the CFL condition, but rather that neural solvers are trained on temporally coarsened data. Specifically, given a classical solver solution, we select every $J$-th step, e.g., $J=100$ in our circular blast case. This means that there are 99 CFD states between $u(t_1)$ and $u(t_2)$, with the timestep between each of these states determined using the CFL condition such that $\Delta t_1$ is actually the sum of 100 smaller timesteps. Applying the CFL condition to $u(t_1)$ will lead to a predicted  $\hat\Delta t_1$ that is on the order of 100 times smaller than the true $\Delta t_1$.
>
> Additionally, the CFL condition relies on the spacing of the spatial discretization, although ML CFD datasets apply spatial coarsening. It also requires specific field variables (e.g., temperature, sound speed), while neural solvers often model only a subset of these quantities. The latter two points result in greater misalignment when attempting to apply the CFL condition to $\hat u(t_1)$, which is likely to be more spatially coarse than the data during the generation process and may not even contain all of the fields needed to compute the condition.
>
> Because the neural solver is trained with coarsened timesteps that are roughly two orders of magnitude larger than the CFL-derived $\hat\Delta t_1$, the inputs $(u(t_1), \hat\Delta t_1)$ fall far outside the timestep range the solver was trained on, leading to worse performance. Therefore, we need a neural CFL model to map from our predicted $\hat u(t_1)$ to a step size $\hat \Delta t$ which is close to the step sizes taken in the training data for states similar to $\hat u(t_1)$ such that the pair $(\hat u(t_1),\hat \Delta t)$ is not OOD for the neural solver. Please see **Section 3.1** for a more detailed discussion of the above points.
>
> We instead perform an alternative ablation on the neural CFL model that avoids this small-timestep issue using a simpler **Mean Prediction model**. The mean prediction model replaces the neural CFL model by instead predicting every step as the mean step size $\bar \Delta t$ from the training data. We have added an ablation on the neural CFL model comparing it to an oracle model and the mean prediction model in **Appendix I**:
>
> > In this section, we evaluate and ablate the neural CFL module in ShockCast by replacing it with two alternatives: an oracle model and a mean prediction model. The oracle conditions the neural solver on the ground-truth $\Delta t$ for each timestep. As discussed in **Section 3.1**, this is not a tractable approach, as the ground truth $\Delta t$ are determined by the full solution and therefore are not known before the solution is computed. However, comparing the performance of ShockCast to the oracle model is of interest, as the neural CFL model is trained using the oracle $\Delta t$ values. Therefore, the oracle represents an upper bound on achievable performance. The mean prediction model is more viable than the oracle and a simpler alternative to the neural CFL model. At each timestep, the mean prediction model predicts the mean $\Delta t$ computed from the training data, independent of the current flow state $u(t)$.
>
> > We present results for the oracle (GT $\Delta t$) and mean prediction (Mean $\Delta t$) models compared to ShockCast using a U-Net backbone with time-conditioned layer norm in **Figures 27 and 28**. In both settings, we observe that the mean model performs substantially worse than ShockCast. This is likely because the predicted $\Delta t$ for the mean prediction model is independent of the current flow state $u(t)$, despite the dependence between these variables observed in **Figure 13**. This results in the mean timestep size $\bar\Delta t$ and current flow state $u(t)$ being an out-of-distribution input pair for the neural solver. We additionally observe that ShockCast achieves performance close to its upper bound in the oracle model. We provide comprehensive comparisons of ShockCast to the oracle model with various neural solver backbones and conditioning methods in **Tables 6 and 7**.

---

> ### Author Response · Authors · 2025-11-28
>
> ## Adaptivity in ML
>
> > *It's probably worth making the connection in the into that adaptive time stepping is not some CFD-specific thing - it is core to ML as well, e.g. "adaptive learning rates" with SGD/Adam/etc.*
>
> We have added a discussion on the connection between adaptive time stepping and other ML methods in **Section 4**:
>
> > Adaptive time-stepping extends beyond PDE modeling applications in ML research, with connections to adaptive learning rates (Duchi et al., 2011; Tieleman, 2012; Kingma & Ba, 2015) and continuous-time generative models (Song et al., 2021; Lipman et al., 2023).
>
> Duchi, John, Elad Hazan, and Yoram Singer. "Adaptive subgradient methods for online learning and stochastic optimization." Journal of machine learning research 12.7 (2011).
>
> Tieleman, Tijmen. "Lecture 6.5‐rmsprop: Divide the gradient by a running average of its recent magnitude." COURSERA: Neural networks for machine learning 4.2 (2012): 26.
>
> Kingma, Diederik P. "Adam: A method for stochastic optimization." arXiv preprint arXiv:1412.6980 (2014).
>
> Song, Yang, et al. "Score-Based Generative Modeling through Stochastic Differential Equations." International Conference on Learning Representations, 2021.
>
> Yaron Lipman, Ricky T. Q. Chen, Heli Ben-Hamu, Maximilian Nickel, and Matthew Le. Flow matching for generative modeling. In The Eleventh International Conference on Learning Representations, 2023.
>
> ## Nonphysical PDE Solutions
>
> > *Probably also worth noting in the intro that the problem with adaptive time stepping is more extreme than what you describe - if you don't have good timesteps, not only will you miss fine-scale flow phenomena, but your simulation will just crash due to predicting negative pressures etc. - before the solution ever has a chance to diverge - see e.g. Patkar et al., "Towards positivity preservation for monolithic two-way solid–fluid coupling" (in particular the issue of positivity preservation that is highlighted - even for relatively low-Mach flows)*
>
> Thank you for this reference. We have added the following discussion of the effects of poorly-chosen step sizes in **Section 2.2**:
>
> > When $u(t)$ is changing rapidly, or more formally, when $\lVert\partial_t u(t)\rVert$ grows large, too large $\Delta t$ can lead to divergence of the numerical solution (Anderson, 2023) or nonphysical solutions containing negative densities or pressures (Patkar et al., 2016).
>
> John D. Anderson. Fundamentals of Aerodynamics. McGraw Hill, New York, 7th edition, 2023.
>
> Patkar, Saket, et al. "Towards positivity preservation for monolithic two-way solid–fluid coupling." Journal of Computational Physics 312 (2016): 82-114.
>
> ## Hybrid Solvers
>
> > *In 2.3, since you are interested in CFD/engineering, it may be worth noting that methods like PINNs are not convergent (they are, in their vanilla versions at least, one-shot predictors of the solution of a PDE, which can't be convergent). Recent hybrid solver approaches like that of Kaneda et al., "A deep conjugate direction method for iteratively solving linear systems" use a neural network for preconditioning but use a classical solver loop to ensure convergence (that paper happens to be for incompressible flow, but same ideas for compressible flow systems).*
>
> We have added the following discussion of hybrid solvers to **Section 2.3**:
>
> > Various approaches have emerged over the last decade, including Physics-Informed Neural Networks (Raissi et al., 2019), hybrid solvers (Kochkov et al., 2021; Kaneda et al., 2023; Sun et al., 2023), and operator learning (Kovachki et al., 2023).
>
> Raissi, Maziar, Paris Perdikaris, and George E. Karniadakis. "Physics-informed neural networks: A deep learning framework for solving forward and inverse problems involving nonlinear partial differential equations." Journal of Computational physics 378 (2019): 686-707.
>
> Kochkov, Dmitrii, et al. "Machine learning–accelerated computational fluid dynamics." Proceedings of the National Academy of Sciences 118.21 (2021): e2101784118.
>
> Kaneda, Ayano, et al. "A deep conjugate direction method for iteratively solving linear systems." International Conference on Machine Learning. PMLR, 2023.
>
> Sun, Zhiqing, Yiming Yang, and Shinjae Yoo. "A neural pde solver with temporal stencil modeling." International Conference on Machine Learning. PMLR, 2023.
>
> Kovachki, Nikola, et al. "Neural operator: Learning maps between function spaces with applications to pdes." Journal of Machine Learning Research 24.89 (2023): 1-97.

---

### Official Review · Reviewer_WzPk · 2025-10-26

**Soundness:** 3
**Presentation:** 2
**Contribution:** 2
**Rating:** 4
**Confidence:** 5

**Summary:**

The paper proposes ShockCast, a two-phase deep learning framework for modeling high-speed flows with adaptive time-stepping.

- Phase one uses a “Neural CFL” model to predict the timestep ∆t;

- Phase two employs a timestep-conditioned neural solver to evolve the flow field.

The authors evaluate their framework on two internally generated supersonic datasets (circular blast and coal dust explosion), comparing multiple backbone architectures and conditioning strategies.

**Strengths:**

- Clear motivation: aiming to addresses PDEs under varying time intervals.

- Clear architecture modularity: Two phases modeling.

- Inclusion of physically inspired features (∇u, wave speed, CFL terms) is a positive step toward physics-awareness.

- Comprehensive validations across several backbones and conditioning methods.

- Figures are clear and the manuscript is overall readable.

**Weaknesses:**

- Conceptual shallowness: The so-called “Neural CFL” model merely regresses ∆t from local features; it does not derive from or guarantee compliance with the true CFL stability condition. There is no theoretical guarantee that predicted timesteps are stable or physically valid.

- Lack of physical consistency: The paper never checks conservation of mass, momentum, or energy, which is essential for high-Mach or compressible flows.

- Experimental validation is weak – Both datasets are self-generated 2D toy problems. Under the same physical conditions, which configuration for each benchmark is the most effective?

- No efficiency or stability analysis – The core motivation of adaptive stepping (e.g., computational savings or others) is never quantified; no wall-clock or rollout stability plots of flow evolution.

- Limited generalization – Only Mach < 3 cases are tested; unclear if the method scales to realistic hypersonic or turbulent regimes.

- Over-claimed novelty – Prior works (e.g., continuous-time neural solvers, time-conditioned FNOs, and physics-aware operator networks) already include similar temporal adaptivity ideas; Also many works for supersonic/hypersonic flow modeling (not the first machine learning framework).

- Method complexity vs. benefit – The two-phase design and multiple conditioning mechanisms add heavy machinery without showing clear improvement over a simpler time-conditioned baseline. Reporting a clear and full coparison (vs. graph-based methods, transformer-based methods, FNO-based methods, and etc. ) on one new table.

**Questions:**

- If the ∆t distribution is known in training data, why we need the two-phases modeling?

- If the two-phases modeling is usefull, the predicted ∆t from Phase one is inherently impossible for it to perfectly match the true ∆t  (based on a continuous value space) . In this case, what is the significance of designing the Phase one?

- Can you demonstrate that ShockCast preserves key physical invariants or avoids divergence during long rollouts?

- How much computational speedup or advantage (vs. one-phase modeling with time variable inputs) is achieved in practice?

- Have you tested the model on higher Mach (>5) or 3D flow cases to assess scalability?

- Have you tested the model on irregular domain cases to assess scalability?

Should you be able to satisfactorily address the points I've raised above, I will accordingly provide a positive rating.

---

> ### Author Response · Authors · 2025-11-28
>
> We thank the reviewer for their thoughtful suggestions and constructive feedback, which have helped strengthen the paper. We have added several new experiments and analyses aimed at addressing the reviewer’s concerns.
>
> ## Additional Evaluation Settings
>
> > *Experimental validation is weak – Both datasets are self-generated 2D toy problems.*
>
> In the following sections, we detail the following new experiments and analyses we have conducted in response to the reviewer's feedback:
>
> 1. **Airfoil Shock**: new dataset and experiments to assess performance on a non-uniform mesh with a non-regular domain
> 2. **Spherical Blast**: new dataset and experiments to assess performance in three spatial dimensions
> 3. **Long Circular Blast**: new dataset and experiments to assess stability over long time integration periods
> 4. **Hypersonic Airfoil Shock**: new dataset and experiments to assess performance with Mach > 5
> 5. **Neural CFL Ablation Study**: Comparison of performance of ShockCast using neural CFL model vs oracle model vs mean prediction model
> 6. **Mass Conservation Analysis**: analysis of predictions to assess physical consistency in terms of mass conservation
>
> > *Under the same physical conditions, which configuration for each benchmark is the most effective?*
>
> The Neural CFL model with all modifications has the best performance in the coal dust explosion setting, while the base neural CFL model has the best performance in the circular blast setting. This could be because the variables modeled in the circular blast setting include all those used to compute the classical CFL condition, while the more complicated form of the CFL condition for multiphase flows results in some of the variables being omitted. The UNet variant of ShockCast performs best in the coal dust explosion setting across all three metrics (correlation time, Turbulence Kinetic Energy error, and mean flow error). Meanwhile, the F-FNO variant of ShockCast performs best in the circular blast setting in all metrics except correlation time, where the U-Net has the best performance. Please see Section 5.4 for further discussion.
>
> ## Irregular Mesh
>
> > *Have you tested the model on irregular domain cases to assess scalability?*
>
> Irregular domains require non-uniform meshes, which necessitates the use of GNN-based backbones for the neural solver and neural CFL modules in ShockCast. To assess the GNN variant of ShockCast, we have added a new Airfoil Shock setting featuring an irregular domain, discussed in Section 5:
>
> >  **Airfoil Shock.** The third setting we consider is a shock-airfoil interaction problem involving a NACA 0012 airfoil immersed in a compressible flow. We initialize a normal shock inside the domain and vary its initial strength between Mach numbers $1.2$ and $2.1$. In addition, we vary the angle of attack of the airfoil between $\pm 8^\circ$, yielding a total of $100$ cases with $90$ used for training and $10$ for evaluation. Once the simulation begins, the shock travels from left to right and impinges on the airfoil, generating strong compression on the windward side followed by shock reflections and downstream vortical structures. The complexity of the flow increases with both the shock strength and the angle of attack. The irregularity of the airfoil geometry allows us to evaluate ShockCast beyond regular domains with uniform discretizations.
>
> >  Finally, for the Airfoil Shock settings, we experiment with MeshGraphNets (MGN) (Pfaff et al., 2021) and Diffusion Graph Nets (DGN) from Valencia et al. (2025), which is based on the Graph U-Net (Lino et al., 2022; Gao & Ji, 2019). Both GNN architectures use affine timestep conditioning.
>
> > Finally, we compare ShockCast with MGN and DGN neural solver backbones in **Figure 8b**, where we find MGN to be more performant in terms of correlation time and mean flow error. We visualize ShockCast predictions for a test airfoil case in **Figure 9**.
>
> We provide more details on the airfoil shock data in **Appendix A.4**, and more details on the neural solvers and the neural CFL models used for the GNN variant of ShockCast in **Appendix B**.
>
> ## 3D Flows
>
> > *Have you tested the model on higher Mach (>5) or 3D flow cases to assess scalability?*
>
> We have added experiments on a newly-generated 3D flow dataset extending the circular blast setting to three spatial dimensions in Section 5:
>
> > To assess the scalability of ShockCast to three spatial dimensions, we additionally generate and evaluate on an extension of this setting to a Spherical Blast.
>
> > In **Figure 7**, we visualize unrolled predictions for a spherical blast case using a 3D U-Net neural solver backbone with time-conditioned layer norm and 3D ConvNeXt neural CFL backbone.
>
> We provide more details on the spherical blast data in **Appendix A.3**.

---

> ### Author Response · Authors · 2025-11-28
>
> ## Stability Analysis
>
> > *No efficiency or stability analysis – The core motivation of adaptive stepping (e.g., computational savings or others) is never quantified; no wall-clock or rollout stability plots of flow evolution.*
>
> > *Can you demonstrate that ShockCast preserves key physical invariants or avoids divergence during long rollouts?*
>
> > *Conceptual shallowness: The so-called “Neural CFL” model merely regresses ∆t from local features; it does not derive from or guarantee compliance with the true CFL stability condition. There is no theoretical guarantee that predicted timesteps are stable or physically valid.*
>
> We have added an analysis of physical invariants and computational savings in the sections **Physical Compliance** and **Computational Savings** below. For rollout stability, we have added experiments on a newly-generated long version of the circular blast setting with more than 100 unrolling steps per trajectory in **Section 5**:
>
> > We additionally assess the stability of ShockCast across many time integrations by generating an alternative extension of this setting to Long Circular Blast by more than doubling the length of the simulation, resulting in trajectories with more than 100 unrolling steps.
>
> > In **Figure 8a**, we present results for the long circular blast setting for U-Net and F-FNO neural solver backbones with time-conditioned layer norm. The solutions remain correlated above 0.9 for approximately half of the simulation time, and the low TKE and mean flow errors demonstrate that the predicted solutions remain stable and in-distribution, which is further supported by the rollout visualizations shown in **Appendix H**.
>
> In **Appendix H**, we have added rollout stability plots:
>
> > In this section, we analyze performance of ShockCast in the long circular blast setting with a U-Net backbone. The rollout error over time is visualized in **Figure 22**. Although the solution predicted by ShockCast has a larger relative error as more steps are taken, the rollout visualizations in **Figures 23 to 26** show that predictions remain stable and in the distribution of solutions.
>
> ## Hypersonic Cases
>
> > *Limited generalization – Only Mach < 3 cases are tested; unclear if the method scales to realistic hypersonic or turbulent regimes.*
>
> > *Have you tested the model on higher Mach (>5) or 3D flow cases to assess scalability?*
>
> We have added experiments on a newly-generated hypersonic version of the airfoil shock setting in Appendix E, where the strength of the initial shock varies from 5 to 6:
>
> > To evaluate the ability of ShockCast to handle hypersonic (Mach $> 5$) settings, we extend the Airfoil Shock setting by varying the initial shock strength from Mach $5$ to Mach $6$. We found that despite having good performance on the airfoil shock setting, moderately-sized GNN models ($16$ message passing layers, embedding dimension of $128$) did not have sufficient capacity to accurately model the hypersonic dynamics despite having a large GPU footprint. Instead, we interpolated the irregular domain to a fine, uniformly spaced $256\times 1024$ mesh to employ a stronger U-Net model. We then lifted the input fields to latent space through a point-wise linear projection applied to the mesh points away from the airfoil surface, and a learned embedding vector for mesh points on and inside of the airfoil. Following the application of the U-Net, the airfoil surface mesh points were masked out of the loss. We visualize rollout predictions for a held-out test case in **Figures 18 to 20.**

---

> ### Author Response · Authors · 2025-11-28
>
> ## Physical Compliance
>
> > *Conceptual shallowness: The so-called “Neural CFL” model merely regresses ∆t from local features; it does not derive from or guarantee compliance with the true CFL stability condition. There is no theoretical guarantee that predicted timesteps are stable or physically valid.*
>
> While our Stability Analyses above find that ShockCast predictions do eventually diverge from the ground truth, stability plots visually confirm that they remain in the distribution of solutions without unstable behavior. Although empirical, we suggest that these validations may be more practically useful than potential theoretical guarantees that current mathematical machinery allow for. Specifically, error bounds would not only require making assumptions on the functional class of the neural solver and neural CFL model, but also assumptions on the function space of the PDE solution data -- see, for example, Assumptions 9 and 10 in Section 8.3 of [1]. However, such assumptions may not be valid for turbulent and/or high-speed flows, where even determining the existence of smooth solutions remains an open problem.
>
> [1] Kovachki, Nikola, et al. "Neural operator: Learning maps between function spaces with applications to pdes." Journal of Machine Learning Research 24.89 (2023): 1-97.
>
>
> > *Lack of physical consistency: The paper never checks conservation of mass, momentum, or energy, which is essential for high-Mach or compressible flows.*
>
> > *Can you demonstrate that ShockCast preserves key physical invariants or avoids divergence during long rollouts?*
>
> Although the chaotic nature of the flows we consider may lead to divergence of the predicted solution from the ground truth, our mean flow error and Turbulence Kinetic Energy error are used to assess the degree to which the predicted solutions statistically match the ground truth. We have also added an experiment on mass conservation in **Appendix G**:
>
> > We analyze the physical consistency of predictions made by ShockCast by examining mass conservation in the circular blast setting. As the boundaries for the blast case do not allow any fluid to enter or exit, the total mass should remain constant across time up to numerical precision. That is, the mass function $M(t):\mathbb{R}\mapsto\mathbb{R}$ defined as
>         \begin{equation*}
>             M(t)=\int_\Omega \rho(x, t)dx
>         \end{equation*}
>         should remain constant, where $\Omega$ is the spatial domain and $\rho$ is the density. To analyze the evolution of the mass function over time, we define the relative mass deviation as
>         \begin{equation*}
>             \frac{|M(t) - M(0)|}{|M(0)|},
>         \end{equation*}
>         which is positive for $t>0$ if the mass has changed since the simulation started. As can be seen in **Figure 21**, ShockCast's predicted total mass stays within $0.2$% of $M(0)$.
>
> ## Neural CFL Ablation (1/3)
>
> > *Method complexity vs. benefit – The two-phase design and multiple conditioning mechanisms add heavy machinery without showing clear improvement over a simpler time-conditioned baseline. Reporting a clear and full coparison (vs. graph-based methods, transformer-based methods, FNO-based methods, and etc. ) on one new table.*
>
> > *How much computational speedup or advantage (vs. one-phase modeling with time variable inputs) is achieved in practice?*
>
> The reviewer’s proposed one-phase baseline (which we refer to as the "oracle" in our newly-added experiments) assumes that the timestep for each evaluation step is already known at inference time. However, as detailed in **Section 3.1** and below, because the timesteps are determined by the solution, they cannot be known ahead of time. Additionally, the timesteps during inference must be chosen in a way that aligns with training. Furthermore, due to coarsening of both space and time and the fact that we only model a subset of the CFD state variables, the CFL condition used during data generation cannot be used to predict the timestep size. In summary, these three points are the primary motivation for our two phase framework:
>
> 1. Timesteps are not known before computing the solution
> 2. The CFL condition does not apply to spatially and temporally coarsened data
> 3. Timesteps must align with the timestep distribution seen during training
>
>
> We detail each of these in **Section 3.1**, and have discussed each of them below.

---

> ### Author Response · Authors · 2025-11-28
>
> ## Neural CFL Ablation (2/3)
>
> ### **1. Timesteps are not known before computing the solution**
>
> Given a neural solver $\phi$ trained using a dataset of onestep pairs with inputs $({u}(t),\Delta t)$ and targets ${u}(t+\Delta t)$, we first introduce an approach that we refer to as **the oracle** for producing the PDE solution at inference time **without the neural CFL model** by using the ground truth timesteps. Consider the evaluation trajectory
>
> $$\{(u(0),\Delta t_0), (u(t_1), \Delta t_1), \ldots\},$$
>
> where $\Delta t_i:=t_{i+1}-t_i$. Then, the oracle does the following:
>
> 1. Starting from $u(0)$, use $\Delta t_0$ to predict $\hat u(t_1)=\phi(u(0),\Delta t_0)$
> 2. Use $\hat u(t_1)$ and $\Delta t_1$ to map to $\hat u(t_2)=\phi(\hat u(t_1),\Delta t_1)$
>
> Because $\Delta t_1$ depends on the (unknown) ground-truth $u(t_1)$, it cannot be computed before the solution is obtained, making the oracle baseline infeasible in practice.
>
> ### **2. The CFL condition does not apply to spatially and temporally coarsened data**
>
> Instead of using the ground truth $\Delta t_1$, one alternative approach could be to apply the CFL condition to $u(t_1)$ to obtain $\Delta t_1$. However, neural solvers are trained on temporally coarsened data. Specifically, given a classical solver solution, we select every $J$-th step, e.g., $J=100$ in our circular blast case. This means that there are 99 CFD states between $u(t_1)$ and $u(t_2)$, with the timestep between each of these states determined using the CFL condition such that $\Delta t_1$ is actually the sum of 100 smaller timesteps. Applying the CFL condition to $u(t_1)$ will lead to a predicted  $\hat\Delta t_1$ that is on the order of 100 times smaller than the true $\Delta t_1$.
>
> Additionally, the CFL condition relies on the spacing of the spatial discretization, although ML CFD datasets apply spatial coarsening. It also requires specific field variables (e.g., temperature, sound speed), while neural solvers often model only a subset of these quantities. The latter two points result in greater misalignment when attempting to apply the CFL condition to $\hat u(t_1)$, which is likely to be more spatially coarse than the data during the generation process and may not even contain all of the fields needed to compute the condition.
>
> ### **3. Timesteps must align with the timestep distribution seen during training**
>
> Because the neural solver is trained with coarsened timesteps that are roughly two orders of magnitude larger than the CFL-derived $\hat\Delta t_1$, the inputs $(u(t_1), \hat\Delta t_1)$ fall far outside the timestep range the solver was trained on, leading to worse performance. Therefore, we need a neural CFL model to map from our predicted $\hat u(t_1)$ to a step size $\hat \Delta t$ which is close to the step sizes taken in the training data for states similar to $\hat u(t_1)$ such that the pair $(\hat u(t_1),\hat \Delta t)$ is not OOD for the neural solver. This addresses the following point from the reviewer:
>
> > *If the two-phases modeling is usefull, the predicted ∆t from Phase one is inherently impossible for it to perfectly match the true ∆t (based on a continuous value space) . In this case, what is the significance of designing the Phase one?*
>
> We do not require $\hat\Delta t_1$ to perfectly match the ground truth; it only needs to fall within the correct regime so that $(u(t_1), \hat\Delta t_1)$ remains in-distribution for the neural solver. The neural CFL model instead serves to ensure that the pair $(\hat u(t_1),\hat \Delta t)$ are in the distribution of the training data. Additionally, for the following reviewer point:
>
> > *If the ∆t distribution is known in training data, why we need the two-phases modeling?*
>
> The step sizes $\Delta t$ are a function of the flow state $u(t)$. To empirically demonstrate that this is the case, we have added **Figure 13** in **Appendix C.2** which shows that $\Delta t$ varies substantially as a function of $t$ and $u(t)$. Therefore, it is not the marginal distribution of $\Delta t$ that we are interested in, but the distribution conditional on $u(t)$. We furthermore experimentally validate this with a **Mean Prediction model**, which replaces the neural CFL model by instead predicting every step as the mean step size $\bar \Delta t$ from the training data (i.e., the mean of the marginal distribution of $\Delta t$).
>
> We have added an ablation on the neural CFL model comparing it to the oracle and the mean prediction model in **Appendix I**:

---

> ### Author Response · Authors · 2025-11-28
>
> ## Neural CFL Ablation (3/3)
>
> > In this section, we evaluate and ablate the neural CFL module in ShockCast by replacing it with two alternatives: an oracle model and a mean prediction model. The oracle conditions the neural solver on the ground-truth $\Delta t$ for each timestep. As discussed in **Section 3.1**, this is not a tractable approach, as the ground truth $\Delta t$ are determined by the full solution and therefore are not known before the solution is computed. However, comparing the performance of ShockCast to the oracle model is of interest, as the neural CFL model is trained using the oracle $\Delta t$ values. Therefore, the oracle represents an upper bound on achievable performance. The mean prediction model is more viable than the oracle and a simpler alternative to the neural CFL model. At each timestep, the mean prediction model predicts the mean $\Delta t$ computed from the training data, independent of the current flow state $u(t)$.
>
> > We present results for the oracle (GT $\Delta t$) and mean prediction (Mean $\Delta t$) models compared to ShockCast using a U-Net backbone with time-conditioned layer norm in **Figures 27 and 28**. In both settings, we observe that the mean model performs substantially worse than ShockCast. This is likely because the predicted $\Delta t$ for the mean prediction model is independent of the current flow state $u(t)$, despite the dependence between these variables observed in **Figure 13**. This results in the mean timestep size $\bar\Delta t$ and current flow state $u(t)$ being an out-of-distribution input pair for the neural solver. We additionally observe that ShockCast achieves performance close to its upper bound in the oracle model. We provide comprehensive comparisons of ShockCast to the oracle model with various neural solver backbones and conditioning methods in **Tables 6 and 7**.
>
> ## Prior works
>
> > *Over-claimed novelty – Prior works (e.g., continuous-time neural solvers, time-conditioned FNOs, and physics-aware operator networks) already include similar temporal adaptivity ideas; Also many works for supersonic/hypersonic flow modeling (not the first machine learning framework).*
>
> In **Section 4**, we have discussed the relation of our work to existing methods for time-conditioned and time-continuous neural solvers:
>
> > However, to the best of our knowledge, we are the first to consider learning to temporally re-mesh by utilizing data with adaptable temporal resolution, which is vital for developing models for high-speed flows.
>
> Additionally, we revised the following statements:
>
> Abstract:
>
> > ~~As ShockCast is the first framework for learning high-speed flows, we~~*We* evaluate our methods by generating three supersonic flow datasets.
>
> Section 1:
>
> > We therefore develop ShockCast, ~~the first~~ *a* machine learning framework ~~(to the best of our knowledge)~~ for temporally-adaptive modeling of high-speed flows.
>
> > Our work ~~represents the first~~ *takes* steps towards developing machine learning models for high-speed flows, where there is great potential for neural acceleration due to the immense computational requirements of classical methods.
>
> Section 4:
>
> > However, to the best of our knowledge, we are the first to consider learning to temporally re-mesh ~~and utilize~~ *by utilizing* data with adaptable temporal resolution, which ~~are both~~*is* vital for developing models for high-speed flows.
>
>
> Section 6:
>
> > Our work ~~represents the first~~ *takes* steps towards developing machine learning models for high-speed flows, where there is great potential for neural acceleration due to the immense computational requirements of classical methods.

---

> > ### Author Response · Authors · 2025-11-30
> >
> > ## Computational Savings
> >
> > > *How much computational speedup or advantage (vs. one-phase modeling with time variable inputs) is achieved in practice?*
> >
> > The *one-phase modeling with time-variable inputs* mentioned by the reviewer refers to our oracle model in the **Neural CFL Ablation** section above. The oracle model will be faster than ShockCast, as it does not include a neural CFL model. However, as we have discussed above, the oracle is not a feasible solution due to its use of ground truth timesteps.
> >
> > > *No efficiency or stability analysis – The core motivation of adaptive stepping (e.g., computational savings or others) is never quantified; no wall-clock or rollout stability plots of flow evolution.*
> >
> > We have added rollout stability analyses to our paper, as described in the section **Rollout Stability** above. Additionally, we agree with the reviewer's point that the manuscript was missing an analysis of computational advantages of our framework, and thus, we have added a runtime comparison of ShockCast to the classical solver in **Appendix C.5**:
> >
> > > We compare the runtime of the classical solver to ShockCast with the F-FNO neural solver backbone using affine timestep conditioning following the best practices described  in McGreivy & Hakim (2024). Specifically, we report both GPU and CPU runtimes for ShockCast, and additionally report both high-fidelity and low-fidelity runtimes for the classical solver. The high-fidelity classical solver refers to the use of the original solver settings, while low-fidelity uses a setting that trades off a degree of accuracy comparable to the neural solver error for increased efficiency. We compare using a circular blast case from the test set with initial pressure ratio $47.575$. The low fidelity solver is achieved by removing the finest level of the mesh. Times are shown in **Figure 15**, while per-step errors relative to the high-fidelity solution are shown in **Figure 16**. The high fidelity, low fidelity, and neural solutions are visualized in **Figure 17**.
> >
> > McGreivy, Nick, and Ammar Hakim. "Weak baselines and reporting biases lead to overoptimism in machine learning for fluid-related partial differential equations." Nature machine intelligence 6.10 (2024): 1256-1269.

---

### Official Review · Reviewer_1474 · 2025-10-27

**Soundness:** 4
**Presentation:** 3
**Contribution:** 3
**Rating:** 8
**Confidence:** 4

**Summary:**

This paper, ShockCast, proposes a novel two-phase deep learning framework for adaptive time-stepping in high-speed flow modeling (e.g., supersonic and hypersonic regimes). High-speed flows exhibit transient sharp gradients (shocks) requiring dynamic adjustment of the timestep size (Δt) using the Courant-Friedrichs-Lewy (CFL) condition, which is computationally expensive for classical solvers.

ShockCast addresses this by decomposing the task:

Neural CFL Phase ($\psi$): A ConvNeXt backbone predicts the optimal, large timestep $\Delta t$ based on the current flow state. This module is trained to emulate the $\Delta t$ choices from the classical solver used for data generation, circumventing issues caused by coarse computational meshes.

Neural Solver Phase ($\phi$): The flow state is evolved by the predicted $\hat{\Delta}t$. The authors introduce three novel timestep conditioning strategies for various neural solver backbones (F-FNO, U-Net, CNO, Transolver): Euler Residuals, Mixture of Experts (MoE), and Affine/Spatial-Spectral Conditioning.

The framework is evaluated on two new supersonic flow datasets—Coal Dust Explosion (multiphase) and Circular Blast (single-phase)—and achieves strong performance in accurately predicting both instantaneous fields and integrated physical quantities (TKE and Mean Flow).

**Strengths:**

Improved Training Objective for Transient Dynamics: The core motivation is strong: adaptive time-stepping naturally balances the training objective by inversely scaling $\Delta t$ according to the rate of change. This more evenly distributes the learning difficulty across states with smooth and sharp gradients (i.e., shocks), a highly pertinent consideration for high-speed flows.


Physically-Informed Neural CFL Model: The Neural CFL phase successfully emulates the true adaptive time mesh, as shown by the close match between predicted and true $\Delta t$ during autoregressive rollout. Furthermore, incorporating physically-motivated inputs like spatial gradients ($\nabla u$) and CFL features substantially improves the $\Delta t$ prediction accuracy for the complex multiphase Coal Dust Explosion scenario


Novel and Effective Conditioning Strategies: The introduction of Euler Residuals and Mixture of Experts (MoE) as timestep conditioning strategies is technically insightful. The results demonstrate that these methods are highly competitive, achieving the lowest TKE error for the Circular Blast (F-FNO backbone with MoE/Euler) and best Mean Flow/TKE performance for the Coal Dust Explosion (U-Net backbone with MoE/Euler)

**Weaknesses:**

1. Missing Quantification of Speedup: The paper's primary motivation is the immense computational cost of classical high-speed flow solvers. However, the results section fails to quantify the final speedup achieved by the full ShockCast pipeline (inference runtime) relative to the original classical solver. Without this figure, the practical utility of the entire framework remains unproven. (Only the classical solver runtime is given in Table 5, min: $\sim 15$K seconds, mean: $\sim 67$K seconds)

2. Complexity of MoE Implementation: The MoE conditioning significantly increases peak training memory (e.g., F-FNO: $18.9$ GiB (Affine) vs. $37.2$ GiB (MoE); Transolver: $41.8$ GiB (Affine) vs. $62.4$ GiB (MoE))9. While the complexity is offset by reducing the latent dimension for some models, the substantial jump in memory requirement suggests that the MoE approach is challenging to implement and scale, requiring clarification on the trade-off.

**Questions:**

Missing Speedup Quantification: The core justification is computational efficiency, yet the paper fails to state the final speedup factor (e.g., $1000\times$) of ShockCast relative to the original classical solver runtime (which can take tens of thousands of seconds). This critical number must be explicitly provided


rade-Off for MoE Complexity: The Mixture of Experts (MoE) component, while showing excellent results, dramatically increases memory consumption (e.g., up to $\sim 62$ GiB)14. The efficiency/performance trade-off needs deeper analysis, as the simpler Euler Residuals or Affine methods often perform comparably

---

> ### Author Response · Authors · 2025-11-28
>
> We thank the reviewer for their thoughtful suggestions and constructive feedback, which have helped strengthen the paper.
>
> ## Runtime Comparison
>
> > *1. Missing Quantification of Speedup: The paper's primary motivation is the immense computational cost of classical high-speed flow solvers. However, the results section fails to quantify the final speedup achieved by the full ShockCast pipeline (inference runtime) relative to the original classical solver. Without this figure, the practical utility of the entire framework remains unproven. (Only the classical solver runtime is given in Table 5, min: $\sim 15$K seconds, mean: $\sim67$K seconds)*
>
> > *Missing Speedup Quantification: The core justification is computational efficiency, yet the paper fails to state the final speedup factor (e.g., $1000\times$) of ShockCast relative to the original classical solver runtime (which can take tens of thousands of seconds). This critical number must be explicitly provided*
>
> We have added a runtime comparison of ShockCast to the classical solver in Appendix C.5:
>
> > We compare the runtime of the classical solver to ShockCast with the F-FNO neural solver backbone using affine timestep conditioning following the best practices described  in McGreivy & Hakim (2024). Specifically, we report both GPU and CPU runtimes for ShockCast, and additionally report both high-fidelity and low-fidelity runtimes for the classical solver. The high-fidelity classical solver refers to the use of the original solver settings, while low-fidelity uses a setting that trades off a degree of accuracy comparable to the neural solver error for increased efficiency. We compare using a circular blast case from the test set with initial pressure ratio $47.575$. The low fidelity solver is achieved by removing the finest level of the mesh. Times are shown in **Figure 15**, while per-step errors relative to the high-fidelity solution are shown in **Figure 16**. The high fidelity, low fidelity, and neural solutions are visualized in **Figure 17**.
>
> McGreivy, Nick, and Ammar Hakim. "Weak baselines and reporting biases lead to overoptimism in machine learning for fluid-related partial differential equations." Nature machine intelligence 6.10 (2024): 1256-1269.
>
> ## MoE Complexity
>
> > *Novel and Effective Conditioning Strategies: The introduction of Euler Residuals and Mixture of Experts (MoE) as timestep conditioning strategies is technically insightful. The results demonstrate that these methods are highly competitive, achieving the lowest TKE error for the Circular Blast (F-FNO backbone with MoE/Euler) and best Mean Flow/TKE performance for the Coal Dust Explosion (U-Net backbone with MoE/Euler)*
>
> > *Complexity of MoE Implementation: The MoE conditioning significantly increases peak training memory (e.g., F-FNO: $18.9$GiB (Affine) vs. $37.2$GiB (MoE); Transolver: $41.8$GiB (Affine) vs. $62.4$GiB (MoE))9. While the complexity is offset by reducing the latent dimension for some models, the substantial jump in memory requirement suggests that the MoE approach is challenging to implement and scale, requiring clarification on the trade-off.*
>
> > *rade-Off for MoE Complexity: The Mixture of Experts (MoE) component, while showing excellent results, dramatically increases memory consumption (e.g., up to $\sim62$GiB)14. The efficiency/performance trade-off needs deeper analysis, as the simpler Euler Residuals or Affine methods often perform comparably*
>
> We thank the reviewer for acknowledging the novelty and effectiveness of our timestep-conditioning strategies. In Section 3.4, we further clarify the computational trade-off introduced by the MoE strategy:
>
> > Because our router weights are dense, we do not leverage the conditional-computation efficiency of sparse MoE architectures (Shazeer et al., 2017; Fedus et al., 2022). As a result, the MoE conditioning mechanism trades increased model capacity for greater computational cost.
>
> In Appendix L, we've additionally added the following statement:
>
> >  As discussed in Section 3.4, the MoE timestep-conditioning strategy does not leverage the efficiency advantages of sparse MoE architectures, which apply conditional computation over experts. While this design allows for a simpler implementation without load-balancing losses (Shazeer et al., 2017; Fedus et al., 2022), future work should explore incorporating sparsity into MoE-based timestep conditioning. Such an approach holds strong potential for scaling time-adaptive neural solvers while balancing computational and memory cost.
>
> Noam Shazeer, Azalia Mirhoseini, Krzysztof Maziarz, Andy Davis, Quoc Le, Geoffrey Hinton, and Jeff Dean. Outrageously large neural networks: The sparsely-gated mixture-ofexperts layer. In International Conference on Learning Representations, 2017.
>
> Fedus, William, Barret Zoph, and Noam Shazeer. "Switch transformers: Scaling to trillion parameter models with simple and efficient sparsity." Journal of Machine Learning Research 23.120 (2022): 1-39.

---

### Official Review · Reviewer_Ssu2 · 2025-10-30

**Soundness:** 2
**Presentation:** 3
**Contribution:** 2
**Rating:** 2
**Confidence:** 3

**Summary:**

The paper proposes ShockCast, a two-phase deep learning framework for adaptive time-stepping in high-speed flow modeling. The first phase employs a Neural CFL model to predict the time-step size ($\Delta t$) based on the current flow state, while the second phase uses a time-conditioned neural solver to evolve the flow field by the predicted $\Delta t$. The authors generate two new datasets (spherical blast and coal dust explosion) and demonstrate that ShockCast effectively handles the sharp gradients and varying time scales in supersonic/hypersonic flows.

**Strengths:**

First work to address adaptive time-stepping in neural solvers for high-speed flows, filling a critical gap in ML-based CFD. Potential to accelerate simulations in aerodynamics, aerospace, and explosion modeling.

**Weaknesses:**

1.Evaluated only on two synthetic datasets (spherical blast & coal dust explosion). Training neural CFL + solver may still be expensive compared to classical adaptive methods.

2.Be overly dependent on data.

3.The proposed component is not sufficiently validated.

**Questions:**

1.How were the initial conditions (e.g., Mach numbers, pressure ratios) for the datasets chosen? Are they representative of real-world scenarios?

2.Can you visualize/analyze which features (e.g., velocity gradients, sound speed) most influence the predicted $\Delta t$?

3.Small errors in $\Delta t$ prediction may compound during autoregressive rollout. How does ShockCast handle stability over long simulations?

4.Does ShockCast generalize to unstructured meshes or three-dimensional (3D) flows?

5.The paper lacks an analysis of error accumulation beyond 100 autoregressive steps.

6.The requirement for high-fidelity solver data for training may be prohibitive for certain users.

7.Please specify the computational costs of the proposed methods and the actual speedup achieved compared to numerical simulations.

---

> ### Author Response · Authors · 2025-11-28
>
> We thank the reviewer for their thoughtful suggestions and constructive feedback, which have helped strengthen the paper. We have added several new experiments and analyses aimed at addressing the reviewer’s concerns.
>
> ## Additional Evaluation Settings
>
> > *1.Evaluated only on two synthetic datasets (spherical blast & coal dust explosion)*
>
> > *3.The proposed component is not sufficiently validated.*
>
> In the following sections, we detail the following new experiments and analyses we have conducted in response to the reviewer's feedback:
>
> 1. **Airfoil Shock**: new dataset and experiments to assess performance on a non-uniform mesh with a non-regular domain
> 2. **Spherical Blast**: new dataset and experiments to assess performance in three spatial dimensions
> 3. **Long Circular Blast**: new dataset and experiments to assess stability over long time integration periods
> 4. **Classical Solver Runtime Comparison**: a best practices comparison of ShockCast to the classical solver
> 5. **Neural CFL Feature Importance Analysis**: analysis of sensitivity of neural CFL predictions in response to perturbations of varying degrees to each feature.
>
> ## Unstructured meshes and 3D flows
>
> > *4.Does ShockCast generalize to unstructured meshes or three-dimensional (3D) flows?*
>
> Unstructured meshes necessitate the use of GNN-based backbones for the neural solver and neural CFL modules in ShockCast. To assess the GNN variant of ShockCast, we have added a new Airfoil Shock setting featuring an irregular domain, discussed in Section 5:
>
> >  **Airfoil Shock.** The third setting we consider is a shock-airfoil interaction problem involving a NACA 0012 airfoil immersed in a compressible flow. We initialize a normal shock inside the domain and vary its initial strength between Mach numbers $1.2$ and $2.1$. In addition, we vary the angle of attack of the airfoil between $\pm 8^\circ$, yielding a total of $100$ cases with $90$ used for training and $10$ for evaluation. Once the simulation begins, the shock travels from left to right and impinges on the airfoil, generating strong compression on the windward side followed by shock reflections and downstream vortical structures. The complexity of the flow increases with both the shock strength and the angle of attack. The irregularity of the airfoil geometry allows us to evaluate ShockCast beyond regular domains with uniform discretizations.
>
> >  Finally, for the Airfoil Shock settings, we experiment with MeshGraphNets (MGN) (Pfaff et al., 2021) and Diffusion Graph Nets (DGN) from Valencia et al. (2025), which is based on the Graph U-Net (Lino et al., 2022; Gao & Ji, 2019). Both GNN architectures use affine timestep conditioning.
>
> > Finally, we compare ShockCast with MGN and DGN neural solver backbones in **Figure 8b**, where we find MGN to be more performant in terms of correlation time and mean flow error. We visualize ShockCast predictions for a test airfoil case in **Figure 9**.
>
> We provide more details on the airfoil shock data in **Appendix A.4**, and more details on the neural solvers and the neural CFL models used for the GNN variant of ShockCast in **Appendix B**.
>
> We have also added experiments on a newly-generated 3D flow dataset extending the circular blast setting to three spatial dimensions in **Section 5**:
>
> > To assess the scalability of ShockCast to three spatial dimensions, we additionally generate and evaluate on an extension of this setting to a Spherical Blast.
>
> > In **Figure 7**, we visualize unrolled predictions for a spherical blast case using a 3D U-Net neural solver backbone with time-conditioned layer norm and 3D ConvNeXt neural CFL backbone.
>
> We provide more details on the spherical blast data in **Appendix A.3**.

---

> ### Author Response · Authors · 2025-11-28
>
> ## Stability Analysis
>
> > *3.Small errors in $\Delta t$ prediction may compound during autoregressive rollout. How does ShockCast handle stability over long simulations?*
>
> > *5.The paper lacks an analysis of error accumulation beyond 100 autoregressive steps.*
>
> We have added experiments on a newly-generated long version of the circular blast setting with more than 100 unrolling steps per trajectory in Section 5:
>
> > We additionally assess the stability of ShockCast across many time integrations by generating an alternative extension of this setting to Long Circular Blast by more than doubling the length of the simulation, resulting in trajectories with more than 100 unrolling steps.
>
> > In **Figure 8a**, we present results for the long circular blast setting for U-Net and F-FNO neural solver backbones with time-conditioned layer norm. The solutions remain correlated above 0.9 for approximately half of the simulation time, and the low TKE and mean flow errors demonstrate that the predicted solutions remain stable and in-distribution, which is further supported by the rollout visualizations shown in **Appendix H**.
>
> In **Appendix H**, we have added rollout stability plots:
>
> > In this section, we analyze performance of ShockCast in the long circular blast setting with a U-Net backbone. The rollout error over time is visualized in **Figure 22**. Although the solution predicted by ShockCast has a larger relative error as more steps are taken, the rollout visualizations in **Figures 23 to 26** show that predictions remain stable and in the distribution of solutions.
>
> ## Runtime Comparison
>
> > *Training neural CFL + solver may still be expensive compared to classical adaptive methods.*
>
> > *7.Please specify the computational costs of the proposed methods and the actual speedup achieved compared to numerical simulations.*
>
> We have added a runtime comparison of ShockCast to the classical solver in Appendix C.5:
>
> > We compare the runtime of the classical solver to ShockCast with the F-FNO neural solver backbone using affine timestep conditioning following the best practices described  in McGreivy & Hakim (2024). Specifically, we report both GPU and CPU runtimes for ShockCast, and additionally report both high-fidelity and low-fidelity runtimes for the classical solver. The high-fidelity classical solver refers to the use of the original solver settings, while low-fidelity uses a setting that trades off a degree of accuracy comparable to the neural solver error for increased efficiency. We compare using a circular blast case from the test set with initial pressure ratio $47.575$. The low fidelity solver is achieved by removing the finest level of the mesh. Times are shown in **Figure 15**, while per-step errors relative to the high-fidelity solution are shown in **Figure 16**. The high fidelity, low fidelity, and neural solutions are visualized in **Figure 17**.
>
> McGreivy, Nick, and Ammar Hakim. "Weak baselines and reporting biases lead to overoptimism in machine learning for fluid-related partial differential equations." Nature machine intelligence 6.10 (2024): 1256-1269.
>
> ## Data Efficiency
>
> > *2.Be overly dependent on data.*
>
> ShockCast uses a comparable amount of data to other data-driven neural solvers. To demonstrate this, we have added a visualization of the distribution of the number of timesteps per solution in **Figure 14** of **Appendix C.2**. All datasets have 90 or fewer training solutions, each with 70 or fewer timesteps, putting an upper bound of $6,300$ on the number of onestep targets, which is within the typical range used by existing neural solvers.
>
> > *6.The requirement for high-fidelity solver data for training may be prohibitive for certain users.*
>
> ShockCast is generally applicable to any setting where adaptive timestepping is used, and is independent of the fidelity of the solver used to generate the data. Additionally, ShockCast does not require the original solver resolution and is compatible with downsampled data, as we demonstrate in our experiments on temporally and spatially coarsened datasets.

---

> ### Author Response · Authors · 2025-11-28
>
> ## Initial Conditions
>
> > *1.How were the initial conditions (e.g., Mach numbers, pressure ratios) for the datasets chosen?*
>
> The initial conditions are randomly sampled from physically meaningful ranges, which we detail further in **Appendix C.1**. We've additionally added a visualization of the distribution of solution parameters in **Figure 12** of **Appendix C.2**.
>
> >  Are they representative of real-world scenarios?
>
> Yes, the parameter ranges were chosen so as to produce high-speed flow regimes commonly used in CFD to study small timescale phenomena such as shock waves, expansion fans, and compressibility effects. Standard benchmarks for evaluating classical solvers often include these settings, and they are closely related to real-world applications of CFD:
>
> 1. **Coal Dust Explosion**: modeling shock-particle interactions relevant in industrial safety/ controlled blasting.
> 2. **Circular Blast**: capturing outward-propagating shock fronts studied in blast-wave mitigation, explosive engineering, and impact modeling.
> 3. **Airfoil Shock**: designing components for aerospace vehicles, such as developing wings that minimize drag and maximize lift under a variety of operating conditions.
>
> ## Neural CFL Feature Importance Analysis
>
> > *2.Can you visualize/analyze which features (e.g., velocity gradients, sound speed) most influence the predicted $\Delta t$?*
>
> We have added a feature importance analysis for the Neural CFL model in **Appendix I**:
>
> > To assess the degree to which each of the input features to the neural CFL model influence predictions, we conduct a feature importance analysis by progressively corrupting each feature in the coal dust explosion setting. We analyze the effect this perturbation has by measuring the Mean Absolute Normalized Error, defined as
>         $$
>         \frac{|\Delta-\hat\Delta|}{\sigma_\Delta},
>         $$
>         where $\Delta$ and $\hat\Delta$ are the ground truth and predicted timesteps, and $\sigma_\Delta$ is the standard deviation of the timestep sizes over the training data. We corrupt inputs by applying dropout of increasing levels one field at a time. Results are shown in **Figure 29**. As can be seen, the flow state $u(t)$ has the largest influence on predictions, followed by the partial derivative with respect to $y$. This is consistent with the physics of the coal dust explosion setting, in which the shock repeatedly reflects between the upper and lower channel boundaries, generating strong vertical gradients that require small timesteps to resolve as the shock approaches the upper or lower boundary.

---

### Author Response · Authors · 2025-11-22
**Update: ShockCast Response**

We thank the reviewers for their thoughtful feedback and for suggesting new experiments that explore several additional settings to improve our work. We are actively running these experiments, however, because they rely on computationally-intensive simulations with our classical solver, they require significant processing time. We are preparing our response as quickly as possible and appreciate the reviewers’ patience as we complete these experiments.

---

### Author Response · Authors · 2025-11-28

We thank the reviewers for their constructive feedback and suggestions which have served to improve the paper. We have added a series of new experiments and analyses in our revised paper which we have summarized below.

## New Datasets

We have generated the following new datasets in response to reviewer feedback to thoroughly evaluate ShockCast:

1. **Airfoil Shock**: shock-airfoil interaction dataset to assess performance on a non-uniform mesh with a non-regular domain. (*Reviewer Ssu2, Reviewer WzPk*)
2. **Spherical Blast**: extension of circular blast dataset to assess performance in three spatial dimensions (*Reviewer Ssu2, Reviewer WzPk*)
3. **Long Circular Blast**: extension of circular blast dataset to assess stability over long time integration periods (*Reviewer Ssu2, Reviewer WzPk*)
4. **Hypersonic Airfoil Shock**: extension of airfoil shock dataset to assess performance with Mach > 5 (*Reviewer WzPk*)

Results on each of these datasets are added to the latest version of our paper.

---
## New Analyses

In addition to the results on the new datasets, we have added the following new analyses:
1. **Neural CFL Ablation Study**: Comparison of performance of ShockCast using neural CFL model vs oracle model vs mean prediction model (*Reviewer WzPk, Reviewer b48r*)
2. **Runtime comparison**: wall-clock comparison of ShockCast vs. the classical solver and a low-fidelity classical baseline (*Reviewer WzPk, Reviewer Ssu2, Reviewer 1474*)
2. **Mass Conservation Analysis**: analysis of predictions to assess physical consistency in terms of mass conservation (*Reviewer WzPk*)
3. **Neural CFL Feature Importance Analysis**: sensitivity study measuring how perturbing each input feature affects timestep predictions (*Reviewer Ssu2*)
---
We hope these additions fully address the reviewers’ concerns, and we thank them again for their valuable insights.

---

### Meta-Review · Area_Chair_SnN3 · 2026-01-05

**Summary:**

The paper received mixed opinions from four reviewers. Here is a summary of the major concerns from the reviewers:

1) Reviewers mentioned that the current validations on two synthetic datasets were insufficient and questioned the method’s efficacy on real-world scenarios, unstructured meshes, and 3D scenes.

2) Multiple reviewers questioned the model’s performance (e.g., computational cost and speedup), especially when compared with classical methods in numerical simulation.

3) The technical novelty over both numerical solvers and machine learning solvers needs further clarification.

4) Several reviewers noted that the method contains some complicated designs and requested further justification.

Notes on AI usage: the authors raised concerns about reviewer Ssu2, who gave the lowest score (2). The authors cited evidence from AI-detection websites indicating that their review was likely generated by AI. I have checked all four reviews with GPTZero, which indicated that Ssu2 (who gave 2, the lowest score) and 1474 (who gave 8, the highest score) are both very likely to be AI-generated. Below are the full results from my run of GPTZero:

- Reviewer Ssu2 (score 2): “We are highly confident this text was AI generated.”
- Reviewer 1474 (score 8): “We are highly confident this text was AI generated.”
- Reviewer WzPk (score 4): “We are uncertain about this document. If we had to classify it, it would likely be considered human.”
- Reviewer b48r (score 6): “We are highly confident this text is entirely human.”

I would like to add that whether a review was AI-generated is only one of many factors (e.g., the paper itself, the technical opinions expressed in each review, reviewers’ confidence and technical competence, and so on) that influence my meta-review's judgment.

**Reviewer Concerns:**

The rebuttal has responded to each concern:

1) The rebuttal reported additional experiments covering 3D scenes and unstructured meshes. The GNN variant of ShockCast that accommodates unstructured meshes is respectable, and the baseline choice (MeshGraphNet) is classic. Overall, I think the new experiments are sufficient to address this concern.

2) The rebuttal added three figures (Figs. 15-17) to report the time cost and error analysis of their model and classic simulators on CPUs/GPUs. Comparing network models with classic solvers is usually non-trivial and requires careful alignment of their computational budgets on hardware platforms. After reading these figures and the relevant text, I think this response has only partially addressed the concern, as the comparison's context is not sufficiently detailed for readers to determine whether it is fair.

3) The rebuttal has re-positioned the paper in the ML community and chose to tone down some bold claims in the original manuscript. The revised claims are more convincing, which I think can pacify the reviewers in doubt.

4) The rebuttal explained the reasons behind the Mixture of Experts (MoE) performance-complexity tradeoff and behind the two-phase design. I have found that most of the detailed justifications in the authors’ response to Reviewer WzPk are reasonable, and I will consider this issue largely resolved.

**Reviewer Scores:**

This seems to be a borderline paper in many ways, and the reviewers’ opinions are polarized.

From what I read, the two positive reviewers (6 and 8) did not raise critical concerns regarding the paper’s technical quality. I think they would have remained positive had they been able to discuss the rebuttal.

The most negative reviewer (2) raised specific questions primarily on the technical methodology and the evaluations. In my opinion, some of these evaluations are helpful but not mandatory for improving the paper, and the rebuttal has reported a reasonable amount of additional experiments to justify their claims. Given the modest confidence in this review and its relatively vague descriptions of the weaknesses, I think the reviewer may possibly be persuaded to become a borderline supporter.

I think it is hard to predict the borderline-negative reviewer (4). On one hand, for some of their concerns, I think it is easy to check whether the rebuttal has properly addressed them, e.g., higher-Mach or 3D cases. On the other hand, I feel questions about over-claimed novelty or model complexity are difficult to judge by objective standards. This reviewer is also very confident (5, absolutely certain). Weighing all these factors, I think the reviewer would likely maintain a borderline score.

Overall, I think it is more likely that the four reviewers would accept this paper than reject it if they could participate fully in the discussion.

---

### Decision · Program_Chairs · 2026-01-26

Accept (Poster)